# 🐝Artificial Hivemind: The Open-Ended Homogeneity of Language Models (and Beyond)

**Liwei Jiang**♠   **Yuanjun Chai**♠   **Margaret Li**♠   **Mickel Liu**♠   **Raymond Fok**♠
**Nouha Dziri**★   **Yulia Tsvetkov**♠   **Maarten Sap**◇   **Yejin Choi**♡

♠University of Washington     ◇ Carnegie Mellon University
★Allen Institute for Artificial Intelligence   ♣Lila Sciences   ♡Stanford University

lwjiang@cs.washington.edu

 **Code**: https://github.com/liweijiang/artificial-hivemind
🤗 **INFINITY-CHAT Collection**: liweijiang/artificial-hivemind

## Abstract

Large language models (LMs) often struggle to generate diverse, human-like creative content, raising concerns about the long-term homogenization of human thought through repeated exposure to similar outputs. Yet scalable methods for evaluating LM output diversity remain limited, especially beyond narrow tasks such as random number or name generation, or beyond repeated sampling from a single model. To address this gap, we introduce INFINITY-CHAT, a large-scale dataset of 26K diverse, real-world, *open-ended user queries that admit a wide range of plausible answers with no single ground truth*. We introduce the first *comprehensive taxonomy* for characterizing the full spectrum of open-ended prompts posed to LMs, comprising 6 top-level categories (e.g., creative content generation, brainstorm & ideation) that further breaks down to 17 subcategories. Using INFINITY-CHAT, we present a large-scale study of mode collapse in LMs, revealing a pronounced **Artificial Hivemind** effect in open-ended generation of LMs, characterized by (1) *intra-model repetition*, where a *single* model consistently generates similar responses, and more so (2) *inter-model homogeneity*, where *different* models produce strikingly similar outputs. INFINITY-CHAT also includes 31,250 human annotations, across absolute ratings and pairwise preferences, with 25 independent human annotations per example. This enables studying collective and individual-specific human preferences in response to open-ended queries. Our findings show that state-of-the-art *LMs*, *reward models*, and *LM judges* are less well calibrated to human ratings on model generations that elicit differing idiosyncratic annotator preferences, despite maintaining comparable overall quality. Overall, INFINITY-CHAT presents the first large-scale resource for systematically studying *real-world open-ended queries* to LMs, revealing critical insights to guide future research for mitigating long-term AI safety risks posed by the *Artificial Hivemind*.

## 1   Introduction

Large language models (LMs) are the core backbone of modern AI systems but often fail to produce the diverse, human-like creativity expected in open-ended tasks that do not have a ground truth answer [88, 93, 92, 52]. This shortfall has sparked growing concerns about the long-term homogenization of human thought, as users are repeatedly exposed to similar outputs [79, 10]. While ensemble methods or model "swarms" have been proposed to enhance diversity [27, 25], their scalable evaluations of diversity in real-world settings are still lacking [31]. Existing benchmarks often target stylized tasks

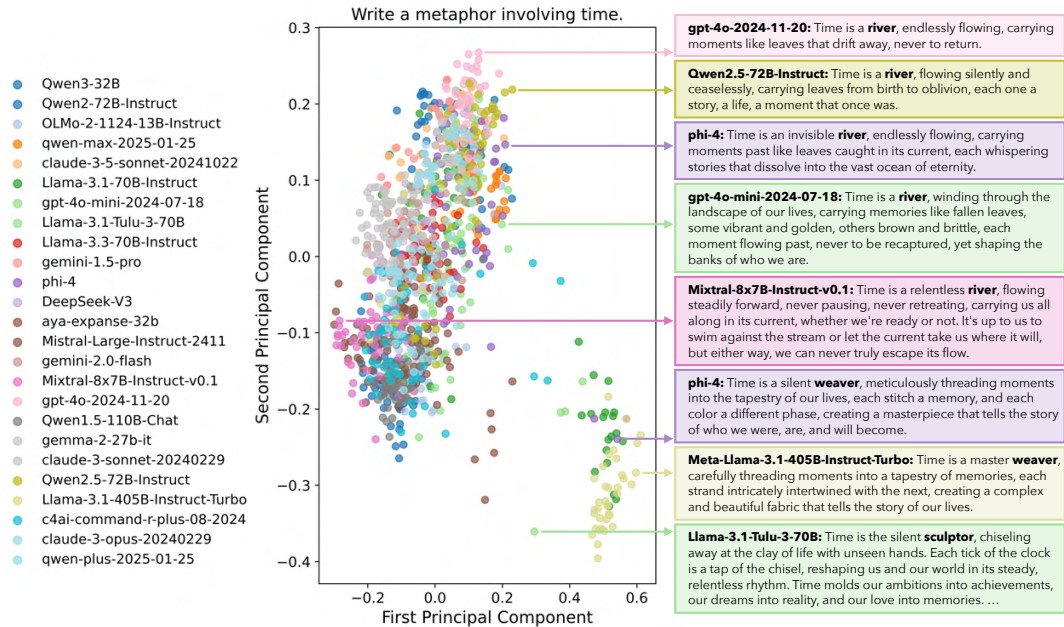

Figure 1: Responses to the query **"Write a metaphor about time"** clustered by applying PCA to reduce sentence embeddings to two dimensions. Each of the 25 models generates 50 responses using top-$p$ sampling ($p = 0.9$) and temperature $= 1.0$. Despite the diversity of model families and sizes, the responses form just two primary clusters: a dominant cluster on the left centered on the metaphor "time is a river," and a smaller cluster on the right revolving around variations of "time is a weaver."

such as persona generation [30], keyword-driven storytelling [13], or random number generation [93, 88], and often rely on narrowly defined tests centered on poetry or figurative language [64, 92]. Yet, these settings fail to capture the open-endedness and pluralism of real-world user interactions.

We introduce INFINITY-CHAT, a large-scale dataset of 26K real-world open-ended queries spanning diverse, naturally occurring prompts mined from WildChat [94]. These queries admit a wide range of plausible answers with no single correct response. We further develop the first comprehensive taxonomy of open-ended LM queries, encompassing 6 top-level categories (e.g., Brainstorm & Ideation, and less explored types such as Speculative & Hypothetical Scenarios, and Skill Development) and 17 subcategories grounded in natural chatbot-user interactions.

Using INFINITY-CHAT, we systematically study *intra-* and *inter-model mode collapse* across 70+ open and closed source LMs (25 detailed in the main paper). We uncover a pronounced **Artificial Hivemind** effect: (1) *intra-model repetition*, where a *single* model repeatedly generates similar outputs, and, more critically, (2) *inter-model homogeneity*, where *different* models independently converge on similar ideas with minor variations in phrasing. The latter warns that model ensembles may not yield true diversity when their constituents share overlapping alignment and training priors.

Beyond generative behaviors, we also examine *whether LMs are calibrated to assess alternative responses of comparable quality* to open-ended queries. To enable this study, we collect 31,250 human annotations on distinct model responses in INFINITY-CHAT, encompassing both absolute quality ratings and pairwise preferences, with *dense annotations from 25 independent annotators per query–response pair*. Our results show that LMs, reward models, and LM-based judges are often miscalibrated with respect to human ratings on *responses that elicit divergent, idiosyncratic preferences among annotators despite comparable overall quality*. This exposes key limitations in current modeling pipelines, which tend to assume a single, consensus notion of quality and thus overlook or fail to reward the diverse, pluralistic preferences that arise in open-ended responses.

Altogether, our work introduces a comprehensive framework for evaluating realistic open-endedness, diversity, and pluralistic alignment in LMs, both within and across LMs. By integrating real-world queries, a taxonomy of query types, and dense human annotations, INFINITY-CHAT provides a useful resource for diagnosing the **Artificial Hivemind** effect and for guiding the development of safer, more expressive, and more resourceful LMs that better empower human creativity.

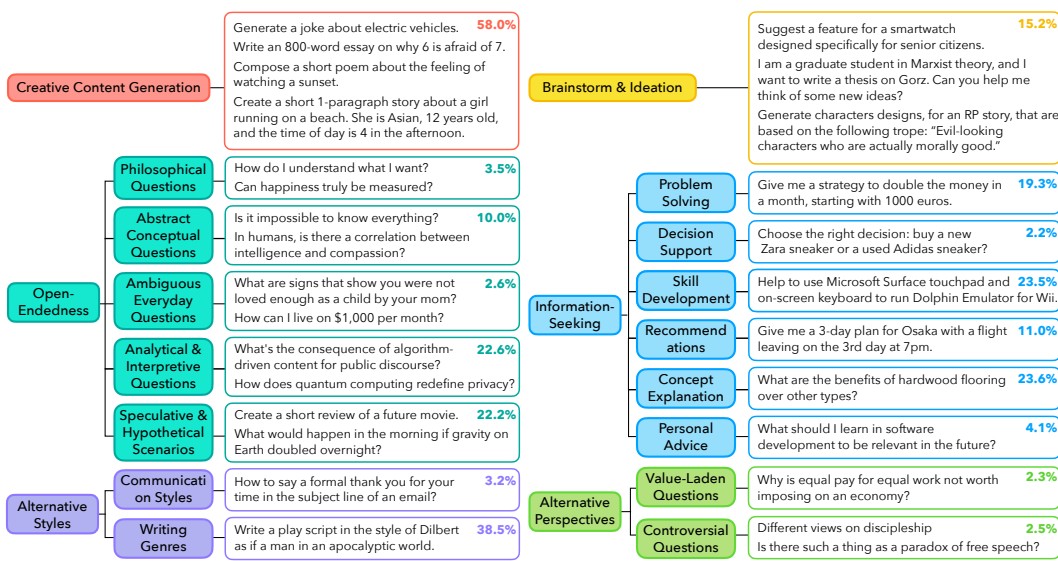

Figure 2: A taxonomy of **real-world open-ended queries that invite diverse model responses** that are mined from in-the-wild user-chatbot interactions, categorized into 6 top-level and 17 fine-grained subcategories, along with their occurrence percentages.

## 2 INFINITY-CHAT: Real-World Open-Ended Queries with Diverse Responses

Most existing LM alignment datasets prioritize response correctness over diversity, and rarely include multiple distinctive responses to the same prompt. This overlooks the inherent variability of open-ended queries, which often admit several equally valid answers. This gap motivates our first central research question: *What types of open-ended queries do users actually pose to language models?*

**Mining in-the-wild open-ended user queries.** We construct INFINITY-CHAT, a dataset of real-world open-ended queries to language models, by filtering and refining user inputs from WildChat [94]. From 37,426 high-quality, single-turn GPT-4 queries (English, non-toxic, 15–200 characters), GPT-4o classifies each by whether it seeks meaningful information, is a greeting or model inquiry, and allows single or multiple valid responses. Ambiguous queries are revised for clarity. The result is an extensive collection of 26,070 open-ended and 8,817 closed-ended queries, which elicit diverse, high-quality LM responses. Full details of the query mining process are provided in §Appendix B.1.

**Categorizing the diverse landscape of open-ended queries.** To understand the types of open-ended queries users pose to LMs, we develop a taxonomy of fine-grained categories. We adopt a semi-automatic process to construct the taxonomy. Starting with ~100 mined queries, we manually assign tentative labels, then iteratively refine and group them into a hierarchical structure. This results in 6 high-level categories and 17 fine-grained sub-categories, as shown in Figure 2. Next, we scale the annotation process to the full set of open-ended queries using GPT-4o. We instruct GPT-4o to label each user query

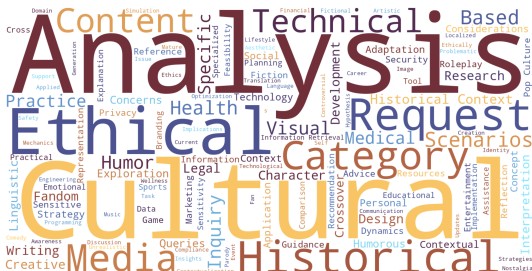

Figure 3: A word cloud visualizing **new open-ended categories** mined from in-the-wild queries.

with one or more of the existing open-ended categories, and to detect novel types beyond the seed categories. Full details of the taxonomy construction process are provided in §Appendix B.2.

As shown in Figure 2, while Creative Content Generation dominates (58.0%), we identify several underexplored yet popular types, such as Alternative Writing Genres (38.5%), Concept Explanation (23.6%), Skill Development (23.5%), Analytical & Interpretive Questions (22.6%), and Hypothetical Scenarios (22.2%). Notably, 15.2% of queries involve Brainstorming & Ideation, underscoring users' reliance on LMs for direct ideas and inspirations, and raising concerns about the long-term risk of homogenized thinking driven by overly uniform AI outputs.

| Similarity Score Ranges | Average | gpt-4o-2024-11-20 | gpt-4o-mini-2024-07-18 | claude-3-5-sonnet-20241022 | claude-3-sonnet-20240229 | claude-3-opus-20240229 | Llama-3.3-70B-Instruct | Llama-3.1-70B-Instruct | Llama-3.1-405B-Instruct-Turbo | gemma-2-27b-it | gemini-2.0-flash | gemini-1.5-pro | qwen-max-2025-01-25 | qwen-plus-2025-01-25 | Qwen3-32B | Qwen2.5-72B-Instruct | Qwen2-72B-Instruct | Qwen1.5-110B-Chat | Mistral-Large-Instruct-2411 | Mixtral-8x7B-Instruct-v0.1 | OLMo-2-1124-13B-Instruct | Llama-3.1-Tulu-3-70B | c4ai-command-r-plus-08-2024 | aya-expanse-32b | DeepSeek-V3 | phi-4 |
|---|---|---|---|---|---|---|---|---|---|---|---|---|---|---|---|---|---|---|---|---|---|---|---|---|---|---|
| 0.9-1.0 | 43.8 | 51.0 | 53.0 | 61.0 | 48.0 | 59.0 | 51.0 | 23.0 | 43.0 | 33.0 | 40.0 | 53.0 | 55.0 | 56.0 | 40.0 | 48.0 | 35.0 | 48.0 | 43.0 | 45.0 | 29.0 | 28.0 | 24.0 | 50.0 | 42.0 | 38.0 |
| 0.8-0.9 | 35.2 | 36.0 | 34.0 | 22.0 | 36.0 | 27.0 | 30.0 | 44.0 | 38.0 | 43.0 | 41.0 | 32.0 | 37.0 | 28.0 | 36.0 | 26.0 | 35.0 | 36.0 | 34.0 | 41.0 | 39.0 | 39.0 | 32.0 | 36.0 | 39.0 | 40.0 |
| 0.7-0.8 | 12.6 | 10.0 | 9.0 | 9.0 | 10.0 | 7.0 | 12.0 | 23.0 | 11.0 | 14.0 | 11.0 | 11.0 | 4.0 | 11.0 | 14.0 | 19.0 | 24.0 | 10.0 | 15.0 | 9.0 | 12.0 | 18.0 | 19.0 | 8.0 | 13.0 | 11.0 |
| 0.6-0.7 | 4.9 | 1.0 | 4.0 | 5.0 | 5.0 | 3.0 | 4.0 | 5.0 | 3.0 | 7.0 | 4.0 | 3.0 | 3.0 | 3.0 | 6.0 | 4.0 | 5.0 | 2.0 | 5.0 | 4.0 | 9.0 | 11.0 | 10.0 | 5.0 | 3.0 | 9.0 |
| 0.5-0.6 | 1.9 | 2.0 | 0.0 | 2.0 | 1.0 | 4.0 | 1.0 | 2.0 | 4.0 | 0.0 | 2.0 | 1.0 | 1.0 | 2.0 | 1.0 | 1.0 | 0.0 | 3.0 | 2.0 | 1.0 | 7.0 | 1.0 | 7.0 | 1.0 | 0.0 | 2.0 |
| 0.4-0.5 | 0.7 | 0.0 | 0.0 | 1.0 | 0.0 | 0.0 | 1.0 | 2.0 | 0.0 | 2.0 | 0.0 | 0.0 | 0.0 | 0.0 | 1.0 | 1.0 | 0.0 | 0.0 | 0.0 | 0.0 | 2.0 | 2.0 | 3.0 | 0.0 | 3.0 | 0.0 |
| 0.3-0.4 | 0.6 | 0.0 | 0.0 | 0.0 | 0.0 | 0.0 | 1.0 | 0.0 | 0.0 | 1.0 | 2.0 | 0.0 | 0.0 | 0.0 | 2.0 | 1.0 | 1.0 | 1.0 | 1.0 | 0.0 | 1.0 | 0.0 | 3.0 | 0.0 | 0.0 | 0.0 |
| 0.2-0.3 | 0.2 | 0.0 | 0.0 | 0.0 | 0.0 | 0.0 | 0.0 | 1.0 | 1.0 | 0.0 | 0.0 | 0.0 | 0.0 | 0.0 | 0.0 | 0.0 | 0.0 | 0.0 | 0.0 | 0.0 | 1.0 | 1.0 | 1.0 | 0.0 | 0.0 | 0.0 |
| 0.1-0.2 | 0.0 | 0.0 | 0.0 | 0.0 | 0.0 | 0.0 | 0.0 | 0.0 | 0.0 | 0.0 | 0.0 | 0.0 | 0.0 | 0.0 | 0.0 | 0.0 | 0.0 | 0.0 | 0.0 | 0.0 | 0.0 | 0.0 | 1.0 | 0.0 | 0.0 | 0.0 |
| 0.0-0.1 | 0.0 | 0.0 | 0.0 | 0.0 | 0.0 | 0.0 | 0.0 | 0.0 | 0.0 | 0.0 | 0.0 | 0.0 | 0.0 | 0.0 | 0.0 | 0.0 | 0.0 | 0.0 | 0.0 | 0.0 | 0.0 | 0.0 | 0.0 | 0.0 | 0.0 | 0.0 |

Figure 4: The heatmap shows **degree of repetition in responses to open-ended queries generated by the same LMs**. For each model, we generate 50 responses per query across 100 open-ended queries from INFINITY-CHAT100. We then compute the average pairwise sentence embedding similarities for each query's response pool and measure the percentage of queries falling into the similarity ranges indicated on the y-axis. Under the sampling parameters (top-$p = 0.9$, temperature $= 1.0$), the average pairwise similarity among responses to the same prompt typically exceeds 0.8. As a baseline, randomly paired responses from the global pool 100% fall within the 0.1–0.2 range.

In addition to our pre-defined categories, we identify 314 novel ones. Figure 3 visualizes a word cloud of the most prominent keywords, such as "Cultural," "Analysis," "Ethical," "Historical," "Media," and "Humor," highlighting previously underexplored dimensions of open-ended query categories.

With INFINITY-CHAT, we introduce the first comprehensive taxonomy of real-world open-ended queries that invite diverse responses. This dataset serves as a rich resource for studying LMs' capacity to generate varied appropriate outputs, and for advancing pluralistic alignment of LMs.

## 3 Artificial Hivemind: Intra- and Inter-Model Homogeneity in LMs

Using a subset of 100 representative open-ended queries from INFINITY-CHAT (denoted **INFINITY-CHAT100**, human verified to be open-ended as detailed in §Appendix B.3), we systematically examine the "Artificial Hivemind" of LMs. We focus on two aspects: (1) **intra-model repetition**, where the same LM fails to generate diverse outputs, and (2) **inter-model homogeneity**, where different models produce similar outputs. Prior studies have explored intra-model repetition at small scales or with synthetic tasks (e.g., random number/name generation) [88, 93]. In contrast, we conduct a large-scale study on real-world open-ended questions, spanning 70+ LMs (25 detailed in the main paper, representing the strongest or largest models from major model families), providing the first systematic analysis of cross-model output convergence. Full experimental setup, complete model results, and examples are provided in §Appendix C.

**Intra-model repetition.** For each model, we sample 50 responses per query from INFINITY-CHAT100, compute the average pairwise embeddings similarity within each response pool[1], and report the percentage of queries falling into different similarity ranges. Despite using high-stochasticity decoding parameters (top-$p = 0.9$, t $= 1.0$), responses from the same model remain highly repetitive, as shown in Figure 4: in 79% of cases, the average similarity exceeds 0.8. Since higher temperatures tend to produce incoherent text, these results show that even under maximally aggressive sampling, LMs still fail to generate diverse responses to open-ended queries.

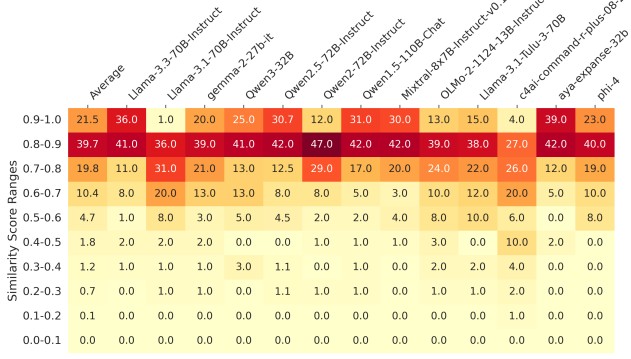

Figure 5: The heatmap shows degree of repetition in responses to open-ended queries generated by the **same LMs**. Using min-$p$ sampling with parameters (top-$p = 1.0$, min-$p = 0.1$, temperature $= 2.0$), the average pairwise similarity among responses to the same prompt typically exceeds 0.8.

---

[1]Sentence embeddings from OpenAI's `text-embedding-3-small` API are used.

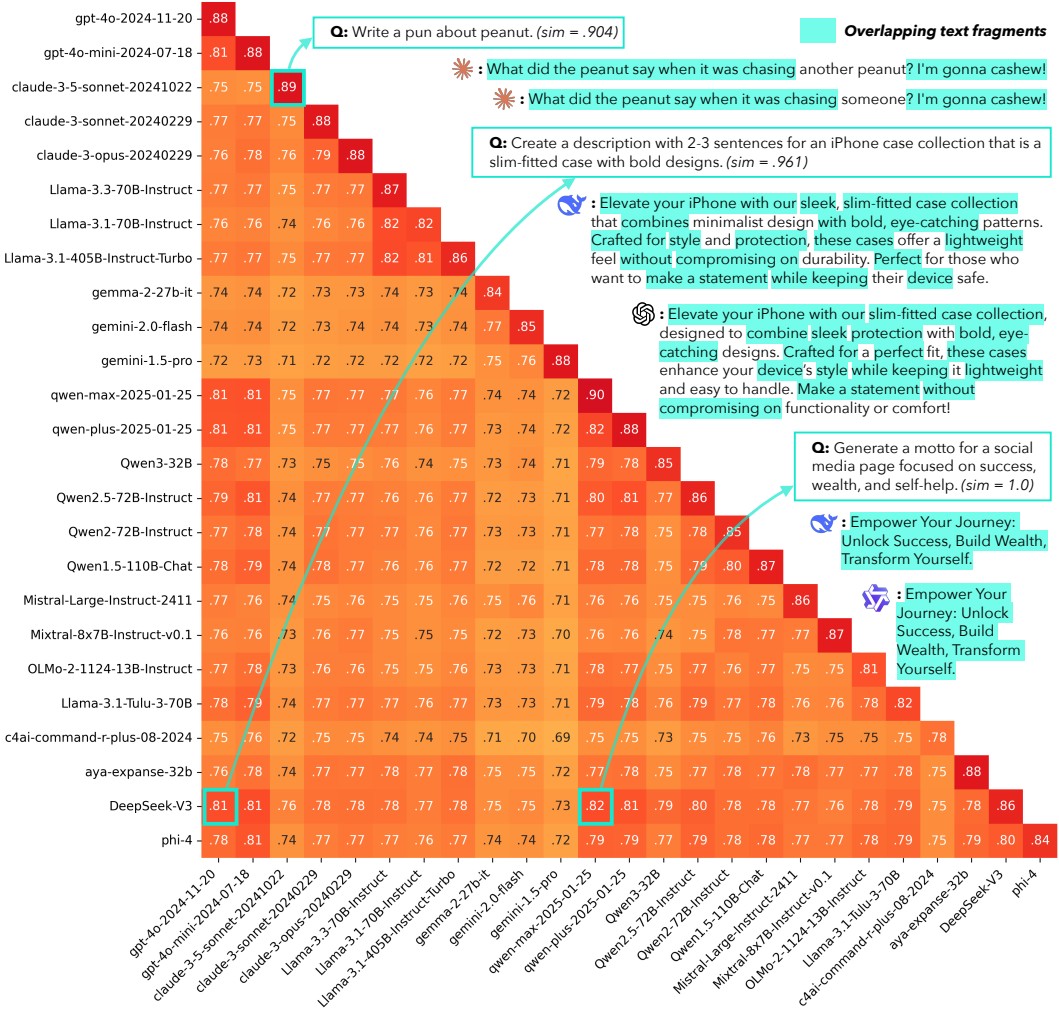

Figure 6: Average pairwise sentence embedding similarities between responses from different models reveal substantial semantic overlap across model outputs. Qualitative examples further illustrate that **different models often produce strikingly similar responses to fully open-ended queries**, including extended verbatim spans, underscoring the extent of repetition across models in open-ended generation tasks. All responses are generated using top-$p = 0.9$ and temperature $= 1.0$.

Recent work introduces min-$p$ decoding [60], a dynamic strategy for enhancing generation diversity that adjusts the sampling threshold based on model confidence. We evaluate min-$p$ decoding with the same setup and compute pairwise sentence embedding similarities. As shown in Figure 5, while min-$p$ reduces extreme repetition (fewer pairs above 0.9), $81\%$ of response pairs still exceed 0.7 similarity and $61.2\%$ exceed 0.8, revealing mode collapse even under diversity-oriented decoding.

Despite its promise, min-$p$ is not widely adopted, as it is better suited for creative tasks and less effective for close-ended ones [60]. Further, addressing LM repetitiveness through decoding alone places the burden on users to choose the right strategies. Thus, more generalizable solutions are needed at the model training level to robustly preserve output diversity without requiring user intervention. For the complete breakdown of results of all models, see §Appendix C.2.

**Inter-model homogeneity.** Not only do individual models repeatedly generate similar content, but different model sizes and families also produce highly repetitive outputs, sometimes sharing substantial phrase overlaps. As shown in Figure 6, the average pairwise similarity between responses from different models ranges from $71\%$ to $82\%$, with some pairs notably higher. For example, `DeepSeek-V3` and `qwen-max-2025-01-25` share a similarity of $0.82$, while `DeepSeek-V3` and `gpt-4o-2024-11-20` reach $0.81$. Interestingly, OpenAI's GPT models and Qwen's API models tend to have higher similarities even with models outside their own families. Although the exact causes remain unclear due to proprietary training details, possible explanations include shared data pipelines across regions or contamination from synthetic data. We highlight the need for future work to rigorously investigate the sources of such cross-model repetition.

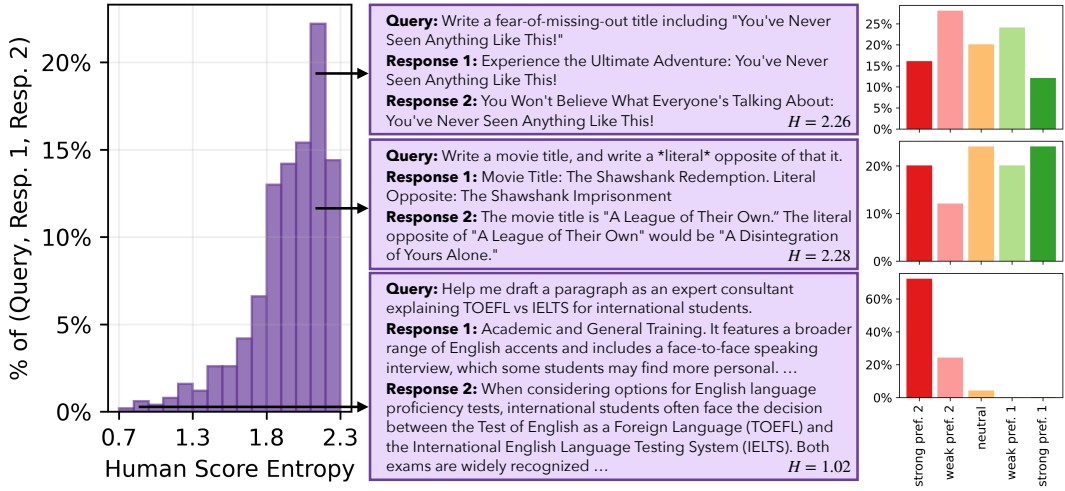

Figure 7: The **left** histogram shows the distribution of Shannon entropy across the 25 human annotations for each **(Query, Response 1, Response 2)** triplet, where annotators judge which response is better. Given open-ended queries, multiple high-quality responses are possible, often leading to disagreement among annotators and, on average, high entropy. With 25 annotations, label distributions can vary widely across triplets. The **middle** panel presents example triplets from different entropy regions, and the **right** bar plots show their corresponding label distributions.

Beyond general trends, we further analyze how repetition emerges at the instance level. As in prior work [52, 59], we observe verbatim phrase overlaps within responses from the same model. Surprisingly, such overlaps are also prevalent across different models, even for fully open-ended queries with large output spaces. For example, Figure 6 shows that DeepSeek-V3 and gpt-4o-2024-11-20 generate overlapping phrases like "Elevate your iPhone with our," "sleek, without compromising," and "with bold, eye-catching" in answer to the query "Create a description with 2-3 sentences for an iPhone case collection that is a slim-fitted case with bold designs." In some cases, models output identical responses: for "Generate a motto for a social media page focused on successes, wealth, and self-help," both qwen-max-2025-01-25 and qwen-plus-2025-01-25 generate "Empower Your Journey: Unlock Success, Build Wealth, Transform Yourself." These instance-level verbatim overlaps illustrate the severity of the "Artificial Hivemind" effect across models. Paraphrases of the same open-ended queries also lead to verbatim overlaps, as illustrated in Tables 17–18 in § Appendix C.4.

Beyond surface-level overlap, repetition also manifests semantically: models convey the same core ideas using different phrasing. As shown in Figure 1 (more in Figure 15-18), for the query "Write a metaphor about time," 50 responses from each of 25 models form just two clusters: a dominant one centered on "time is a river" and a secondary one on "time is a weaver." This convergence of abstract concepts reveals the depth of the "Artificial Hivemind" exposed in more subtle forms.

To quantify response uniformity across models, *we examine the extent to which outputs from different models become indistinguishable from one another.* Given 25 unique models, each generating 50 outputs to queries from INFINITY-CHAT, we ideally expect greater diversity across different models than from within a single model. To measure this, we identify the top $N$ most similar outputs for each query and count the unique models contributing to that set. A higher count suggests stronger cross-model similarity. As shown in Figure 8, the most similar responses often originate from multiple models. For instance, with $N = 50$, perfectly disjoint responses would yield all 50 from a single model. Yet, we find an average of ∼8 unique models per top-50 cluster, with some queries exceeding 10, indicating distinct models frequently generate highly similar content, sometimes resulting in higher *inter-* than *intra-model* similarity. See full inter-model homogeneity results in §Appendix C.3.

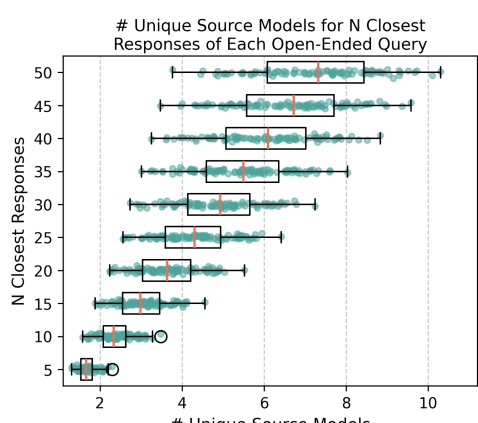

Figure 8: The avg.# of unique source models among the top-$N$ most similar responses to each open-ended query across 25 LMs.

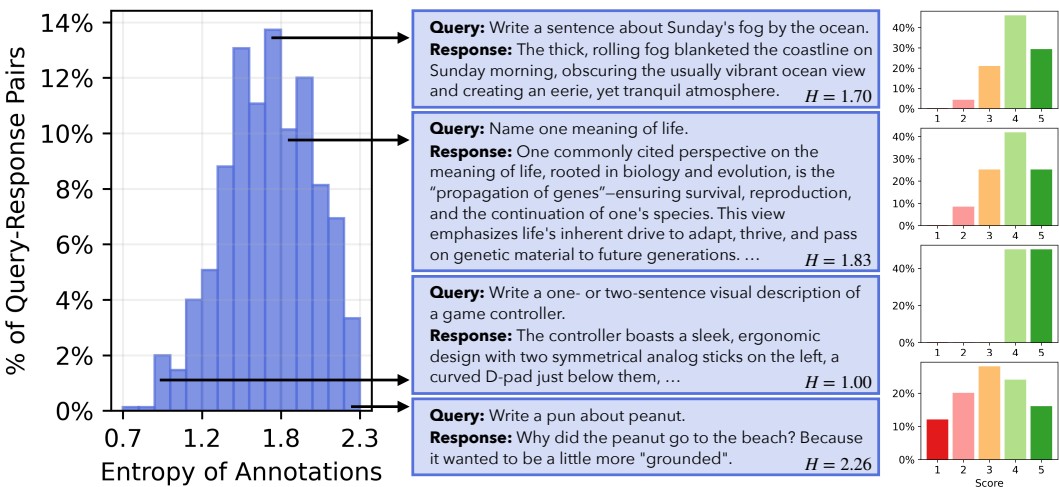

Figure 9: The **left** histogram shows the distribution of Shannon Entropy computed over the 25 human ratings for each **(Query, Response)** pair. Given the open-ended nature of the queries, multiple responses can be valid, leading to diverse preferences across annotators. As a result, the annotation label distributions vary significantly across examples. The **middle** panel presents representative (Query, Response) pairs from different entropy regions, and the **right** bar plots display their corresponding label distributions.

To summarize, our work provides further evidence of high syntactic repetition across different models. While a full causal analysis is beyond the scope of this study, our findings motivate future research to investigate whether such repetition arises from pretraining data, alignment processes, memorization, contamination, or generalization.

# 4 How Do LMs, Reward Models, and LM Judges Handle Alternative Responses to Open-Ended Queries?

Having established the generative homogeneity of LMs, in this section, we examine whether the *ratings* of *LMs*, *reward models*, and *LM judges* are calibrated to match human scores given different responses to open-ended queries from INFINITY-CHAT.

## 4.1 Gathering Distributional Annotations Across *Many* Humans

Humans may have divergent preferences over similar-quality alternative responses to open-ended queries. To study how models handle such diversity, we need densely annotated data that captures distributional human preferences. Existing alignment datasets, like HelpSteer3 [86], typically contain only sparse labels (e.g., 3 annotators per item). To address this, we collect both *absolute ratings* (1–5 scale for response quality) and *pairwise preference ratings* (strong/weak preference between two responses to the same query), each with extensive annotations. For absolute ratings, we sample 15 responses for each of 50 prompts from INFINITY-CHAT100 and collect 25 ratings per (Query, Response), yielding $25 \times 15 \times 50 = 18{,}750$ labels. For pairwise preference rating, we sample 10 response pairs per prompt and gather 25 annotations per (Query, Response 1, Response 2), totaling $25 \times 10 \times 50 = 12{,}500$ labels. This is the first large-scale human-annotated dataset with dense human ratings on alternative responses to the same open-ended queries, providing both absolute and pairwise preference labels for fine-grained analysis of human idiosyncratic and collective preference distributions. Full details of the human annotation process are provided in §Appendix D.1.

Figure 7 shows the distribution of Shannon entropy over human preference annotations for (Query, Response 1, Response 2) triplets. Annotators often disagree on which response is better, resulting in entropy skewed toward the higher end. As shown by the bar charts on the right, label distributions vary widely across examples: some response pairs show near-uniform support across all options, indicating substantial annotator disagreement for alternative responses of open-ended queries. Figure 9 shows a similar trend in the entropy of human annotations for absolute ratings of (Query, Response) pairs.

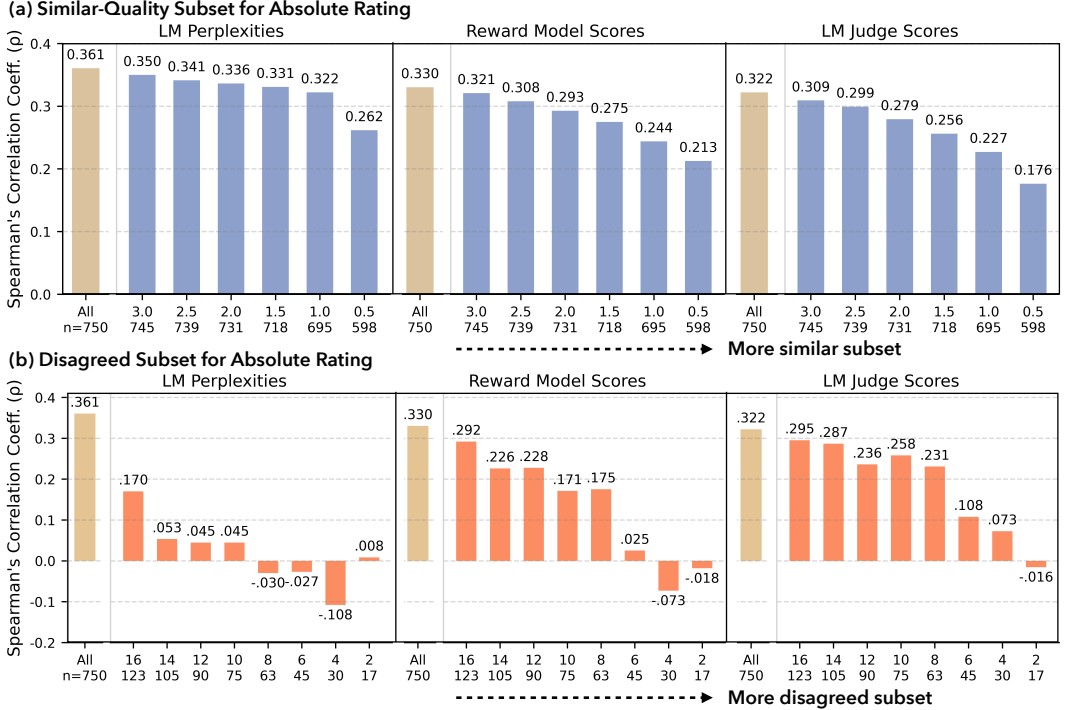

Figure 10: We compute Spearman's correlation coefficients between human-annotated and model-generated **absolute rating** scores, including *LM* perplexity scores, *reward model* scalar outputs, and *LM judge* scalar ratings. Correlations are calculated for the **full set**, as well as two groups of subsets: (a) responses with **similar human-rated quality**, and (b) responses with **high human disagreement**. The results show that correlations are notably lower in these two subsets, indicating weaker alignment between model scores and human judgments in cases of subtle or contested quality differences.

## 4.2 Gathering LMs, Reward Models, and LM Judges Ratings

We aim to assess how LMs, reward models, and LM judges align with human ratings when evaluating alternative responses to open-ended queries. Specifically, we compare 3 types of model-generated ratings against human annotations. **LM** scores are derived from response perplexity given the query. **Reward model** scores are based on standardized scalar reward outputs. **LM judge** ratings follow standard prompting protocols using two rubrics: an overall quality score and the HHH rubric (Helpfulness, Harmlessness, Honesty) [9]. See §Appendix D.2 for the full list of 56 state-of-the-art LMs, 6 top-ranked reward models (per RewardBench [49]), 4 LM judges (including GPT-4o and Prometheus [40] variants), and details on the rating procedures.

## 4.3 Comparing Model Ratings to Human Scores for Responses to Open-Ended Queries

We examine how model ratings align with human judgments on (1) *similar-quality alternative responses* to the same open-ended queries and (2) responses with *high annotator disagreement*.

**Motivation for comparing model scores to average human ratings.** Our motivation stems from how reward models (or LM judges) are used in training to evaluate responses to open-ended queries without a single ground truth. Different annotators may prefer different answers, yet their average ratings are often similar, implying multiple responses can be equally high-quality. Current reward models, however, fail to capture this equivalence, assigning diverging scores and causing downstream models to overvalue one response despite comparable human approval. To address this, we collect 25 human ratings per example to capture diverse preferences, using the average score to reflect shared human judgment. We then test whether LMs, reward models, and LM judges correlate less reliably with responses that humans broadly consider comparably good, hence our choice to compute human correlation using average human ratings.

**Models show weaker alignment with human ratings for alternative responses of similar quality.** We hypothesize that models are less aligned with human judgments on similar-quality examples, as models are typically trained with more clearly differentiated responses.

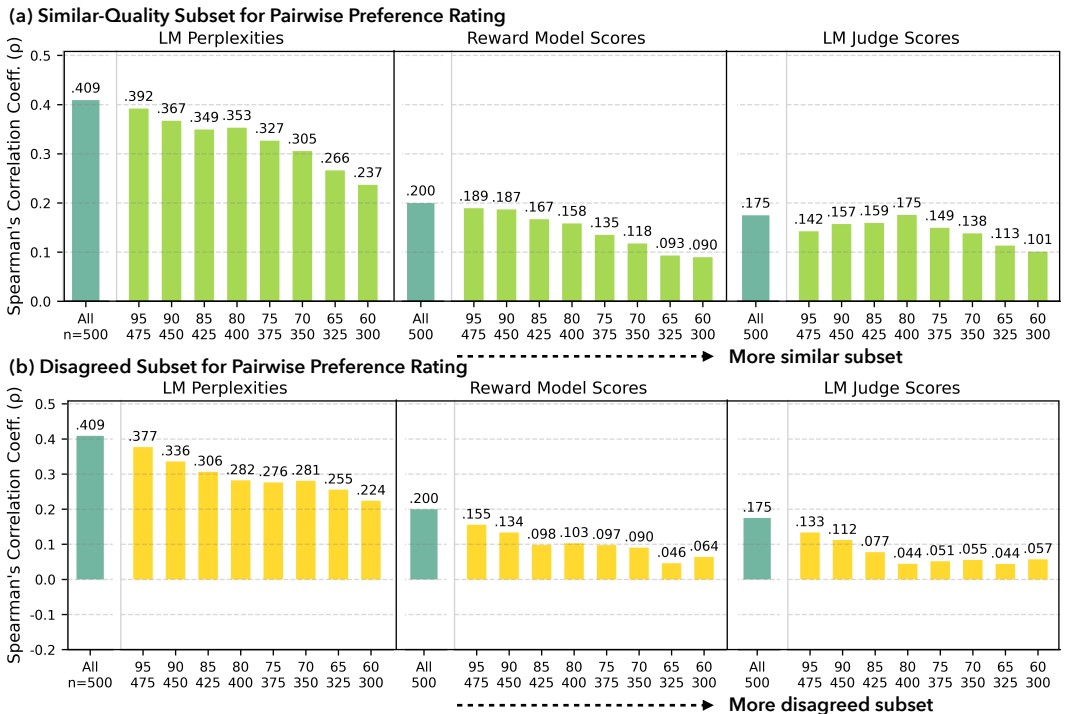

Figure 11: We compute Spearman's correlation coefficients between human-annotated and model-generated **pairwise preference rating** scores, comparing the **full set** to two groups of subsets: (a) responses with **similar human-rated quality**, and (b) responses with **high human disagreement**.

For the *absolute rating* setup, we identify similar-quality (Query, Response) pairs by filtering out outliers using Tukey's fences [20]. This method defines outliers as points beyond $Q_1 - k \cdot \text{IQR}$ or $Q_3 + k \cdot \text{IQR}$, where $Q_1$ and $Q_3$ are the 25th and 75th percentiles and $\text{IQR} = Q_3 - Q_1$. We vary the constant $k$ from 0.5 (aggressive filtering) to 3.0 (conservative filtering) in increments of 0.5 to generate subsets of increasing similarity. We then compute Pearson correlations between model and human absolute ratings on the full set and these filtered subsets, as shown in Figure 10 (a). For the *pairwise preference rating* setup, we identify similar-quality (Query, Response 1, Response 2) triplets by ranking examples by how many annotators rate the two responses as similar in quality. We select the top 60%–95% most similar examples to form subsets, with the 60% subset containing examples of higher similarity. We then compute Pearson correlations between the score differences of model ratings and those of human ratings across the full set and each subset, as shown in Figure 11 (a).

*Our results show that correlations between human ratings and those of LMs, reward models, and LM judges drop significantly on similar-quality subsets, for both absolute and pairwise preference rating setups.* Since there is no single gold-standard approach for selecting subsets of responses with similar quality given our data structure, we additionally report results using alternative subset selection methods in Table 19 (§ Appendix D.3). Our findings remain consistent across methods, highlighting the need for better modeling of fine-grained distinctions among equally high-quality responses to open-ended queries. For full results, including alternative grouping methods and model-level breakdowns, see § Appendix D.3.

**Model judgments are less aligned where annotators disagree.** We hypothesize that model ratings are less calibrated to human judgments on examples with high annotator disagreement, as models are primarily trained on examples with higher human agreement.

For the *absolute rating* setup, we identify disagreement by ranking (Query, Response) pairs by Shannon entropy across 25 human labels. We then select the top $2, 4, 6, 8, 10, 12, 14, 16\%$ highest-entropy examples as disagreed subsets. Figure 10 (b) shows the Pearson correlations between model and human ratings across the full set and these subsets. For the *pairwise preference rating* setup, we quantify disagreement for each (Query, Response 1, Response 2) triplet using percentage disagreement: $P_{\text{disagree}} = 1 - \frac{\max(C_{\text{prefer 1}}, C_{\text{prefer 2}}) + 0.5 \cdot C_{\text{tie}}}{C_{\text{total}}}$, where $C$ denotes the count of annotations per preference type. We retain the top 60% to 95% most disagreed examples, with the 60% subset

representing stronger disagreement. Pearson correlations between model and human score differences across the full set and each subset are shown in Figure 11 (b).

*Our results show that correlations with human ratings across models drop substantially for examples with high annotator disagreement, in both absolute and pairwise rating setups.* We also report results using alternative subset selection methods in Table 20 (§ Appendix D.4). The findings remain consistent across methods, highlighting the need for more nuanced modeling of idiosyncratic human disagreement to better capture the broad spectrum of open-ended possibilities. For complete results, including alternative grouping methods and model-level breakdowns, see § Appendix D.4.

## 5 Related Work

**The diversity collapse problem of LMs.** Diversity collapse, characterized by the inability of LMs to generate diverse outputs, presents a significant challenge to pluralistic alignment research [93, 75, 23, 90, 52, 88]. Prior studies identify some key factors contributing to diversity collapse, including training on synthetic data [32, 83, 90, 76], LM alignment [56, 41, 43], and insufficient diversity in training data [15]. Potential consequences of diversity collapse include reduced creativity, loss of minority perspectives, spread of bias, and overall decline in model utility and trustworthiness [5, 38, 21]. In response, a range of mitigation strategies are proposed, such as training corpora diversification [84, 65, 85, 33], training algorithm modifications [92, 53], alternative decoding [82, 60] and prompting [91, 11] strategies.

**Measuring the creativity and divergent thinking of language models.** Recent efforts to measure the creativity and divergent thinking of LMs often adapt established psychometric tests. For example, [16] utilizes a divergent association task (DAT) by asking models to generate semantically distant or unrelated words [61]. Similarly, tasks such as the Alternate Uses Test (AUT) [66], the Torrance Tests of Creative Thinking (TTCT) [1, 13], Human Evaluation, and LLM-as-a-judge [45] are employed to assess dimensions like fluency, originality, complexity, and effective semantic diversity [77] of LM responses. Despite these demonstrated capabilities, LLM-generated creative content tends towards homogeneity, even when individual outputs achieve high creativity scores [87]. To address these evaluation complexities, some benchmarks, such as [69, 54, 64, 92], focus on specific creative abilities like scientific idea and code generation. Other works propose new metrics [59]. While LMs show promise in creative tasks, comprehensively evaluating their creativity remains an active and challenging research area [34]. We conduct a large-scale systematic study of real-world open-ended user queries and provide a comprehensive taxonomy, query dataset, and dense human annotations to improve evaluation and model training for reducing mode collapse in language models.

**Disagreement and pluralistic alignment of language models.** Advances in AI value alignment research have substantially improved LM utility and safety, through enhanced training processes [72, 62, 67], and the use of both synthetic data [30] and human datasets [8, 29]. Yet, a significant challenge remains: the potential for monolithic value representation [70, 71]. In contrast, the emerging focus on pluralistic alignment emphasizes the need for AI to serve the varied demands of a wide population [78, 2]. This shift is driving innovation in methods [19, 60, 48, 14, 26, 80], benchmarks [12], and data collection strategies [42, 73] to support this vision of diversity. Additionally, approaches leveraging multiple LMs interacting through system messages are also being explored to boost variety [81, 17, 57]. Parallel efforts are dedicated to quantifying and improving the cultural diversity exhibited by LLMs [68, 74, 18, 58, 51]. Yet, a common characteristic of many existing pluralistic alignment work is its reliance on predefined diversity dimensions, like demographics [47, 55], personality style [50, 35, 95, 63], and cultural background [89, 22, 4]. To enable models that genuinely cater to individuality without relying on stereotypes, individual-level alignment is needed [36, 95].

## 6 Conclusion

Our work introduces INFINITY-CHAT, a large-scale resource designed to evaluate LMs' diversity in naturally occurring, open-ended settings. Through comprehensive analysis, we uncover the "Artificial Hivemind" effect, highlighting both intra-model repetition and inter-model homogeneity of current LMs. By coupling a diverse taxonomy of prompts with dense human preference annotations, INFINITY-CHAT provides a new foundation for diagnosing, benchmarking, and ultimately mitigating mode collapse in generative AI. We hope this resource catalyzes future efforts to foster genuine diversity in model outputs and guard against the homogenization of human expression.

## Acknowledgment

We thank Alon Albalak for his valuable contributions to the extended version of this work. This work was supported by DARPA under the ITM program (FA8650-23-C-7316). This work was also supported in part by the Defense Advanced Research Projects Agency's (DARPA) SciFy program (Agreement No. HR00112520300). The views expressed are those of the author and do not reflect the official policy or position of the Department of Defense or the U.S. Government.

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

# Appendices

# A Discussions

## A.1 Limitations

While comprehensive with 26K queries, INFINITY-CHAT represents only a snapshot of the vast space of possible open-ended queries and may not capture all forms of creative divergence across different contexts. Moreover, the focus on English-language prompts derived from WildChat potentially underrepresents linguistic, cultural, and regional diversity in user interactions with language models, limiting the generalizability of our findings to non-English contexts and diverse cultural perspectives. To construct our taxonomy of queries, we used GPT-4o to efficiently label user queries, and we found human annotators achieved 74.7% agreement with these automatic labels. While this accuracy could be improved, e.g., by labeling with stronger models or through model ensembling approaches, this performance is comparable to typical human annotation accuracy and we deem sufficient to represent the distribution of categories within our open-ended taxonomy. Finally, while our analysis successfully reveals clear patterns such as the "time is a river/weaver" dichotomy, it may oversimplify the multidimensional nature of creative expression, and relying on semantic similarity of text embeddings to quantify diversity may lack sufficient expressiveness to capture the full spectrum of creative variation in generated responses.

While extending INFINITY-CHAT to multilingual and multicultural settings is an important direction, we anticipate that similar homogenization issues will likely arise across languages and cultures, given the global overlap in pretraining data sources and alignment practices. We encourage future work to pursue such extensions, and we hope our benchmark provides a comprehensive foundation that multilingual research can build upon. Our taxonomy was designed to be language-agnostic and can support adaptation when appropriate resources and expertise are available. We'll further strengthen this discussion in camera-ready.

Diversity and quality represent two key dimensions in evaluating responses to open-ended queries. In this work, we focused primarily on diversity, with fixed model decoding configurations, without studying quality. We selected a decoding configuration that generated empirically coherent text and investigated whether models can be guided to produce responses that maximize diversity while maintaining reasonable quality. Furthermore, we show that existing LMs, LM judges, and reward models struggle to reliably distinguish between equal-quality responses for open-ended queries. This motivates the need for further investigation into reliable automatic evaluation of equal-quality responses for open-ended queries, beyond human annotation. Finally, although our analysis identifies clear evidence of an "Artificial Hivemind" effect across models, it falls short of establishing the underlying mechanisms causing this homogenization. Future research can mechanistically disentangle causality, whether stemming from shared training data, memorization, generalization, or other factors.

Our work has significant implications for AI development and society, providing a valuable dataset for diagnosing and potentially mitigating LM homogenization and accelerating the development of more diverse, creative AI systems that align with diverse human needs and perspectives. However, this "Artificial Hivemind" effect also raises serious concerns about models' long-term impact on human creativity. If users increasingly rely on such systems for creative tasks, exposure to homogenized outputs could subtly influence human thinking patterns and reduce overall cultural and intellectual diversity. Furthermore, it could exacerbate existing biases and limit representation of marginalized perspectives. When LMs converge on dominant cultural expressions—such as Western-centric metaphors like "time is a river"—they may inadvertently suppress alternative worldviews and traditions.

The "Artificial Hivemind" effect also highlights fundamental questions about what values we want AI systems to embody. Should AI prioritize efficiency and consistency, or diversity and novelty? These choices reflect deeper societal values about creativity, culture, and human flourishing. The ethical implications of LM homogenization extend far beyond technical considerations, touching on fundamental questions about human agency, cultural preservation, and the kind of future we want to create with these technologies. Addressing these challenges requires not just technical innovation, but sustained ethical reflection and inclusive dialogue about the role of AI in human creative expression.

## A.2 Future Directions

Our findings open several promising directions for advancing the study and mitigation of behavioral convergence in large language models.

**Foundation and training analysis.** We plan to extend the Artificial Hivemind testbed to foundation models without instruction-following capabilities to better disentangle the respective roles of pre-training and post-training in shaping convergent behaviors. Moreover, we aim to quantify the relative contributions of different post-training pipelines—such as supervised fine-tuning, RLHF/RLAIF, and constitutional training—to the emergence of homogenized responses.

**Mitigation and alignment strategies.** Future work will explore diversity-aware training objectives and alignment schemes that explicitly reward exploration of multiple valid modes while preserving response quality. We also intend to benchmark decoding strategies (e.g., diverse beam search, nucleus sampling variants) under the Artificial Hivemind metric to evaluate their effectiveness in counteracting homogenization.

**Practical integration.** To make our framework actionable, we will integrate Artificial Hivemind into red-teaming workflows to stress-test model robustness and coverage under open-ended queries. Additionally, our dataset can serve as a training prompt resource for reinforcement learning methods that explicitly encourage diversity during alignment. Finally, Hivemind's diagnostic signals can inform curriculum design—gradually exposing models to increasingly open-ended prompts that are most susceptible to mode collapse.

Together, these directions position Artificial Hivemind as not only a descriptive diagnostic tool, but also a foundation for developing training, decoding, and evaluation practices that foster greater diversity and individuality in language models.

### A.3 Broader Implications

**Societal Implications.** While some degree of convergence among large language models (LLMs) is statistically expected, its societal consequences warrant close scrutiny. As billions of users increasingly depend on LLMs for creative, educational, and decision-making purposes, understanding and quantifying behavioral homogenization becomes critical. Emerging evidence shows measurable shifts in human writing styles, creative ideation, and divergent thinking following the widespread adoption of systems like ChatGPT [46, 6, 7, 24, 44, 79]. These findings suggest that model-level convergence may propagate into human expression, amplifying uniformity in linguistic and cognitive patterns at scale. By introducing quantitative tools to measure and track such convergence, our work provides an empirical foundation for assessing the long-term cultural and epistemic consequences of LLM-mediated communication.

**Implications for Data Distillation and Model Training.** Our results also have immediate implications for synthetic data generation and model training practices. While distilling knowledge from LLMs has proven transformative for efficiency and scalability, it is well known that relying on a single model as a teacher can intensify mode collapse, reinforcing narrow response patterns, diminishing output diversity, and, in extreme cases, leading to degenerative feedback loops [3, 39]. To counter this, recent approaches employ model swarms and multi-agent frameworks that aggregate outputs from multiple models in pursuit of diversity [28, 37].

However, our findings reveal a fundamental limitation: even across distinct state-of-the-art models, diversity in open-ended tasks is far from guaranteed. Models often converge toward highly similar answers, undermining the assumed benefits of multi-model distillation. This insight carries significant implications for research on synthetic data curation, ensemble-based alignment, and constitutional AI. Without rigorous diagnostic frameworks such as Artificial Hivemind, practitioners risk overestimating cross-model diversity and unintentionally perpetuating homogenization across generations of language models.

# B  INFINITY-CHAT: A Dataset of In-The-Wild Open-Ended User Queries

In this section, we describe the process of mining open-ended and closed-ended queries used to construct INFINITY-CHAT.

## B.1  Details of the Mining of In-The-Wild Open-Ended User Queries

To understand the types of real-world open-ended queries posed to LMs across diverse use cases, we carefully curate and select such queries from WILDCHAT, a dataset containing a rich variety of real-world conversations from in-the-wild users [94].

To narrow the scope of queries for closer analysis, we first extract user queries from `allenai/WildChat-1M` that meet the following criteria: (1) written in English; (2) labeled as non-toxic and non-harmful according to the built-in flags in the original WILDCHAT dataset; (3) directed to GPT-4 models; and (4) of moderate length (i.e.,, between 15 and 200 characters). This filtering process yields a total of 37,426 query candidates.

We then classify the remaining queries using `gpt-4o-2024-11-20` along three dimensions: (1) **Meaningful Information**: whether the query poses a meaningful question or seeks substantive information; (2) **Greeting/Model Inquiry**: whether the query is a greeting or an inquiry about the model itself (e.g.,, "Are you an AI?"); and (3) **Response Type (*Single* or *Multiple*)**: whether the query is expected to yield a single specific answer or multiple valid, diverse responses. During this classification process, we also revise queries in real-time if they are unclear or ambiguous.

After applying automatic filtering, we obtain 26,070 open-ended queries and 8,817 closed-ended queries (e.g.,, yes/no questions, queries with precise answers, or well-scoped writing tasks). Examples of open-ended and closed-ended queries are shown in Table 3 and Table 4, respectively. We refer to this dataset as INFINITY-CHAT, which consists of open-ended queries designed to elicit diverse and high-quality responses. The prompts used to classify user queries as either open-ended or closed-ended are shown in Figures 12 and 13.

## B.2  Details of Defining the Open-Ended Query Taxonomy

Table 2 presents additional examples illustrating the open-endedness taxonomy (Figure 2 in the main paper) in greater detail. Figure 14 shows the prompt used to annotate both new and existing open-ended categories of user queries. The automatic taxonomy mining is performed using `gpt-4o-2024-11-20`.

## B.3  Human Validation of the Open-endedness of Queries in INFINITY-CHAT

To strengthen our dataset's validity, we conduct a human study to verify the open-endedness of queries in INFINITY-CHAT. We sample 100 evaluation examples from the dataset and recruit 86 participants on Prolific, assigning three participants to each query. All participants meet the following qualifications: English as a primary, first, or fluent language; an approval rate above 99%; 500–10,000 prior submissions; and an education level beyond high school. Each participant is instructed to answer two questions for each query:

**Q1**: **Is the user query open-ended? In other words, can it reasonably allow for several different, similarly valid answers, rather than just a single, fairly deterministic one?**

- A: Yes — this query is open-ended and can have multiple possible answers
- B: No — this query is NOT open-ended and has a single, fairly deterministic answer

We used majority vote among three annotators per question to assess query open-endedness. By this criterion, 89% of queries were judged open-ended, providing strong evidence of high open-endedness (per Q1). With a more inclusive criterion, counting a query as open-ended if at least one annotator agreed, 100% of queries qualified as open-ended.

**Q2**: **If the query is open-ended, how many different, reasonable alternative answers do you think could apply?**

- A: Fewer than 3

- B: 3 to 10
- C: 10 to 20
- D: More than 20

Beyond the binary classification of open-endedness, we assessed the breadth of reasonable answers. As shown in Table 1, annotators judged 81.27% of queries to allow 3+ alternatives, and 34.66% to permit over 20, showing that real-world queries are rarely narrow or deterministic. This underscores the need for diversity-aware LLM evaluation, as simplistic benchmarks with limited answer ranges fail to capture the true complexity of user queries.

Table 1: Human ratings of the degree of open-endedness of queries in INFINITY-CHAT.

| Answer | Percentage (%) |
|---|---|
| Fewer than 3 | 18.73 |
| 3 to 10 | 31.87 |
| 10 to 20 | 14.74 |
| More than 20 | 34.66 |

We further examined how annotators perceived the degree of open-endedness across prompts. Interestingly, their judgments varied considerably across individuals. Below are sample annotations from three annotators per prompt:

- [More than 20, 10 to 20, 3 to 10]
- [More than 20, 3 to 10, More than 20]
- [More than 20, More than 20, More than 20]
- [Fewer than 3, More than 20, More than 20]
- [Fewer than 3, 10 to 20, More than 20]

This variability indicates that assigning a definitive level of open-endedness to each query is inherently challenging. Nevertheless, we conducted a coarse-grained grouping based on the following rule: (1) *High open-endedness*, if *any* annotator labeled a prompt as "More than 20 responses." (2) *Low open-endedness* otherwise.

Using this classification, we analyzed model response diversity based on sentence similarity across 42 models. The average similarity scores were: High open-endedness = 0.800, Low open-endedness = 0.837. These results suggest that, even for prompts perceived as more open-ended, models do not necessarily produce more diverse responses.

## Prompt for Filtering Open-Ended User Queries (Part 1)

You are an expert in analyzing user queries to classify their intent, assess their clarity, and improve unclear language while maintaining the original meaning.

Your task is to:

- Determine whether the user query is a *meaningful question* or *seeks meaningful information*. If it is not meaningful or gibberish, classify it as such with reasoning.
- *Only if the query is meaningful and not a greeting or model inquiry*, determine whether the user query is a *greeting* or is *inquiring about the model itself*. Classify such queries appropriately with reasoning. *Skip this classification for non-meaningful queries.*
- *Only if the query is meaningful and not a greeting or model inquiry*, assess whether the query might result in *diverse, equally valid alternative responses* or has a *single optimal response*. *Skip this classification for non-meaningful or greeting queries.*
- Revise the query *minimally* to correct any grammatical, spelling, or language errors while retaining the original meaning and intent.
- Return your classifications, reasoning, and any necessary revisions in JSON format.

Classify each query into one or more of the following categories:

- Meaningful Information: The query is a meaningful question or seeks meaningful information. - Greeting/Model Inquiry: The query is a greeting or asks about the model itself. - Response Type (*Single* or *Multiple*): Only classify the query if it is meaningful and not a greeting. Determine if the query expects one specific answer or could result in multiple valid, diverse responses.

For each query, provide:

- The original query.
- The revised query (if applicable).
- Brief reasoning and classifications for each task in the JSON format.

Example 1:

Query: Wht is sqr root of 64?

Output:

```
{
  "original_query": "Wht is sqr root of 64?",
  "revised_query": "What is the square root of 64?",
  "meaningful_information": {
    "reasoning": "This query seeks factual information, which is clear
    and specific.",
    "classification": true
  },
  "greeting_model_inquiry": {
    "reasoning": "The query is not a greeting or asking about the model.",
    "classification": false
  },
  "response_type": {
    "reasoning": "This query has a single factual answer, which is
    precise ('8').",
    "classification": "Single"
  }
}
```

Figure 12: Prompt for Filtering Open-Ended User Queries (Part 1)

## Prompt for Filtering Open-Ended User Queries (Part 2)

Example 2: Query: Hi, how are you?
Output:

```
{
  "original_query": "Hi, how are you?",
  "revised_query": null,
  "meaningful_information": {
    "reasoning": "The query does not seek meaningful information.",
    "classification": false
  },
  "greeting_model_inquiry": {
    "reasoning": "The query is a greeting intended for the model.",
    "classification": true
  },
  "response_type": null
}
```

Example 3:
Query: Tell me ideas on making work commun better.
Output:

```
{
  "original_query": "Tell me ideas on making work commun better.",
  "revised_query": "Tell me ideas on making work communication better.",
  "meaningful_information": {
    "reasoning": "The query seeks meaningful information by requesting ideas
    for improvement.",
    "classification": true
  },
  "greeting_model_inquiry": {
    "reasoning": "The query is not a greeting or asking about the model.",
    "classification": false
  },
  "response_type": {
    "reasoning": "This query invites multiple perspectives and strategies,
    each potentially valid and differing in content.",
    "classification": "Multiple"
  }
}
```

Example 4:
Query: asdjklqwe?
Output:

```
{
  "original_query": "asdjklqwe?",
  "revised_query": null,
  "meaningful_information": {
    "reasoning": "The query does not contain any discernible meaning or intent
    and cannot be interpreted as a valid question or request for information.",
    "classification": false
  },
  "greeting_model_inquiry": null,
  "response_type": null
}
```

Now classify this query and return the result in JSON format:
Query: {USER_QUERY}
Output:

Figure 13: Prompt for Filtering Open-Ended User Queries (Part 2)

You are an expert in analyzing user queries to determine their open-ended categories.

You will be provided with an open-ended user query directed at a language model. Such queries often allow for multiple valid responses. Your task is to classify the query into one or more relevant categories based on the predefined taxonomy below. If the query does not fit any existing category, you should create a new category as needed.

For each assigned category (whether predefined or newly created), provide a brief justification explaining why the query falls under that category.

Predefined Categories:
- Creative Content Generation
- Ideation and Brainstorming
- Philosophical Questions
- Abstract Conceptual Questions
- Ambiguous Everyday Questions
- Analytical and Interpretive Questions
- Speculative and Hypothetical Scenarios
- Value-Laden Questions with Alternative Perspectives
- Opinion-Based Questions with Alternative Perspectives
- Controversial Questions with Alternative Perspectives
- Alternative Communication Styles
- Alternative Writing Genres
- Information-Seeking about Problem Solving
- Information-Seeking about Decision Support
- Information-Seeking about Skill Development
- Information-Seeking about Recommendations
- Information-Seeking about Concept Explanations
- Information-Seeking about Personal Advice

Response Format:

Your response should follow the structured format below. Ensure that all relevant categories are included.

```
{
    "query": The user query that you are classifying,
    "categories": [
        {
            "category": The category that you have classified the user
            query into,
            "type": predefined or new,
            "justification": A short justification for your classification
        },
        ...
    ]
}
```

Please classify the following query accordingly:

Query: This is about the Monster Hunter series. Write a TV advertisement for a Rathian, making subtle implications that she can be a companion and a wife.

Output:

Figure 14: Prompt for Classifying the Category of an Open-Ended User Query

Table 2: Taxonomy of open-ended queries that can benefit from model generations with diversity.

| Category | Sub Cat. | # | Representative Examples |
|---|---|---|---|
| Creative Content Generation | - | 58.0 | Compose a short poem about the feeling of watching a sunset. Write an 800-word essay on why 6 is afraid of 7. Generate a joke about electric vehicles. |
| Brainstorming & Ideation | - | 15.2 | I am a graduate student in Marxist theory, and I want to write a thesis on Gorz. Can you help me think of some new ideas? Suggest a feature for a smartwatch designed specifically for senior citizens. |
| Open -Endedness | Philosophical Questions | 3.5 | How do I understand what I want? Is it impossible to know everything? |
| | Abstract Conceptual Questions | 10.0 | Is honey magical? Is there life on other planets? In humans, is there a correlation between intelligence and compassion? |
| | Ambiguous Everyday Questions | 2.6 | I'm 60 years old and haven't achieved anything. What have I missed out on? What are signs that show you were not loved enough as a child by your mom? I have an apartment that costs about $100 a month. How do I survive? |
| | Analytical & Interpretive Questions | 22.6 | What's the consequence of algorithm-driven content for public discourse? How do global economic inequalities shape international relations? What is the future of work in an increasingly automated economy? |
| | Speculative & Hypothetical Scenarios | 22.2 | How might society evolve if telepathy became possible? Create a short review of a movie that doesn't exist yet. How will AI reshape the way humans interact with one another in 50 years? |
| Alternative Perspectives | Value-Laden Questions | 2.3 | Why is equal pay for equal work not worth imposing on an economy? Give me strong arguments as to why some people allow the death penalty. |
| | Controversial Questions | 2.5 | Different views on discipleship Is there such a thing as a paradox of free speech? |
| Alternative Styles | Communication Styles | 3.2 | Write an email to organize a catch-up with a referral partner. How to say a formal thank you for your time in the subject line of an email? |
| | Writing Genres | 38.5 | Can you write five tweets in the style of Dril about El Salvador? Write a play script in the style of Dilbert as if a man in an apocalyptic world. |
| Information -Seeking | Problem Solving | 19.3 | Give me a strategy to double the money in a month, starting with 1000 euros. How do I code an online forum using LAMP? |
| | Decision Support | 2.2 | Choose the right decision: buy a new Zara sneaker or a used Adidas sneaker? What is the best investment if I invest 60,000 euros? |
| | Skill Development | 23.5 | How can I make a zombie survivor game on Scratch? Help me use my Microsoft Surface touchpad and on-screen keyboard to run Dolphin Emulator for the Wii. |
| | Recommen -dations | 11.0 | What is a good secondhand market laptop for learning Python? Give me a 3-day plan for Osaka with a flight leaving on the 3rd day at 7pm. What is the best and most profitable day trading strategy of all time? |
| | Concept Explanations | 23.6 | What are the benefits of hardwood flooring over other types? Do you know the demon Morloch? |
| | Personal Advice | 4.1 | What should I learn in software development to be relevant in the future? My girlfriend is buying a house. I plan to marry her in a year or so. I don't know if it's fair for me to pay for the house or not. What's the best solution? |

Table 3: Example open-ended queries where diversity is important, collected from WildChat.

| Representative Open-Ended Queries from WildChat |
|---|

Write me 3 short tips for self-development.

Create a title with the prefix 'best', as a one-liner, using only strings, less than 100 characters.

Paraphrase this: We're checking if the domain 'aspris.ae' is included in our scope.

Name a hot English word below 10 letters.

Create a sentence using a minimum of 2 R-colored vowels.

Give an example of a linear graph in graph theory.

Give me a trivia question about blue birds and its corresponding answer.

Write me a 1-paragraph essay about the development of the economy during the Han Dynasty.

In three sentences, describe a girl wandering around in Vietnam.

Can you give an example of a life goal related to self-image?

Rave about the significance of rivers in a paragraph.

Write a sentence where the last word is 'apple'.

Write a May the 4th joke.

Write an essay on the importance of the Roman Empire and its impact on future generations. Max: 100 words.

Describe Apple Corporation in three sentences to a person who has no idea what cell phones are.

Name an economic value of an additional year of schooling.

Name one meaning of life.

Write an one-paragraph kid's story with a prince, a princess, and a dragon. When all hope is lost, the prince orders a magic sword from Amazon and slays the dragon. The other parts of the story are up to you.

Give me a tip to be more organized at work. I'm a high school teacher.

Explain computational irreducibility like I'm 5.

Make an analogy of the relationship between US and China.

Provide a few sentences on Sisu Cinema Robotics.

Give a numerical example to illustrate the concept of partial derivative.

Help me draft a paragraph as an expert consultant explaining TOEFL vs IELTS for international students.

Come up a short blurb to introduce a religion called The Next Exodus Society.

Write a headline for a company called "USBC CONSTITUTION" that encourages companies to donate their waste for recycling in exchange for money for the donated waste.

Write a Google ad with 2 sentences and a 30-character limit per sentence for mobile car detailing.

Give me a tip for managing a team of coworkers.

What is the difference between analysis and design? Can one begin to design without analysis? Why? Be concise.

Output a hard question to humanity (super concise and short), independent of theme.

Provide an example of the name of an optimization technique used in machine learning.

Briefly explain the potential uses of biofuels in 2-3 sentences.

Write a metaphor involving time.

If there were double the amount of oxygen in the air, what would happen? Write in 100 words.

Generate a paragraph on why introspection is very important for growth, and provide guidance on listening to yourself more than heeding other people's opinions.

Generate a one-liner title for 'Elephant' and 'sticker'.

Write a 30-word essay on global warming.

Generate a motto for a social media page focused on success, wealth, and self-help.

Can you give me an incredible STEM fair idea that is affordable and relate to issues in Vietnam?

Write a tweet about: This is a video from this morning's crazy sunrise at the beach.

Write a funny two-sentence birthday card message for a teammate who is 50 years old and loves going to a Toby Carvery restaurant.

Generate a description for 'Sticking to Cuteness: The Panda Way.'

Give me a short phrase to put on my portfolio webpage about being an amateur data analyst, data scientist, and Next.js web page developer.

Create a short 1-paragraph story about a boy running on a beach. He is Asian, 12 years old, and the time of day is 4 in the afternoon.

Can you give me a powerful rhetorical question for an essay about the harms of social media on teens?

Give me the names of 3 instrumental songs that best match the mood of a rainy night.

Give me the name of a instrumental song that matches the mood of a rainy night.

Describe Deadpool in a short paragraph. Make sure it's accessible to children.

Write a personal tweet about how I am walking right now at sunrise to the lighthouse; it's spring but cold like winter.

Write a fear-of-missing-out title including "You've Never Seen Anything Like This!"

Write a short FOMO title for a video of a shell on the beach

One other way to say: "Fingers crossed that everything goes well."

Write 3 to 4 lines about India.

Write 100-300 words on how stress affects the body and mind.

Table 4: Example closed-ended queries where diversity is NOT as important.

| Category | Representative Examples |
|---|---|
| Yes/No Questions | Is 'one's lineage' grammatically correct? |
| | Is a single cell visible under a microscope? |
| | Can humans have natural golden bronze skin? |
| | Can you access files on the internet directly? |
| | My friend from Russia said that in Russia there is no division into |
| | cities and towns, but only cities. Is that true? |
| Questions with Precise Answers | What is the plural form of the ancient Greek polis? |
| | When was marriage made into an institution in Europe? |
| | What is the ULA in space exploration? |
| | In a certain language, (A) 'hu ma sam' means 'Water is life'. |
| | (B) 'sam na zo' means 'Glass of water'. (C) 'chi zo ma' means 'life of PI'. |
| | Which of the following represents 'PI' in that language? |
| | How many atoms are contained in 6.71 grams of sulfur? |
| Well-Scoped Writing | Write an email to Kathy; thank her for her fast reply and also tell her that |
| | we received her invoice. Inform her that we are doing the paperwork for |
| | this invoice and will update her about any further developments. |

# C  Artificial Hiveminds: Examining the Intra- and Inter- Model Homogeneity

In this section, we present a detailed analysis of intra- and inter-model homogeneity across a broad range of open-source and closed-source language models.

## C.1  Evaluation Setups

For both intra- and inter-model analyses, we adopt a unified generation protocol and reuse the same model outputs across both settings. We use 100 open-ended prompts from INFINITY-CHAT100, a carefully selected subset of representative queries from INFINITY-CHAT, as the seed prompt set for generation. A comprehensive list of language models is curated for the analysis. The full list of models considered, as well as the subset presented in the main paper, is provided in Table 5. Due to space constraints, we select representative models for the main paper based on their scale or strength within each model family.

For all HuggingFace models, generations are performed on NVIDIA A100 or H100 GPUs, depending on availability. For closed-source models such as OpenAI, Anthropic, Gemini, and Qwen, we use their respective APIs to obtain responses. For a small number of the largest models (e.g., `deepseek-ai/DeepSeek-V3`) that exceed our local GPU capacity, we use TogetherAI to generate responses. Regardless of the generation method, all models follow the same decoding configurations. For the top-$p$ setup, we use $p = 0.9$, temperature $= 1.0$, and a maximum generation length of 2048 tokens. For the minimum-$p$ setup, we use $p = 1.0$, min-$p = 0.1$, temperature $= 2.0$, and the same maximum generation length. Under each decoding configuration, each model independently generates 50 responses for every open-ended prompt in INFINITY-CHAT100. All similarities are computed using cosine similarity between sentence embeddings of the responses, generated by OpenAI's embedding model `text-embedding-3-small`.

For concrete examples of response pairs and their corresponding similarity scores, see Tables 12–16.

## C.2  Intra-Model Homogeneity

Table 6 presents the extended results of intra-model repetition for all models listed in Table 5, serving as a detailed counterpart to the results summarized in Figure 6.

## C.3  Inter-Model Homogeneity

Table 7-11 present the extended results of inter-model homogeneity for all models listed in Table 5, serving as a detailed counterpart to the results summarized in Figure 6.

## C.4  Examining How Paraphrased Queries Influence Response Homogeneity

We conduct an experiment to examine how prompt paraphrasing affects response similarity across language models. Starting with 30 prompts from our evaluation set INFINITY-CHAT100, we use `gpt-4.1-2025-04-14` to generate 4 paraphrases for each original prompt, ensuring the paraphrases maintain similar semantic meaning without drastic changes in connotation, with further LLM judge verification. For each original prompt and its paraphrases (30 queries × 5 variants = 150 prompts total), we generate 20 responses using 42 representative models from different model families and sizes. We use consistent sampling parameters from our previous experiments: top-p= 0.9 and temperature= 1. We then obtain sentence embeddings for all responses using OpenAI's `text-embedding-3-small` model.

To measure response consistency, we compute two types of semantic similarity scores:

- **Within-prompt similarity**: Average pairwise similarity among the 20 responses generated from the same original prompt
- **Cross-paraphrase similarity**: Average similarity between responses from the original prompt and responses from its paraphrases

Our results across all 42 models show that the *within-prompt similarity* averaged 0.821, while the *cross-paraphrase similarity* averaged 0.781. Although responses to the original prompts showed

slightly higher similarity scores, responses to paraphrased prompts also demonstrated high similarity (difference of only 0.04). This suggests that language models generate relatively consistent responses even when prompts are paraphrased.

To better illustrate the observed similarities, we include concrete examples that highlight how model responses vary across paraphrased prompts and across different models in Table 17 and 18. In some cases, models exhibit high-level conceptual similarity, for instance, in Table 17, all responses frame the metaphor around the idea that "time is a river." In other cases, the similarity is more surface-level; for example, in Table 18, many responses reuse the word "profound" in the opening sentence.

Table 5: The full list of models that we consider and models that are selected for the main paper.

| Type | In Main? | HuggingFace Model ID |
|---|---|---|
| Open-Source | | meta-llama/Llama-3.1-8B-Instruct |
| | x | meta-llama/Llama-3.1-70B-Instruct |
| | x | meta-llama/Meta-Llama-3.1-405B-Instruct-Turbo |
| | | meta-llama/Llama-3.2-1B-Instruct |
| | | meta-llama/Llama-3.2-3B-Instruct |
| | x | meta-llama/Llama-3.3-70B-Instruct |
| | | google/gemma-2-2b-it |
| | | google/gemma-2-9b-it |
| | x | google/gemma-2-27b-it |
| | | google/gemma-1.1-2b-it |
| | | google/gemma-1.1-7b-it |
| | | Qwen/Qwen1.5-0.5B-Chat |
| | | Qwen/Qwen1.5-1.8B-Chat |
| | | Qwen/Qwen1.5-4B-Chat |
| | | Qwen/Qwen1.5-7B-Chat |
| | | Qwen/Qwen1.5-14B-Chat |
| | | Qwen/Qwen1.5-32B-Chat |
| | | Qwen/Qwen1.5-72B-Chat |
| | x | Qwen/Qwen1.5-110B-Chat |
| | | Qwen/Qwen2-0.5B-Instruct |
| | | Qwen/Qwen2-1.5B-Instruct |
| | x | Qwen/Qwen2-72B-Instruct |
| | | Qwen/Qwen2.5-0.5B-Instruct |
| | | Qwen/Qwen2.5-1.5B-Instruct |
| | | Qwen/Qwen2.5-3B-Instruct |
| | | Qwen/Qwen2.5-7B-Instruct |
| | | Qwen/Qwen2.5-14B-Instruct |
| | | Qwen/Qwen2.5-32B-Instruct |
| | x | Qwen/Qwen2.5-72B-Instruct |
| | | Qwen/Qwen2.5-7B-Instruct-1M |
| | | Qwen/Qwen2.5-14B-Instruct-1M |
| | | Qwen/Qwen3-0.6B |
| | | Qwen/Qwen3-1.7B |
| | | Qwen/Qwen3-4B |
| | | Qwen/Qwen3-8B |
| | | Qwen/Qwen3-14B |
| | x | Qwen/Qwen3-32B |
| | x | deepseek-ai/DeepSeek-V3 |
| | | mistralai/Mistral-Small-24B-Instruct-2501 |
| | | mistralai/Mistral-7B-Instruct-v0.1 |
| | | mistralai/Mistral-7B-Instruct-v0.2 |
| | | mistralai/Mistral-7B-Instruct-v0.3 |
| | | mistralai/Ministral-8B-Instruct-2410 |
| | | mistralai/Mistral-Nemo-Instruct-2407 |
| | | mistralai/Mistral-Small-Instruct-2409 |
| | x | mistralai/Mistral-Large-Instruct-2411 |
| | x | mistralai/Mixtral-8x7B-Instruct-v0.1 |
| | | microsoft/Phi-3-mini-128k-instruct |
| | | microsoft/Phi-3.5-mini-instruct |
| | x | microsoft/phi-4 |
| | | CohereForAI/aya-expanse-8b |
| | x | CohereForAI/aya-expanse-32b |
| | x | CohereForAI/c4ai-command-r-plus-08-2024 |
| | | CohereForAI/c4ai-command-r-08-2024 |
| | x | allenai/OLMo-2-1124-13B-Instruct |
| | | allenai/OLMo-2-1124-7B-Instruct |
| | | allenai/Llama-3.1-Tulu-3-8B |
| | x | allenai/Llama-3.1-Tulu-3-70B |
| Closed-Source | | gpt-4o-2024-11-20 |
| | x | gpt-4o-2024-08-06 |
| | | gpt-4o-2024-05-13 |
| | x | gpt-4o-mini-2024-07-18 |
| | | gpt-4-turbo-2024-04-09 |
| | x | claude-3-5-sonnet-20241022 |
| | | claude-3-5-haiku-20241022 |
| | x | claude-3-sonnet-20240229 |
| | | claude-3-haiku-20240307 |
| | x | claude-3-opus-20240229 |
| | | gemini-1.5-flash |
| | x | gemini-1.5-pro |
| | x | gemini-2.0-flash |
| | | gemini-2.0-flash-lite-preview-02-05 |
| | x | qwen-max-2025-01-25 |
| | x | qwen-plus-2025-01-25 |
| | | qwen-turbo-2024-11-01 |

Table 6: Full results of the intra-model repetition analysis in Figure 4 of the main paper.

| Similarity Range | 0.9-1.0 | 0.8-0.9 | 0.7-0.8 | 0.6-0.7 | 0.5-0.6 | 0.4-0.5 | 0.3-0.4 | 0.2-0.3 | 0.1-0.2 | 0.0-0.1 |
|---|---|---|---|---|---|---|---|---|---|---|
| **Average** | **33.65** | **36.47** | **16.29** | **7.46** | **3.00** | **1.54** | **1.13** | **0.43** | **0.04** | **0.00** |
| gpt-4o-2024-11-20 | 51.00 | 36.00 | 10.00 | 1.00 | 2.00 | 0.00 | 0.00 | 0.00 | 0.00 | 0.00 |
| gpt-4o-2024-08-06 | 44.00 | 37.00 | 12.00 | 4.00 | 1.00 | 0.00 | 2.00 | 0.00 | 0.00 | 0.00 |
| gpt-4o-2024-05-13 | 40.00 | 36.00 | 16.00 | 5.00 | 1.00 | 1.00 | 1.00 | 0.00 | 0.00 | 0.00 |
| gpt-4o-mini-2024-07-18 | 53.00 | 34.00 | 9.00 | 4.00 | 0.00 | 0.00 | 0.00 | 0.00 | 0.00 | 0.00 |
| gpt-4-turbo-2024-04-09 | 38.00 | 44.00 | 11.00 | 5.00 | 1.00 | 0.00 | 1.00 | 0.00 | 0.00 | 0.00 |
| claude-3-5-sonnet-20241022 | 61.00 | 22.00 | 9.00 | 5.00 | 2.00 | 1.00 | 0.00 | 0.00 | 0.00 | 0.00 |
| claude-3-5-haiku-20241022 | 56.00 | 33.00 | 7.00 | 3.00 | 1.00 | 0.00 | 0.00 | 0.00 | 0.00 | 0.00 |
| claude-3-sonnet-20240229 | 48.00 | 36.00 | 10.00 | 5.00 | 1.00 | 0.00 | 0.00 | 0.00 | 0.00 | 0.00 |
| claude-3-haiku-20240307 | 48.00 | 33.00 | 15.00 | 2.00 | 1.00 | 0.00 | 1.00 | 0.00 | 0.00 | 0.00 |
| claude-3-opus-20240229 | 59.00 | 27.00 | 7.00 | 3.00 | 4.00 | 0.00 | 0.00 | 0.00 | 0.00 | 0.00 |
| deepseek-ai/DeepSeek-V3 | 42.00 | 39.00 | 13.00 | 3.00 | 0.00 | 3.00 | 0.00 | 0.00 | 0.00 | 0.00 |
| meta-llama/Meta-Llama-3.1-405B-Instruct-Turbo | 43.00 | 38.00 | 11.00 | 3.00 | 4.00 | 0.00 | 0.00 | 1.00 | 0.00 | 0.00 |
| meta-llama/Llama-3.1-8B-Instruct | 19.00 | 52.00 | 14.00 | 5.00 | 6.00 | 1.00 | 2.00 | 1.00 | 0.00 | 0.00 |
| meta-llama/Llama-3.1-70B-Instruct | 23.00 | 44.00 | 23.00 | 5.00 | 2.00 | 2.00 | 0.00 | 1.00 | 0.00 | 0.00 |
| meta-llama/Llama-3.2-1B-Instruct | 5.00 | 34.00 | 38.00 | 8.00 | 5.00 | 6.00 | 1.00 | 3.00 | 0.00 | 0.00 |
| meta-llama/Llama-3.2-3B-Instruct | 20.00 | 44.00 | 20.00 | 5.00 | 5.00 | 2.00 | 2.00 | 2.00 | 0.00 | 0.00 |
| meta-llama/Llama-3.3-70B-Instruct | 51.00 | 30.00 | 12.00 | 4.00 | 1.00 | 1.00 | 1.00 | 0.00 | 0.00 | 0.00 |
| google/gemma-2-2b-it | 19.00 | 46.00 | 19.00 | 9.00 | 3.00 | 2.00 | 2.00 | 0.00 | 0.00 | 0.00 |
| google/gemma-2-9b-it | 30.00 | 41.00 | 19.00 | 6.00 | 0.00 | 2.00 | 2.00 | 0.00 | 0.00 | 0.00 |
| google/gemma-2-27b-it | 33.00 | 43.00 | 14.00 | 7.00 | 0.00 | 2.00 | 1.00 | 0.00 | 0.00 | 0.00 |
| google/gemma-1.1-2b-it | 17.00 | 39.00 | 30.00 | 8.00 | 3.00 | 2.00 | 1.00 | 0.00 | 0.00 | 0.00 |
| google/gemma-1.1-7b-it | 22.00 | 45.00 | 18.00 | 12.00 | 2.00 | 1.00 | 0.00 | 0.00 | 0.00 | 0.00 |
| Qwen/Qwen2.5-0.5B-Instruct | 1.00 | 23.00 | 24.00 | 27.00 | 7.00 | 13.00 | 2.00 | 3.00 | 0.00 | 0.00 |
| Qwen/Qwen2.5-1.5B-Instruct | 3.00 | 39.00 | 20.00 | 16.00 | 11.00 | 8.00 | 2.00 | 1.00 | 0.00 | 0.00 |
| Qwen/Qwen2.5-3B-Instruct | 14.00 | 41.00 | 26.00 | 12.00 | 3.00 | 2.00 | 1.00 | 1.00 | 0.00 | 0.00 |
| Qwen/Qwen2.5-7B-Instruct | 31.00 | 35.00 | 23.00 | 5.00 | 4.00 | 1.00 | 0.00 | 1.00 | 0.00 | 0.00 |
| Qwen/Qwen2.5-14B-Instruct | 34.00 | 35.00 | 22.00 | 5.00 | 2.00 | 1.00 | 0.00 | 1.00 | 0.00 | 0.00 |
| Qwen/Qwen2.5-32B-Instruct | 36.00 | 44.00 | 16.00 | 2.00 | 1.00 | 0.00 | 0.00 | 1.00 | 0.00 | 0.00 |
| Qwen/Qwen2.5-72B-Instruct | 48.00 | 26.00 | 19.00 | 4.00 | 1.00 | 1.00 | 1.00 | 0.00 | 0.00 | 0.00 |
| Qwen/Qwen2.5-7B-Instruct-1M | 31.00 | 36.00 | 16.00 | 12.00 | 3.00 | 0.00 | 2.00 | 0.00 | 0.00 | 0.00 |
| Qwen/Qwen2.5-14B-Instruct-1M | 43.00 | 27.00 | 16.00 | 9.00 | 3.00 | 1.00 | 0.00 | 1.00 | 0.00 | 0.00 |
| Qwen/Qwen2-0.5B-Instruct | 0.00 | 20.00 | 21.00 | 23.00 | 20.00 | 9.00 | 4.00 | 2.00 | 1.00 | 0.00 |
| Qwen/Qwen2-1.5B-Instruct | 2.00 | 32.00 | 24.00 | 22.00 | 11.00 | 3.00 | 4.00 | 2.00 | 0.00 | 0.00 |
| Qwen/Qwen2-72B-Instruct | 35.00 | 35.00 | 24.00 | 5.00 | 0.00 | 0.00 | 1.00 | 0.00 | 0.00 | 0.00 |
| Qwen/Qwen1.5-0.5B-Chat | 0.00 | 23.00 | 27.00 | 24.00 | 12.00 | 6.00 | 6.00 | 2.00 | 0.00 | 0.00 |
| Qwen/Qwen1.5-1.8B-Chat | 12.00 | 41.00 | 27.00 | 9.00 | 4.00 | 3.00 | 3.00 | 1.00 | 0.00 | 0.00 |
| Qwen/Qwen1.5-4B-Chat | 6.00 | 46.00 | 24.00 | 14.00 | 4.00 | 2.00 | 3.00 | 1.00 | 0.00 | 0.00 |
| Qwen/Qwen1.5-7B-Chat | 26.00 | 40.00 | 16.00 | 10.00 | 4.00 | 2.00 | 1.00 | 1.00 | 0.00 | 0.00 |
| Qwen/Qwen1.5-14B-Chat | 37.00 | 33.00 | 22.00 | 3.00 | 5.00 | 0.00 | 0.00 | 0.00 | 0.00 | 0.00 |
| Qwen/Qwen1.5-32B-Chat | 37.00 | 37.00 | 17.00 | 7.00 | 1.00 | 1.00 | 0.00 | 0.00 | 0.00 | 0.00 |
| Qwen/Qwen1.5-72B-Chat | 50.00 | 35.00 | 9.00 | 3.00 | 1.00 | 1.00 | 1.00 | 0.00 | 0.00 | 0.00 |
| Qwen/Qwen1.5-110B-Chat | 48.00 | 36.00 | 10.00 | 2.00 | 3.00 | 0.00 | 1.00 | 0.00 | 0.00 | 0.00 |
| mistralai/Mistral-Small-24B-Instruct-2501 | 31.00 | 39.00 | 17.00 | 4.00 | 4.00 | 2.00 | 2.00 | 1.00 | 0.00 | 0.00 |
| mistralai/Mistral-7B-Instruct-v0.1 | 24.00 | 42.00 | 23.00 | 5.00 | 1.00 | 2.00 | 2.00 | 1.00 | 0.00 | 0.00 |
| mistralai/Mistral-7B-Instruct-v0.2 | 38.00 | 40.00 | 13.00 | 4.00 | 3.00 | 1.00 | 1.00 | 0.00 | 0.00 | 0.00 |
| mistralai/Mistral-7B-Instruct-v0.3 | 30.00 | 39.00 | 18.00 | 8.00 | 3.00 | 1.00 | 1.00 | 0.00 | 0.00 | 0.00 |
| mistralai/Ministral-8B-Instruct-2410 | 10.00 | 38.00 | 25.00 | 14.00 | 7.00 | 2.00 | 3.00 | 1.00 | 0.00 | 0.00 |
| mistralai/Mistral-Nemo-Instruct-2407 | 19.00 | 42.00 | 22.00 | 9.00 | 3.00 | 1.00 | 4.00 | 0.00 | 0.00 | 0.00 |
| mistralai/Mistral-Small-Instruct-2409 | 23.00 | 42.00 | 20.00 | 6.00 | 6.00 | 3.00 | 0.00 | 0.00 | 0.00 | 0.00 |
| mistralai/Mistral-Large-Instruct-2411 | 43.00 | 34.00 | 15.00 | 5.00 | 2.00 | 0.00 | 1.00 | 0.00 | 0.00 | 0.00 |
| mistralai/Mixtral-8x7B-Instruct-v0.1 | 45.00 | 41.00 | 9.00 | 4.00 | 1.00 | 0.00 | 0.00 | 0.00 | 0.00 | 0.00 |
| microsoft/phi-4 | 38.00 | 40.00 | 11.00 | 9.00 | 2.00 | 0.00 | 0.00 | 0.00 | 0.00 | 0.00 |
| microsoft/Phi-3.5-mini-instruct | 33.00 | 42.00 | 12.00 | 8.00 | 2.00 | 2.00 | 1.00 | 0.00 | 0.00 | 0.00 |
| microsoft/Phi-3-mini-128k-instruct | 15.00 | 40.00 | 22.00 | 13.00 | 5.00 | 2.00 | 2.00 | 1.00 | 0.00 | 0.00 |
| o1-2024-12-17 | 27.00 | 37.00 | 20.00 | 9.00 | 4.00 | 0.00 | 3.00 | 0.00 | 0.00 | 0.00 |
| o1-mini-2024-09-12 | 38.00 | 40.00 | 17.00 | 4.00 | 1.00 | 0.00 | 0.00 | 0.00 | 0.00 | 0.00 |
| o1-preview-2024-09-12 | 40.00 | 33.00 | 16.00 | 8.00 | 1.00 | 2.00 | 0.00 | 0.00 | 0.00 | 0.00 |
| o3-mini-2025-01-31 | 34.00 | 35.00 | 18.00 | 10.00 | 0.00 | 2.00 | 1.00 | 0.00 | 0.00 | 0.00 |
| CohereForAI/aya-expanse-8b | 40.00 | 44.00 | 7.00 | 7.00 | 1.00 | 0.00 | 1.00 | 0.00 | 0.00 | 0.00 |
| CohereForAI/aya-expanse-32b | 50.00 | 36.00 | 8.00 | 5.00 | 1.00 | 0.00 | 0.00 | 0.00 | 0.00 | 0.00 |
| CohereForAI/c4ai-command-r-plus-08-2024 | 24.00 | 32.00 | 19.00 | 10.00 | 7.00 | 3.00 | 3.00 | 1.00 | 1.00 | 0.00 |
| CohereForAI/c4ai-command-r-08-2024 | 27.00 | 35.00 | 14.00 | 16.00 | 2.00 | 4.00 | 1.00 | 1.00 | 0.00 | 0.00 |
| allenai/OLMo-2-1124-13B-Instruct | 29.00 | 39.00 | 12.00 | 9.00 | 7.00 | 2.00 | 1.00 | 1.00 | 0.00 | 0.00 |
| allenai/OLMo-2-1124-7B-Instruct | 30.00 | 38.00 | 15.00 | 10.00 | 3.00 | 3.00 | 1.00 | 0.00 | 0.00 | 0.00 |
| allenai/Llama-3.1-Tulu-3-8B | 27.00 | 36.00 | 22.00 | 8.00 | 3.00 | 1.00 | 2.00 | 0.00 | 1.00 | 0.00 |
| allenai/Llama-3.1-Tulu-3-70B | 28.00 | 39.00 | 18.00 | 11.00 | 1.00 | 2.00 | 0.00 | 1.00 | 0.00 | 0.00 |
| qwen-max-2025-01-25 | 55.00 | 37.00 | 4.00 | 3.00 | 1.00 | 0.00 | 0.00 | 0.00 | 0.00 | 0.00 |
| qwen-plus-2025-01-25 | 56.00 | 28.00 | 11.00 | 3.00 | 2.00 | 0.00 | 0.00 | 0.00 | 0.00 | 0.00 |
| qwen-turbo-2024-11-01 | 37.00 | 33.00 | 20.00 | 8.00 | 1.00 | 1.00 | 0.00 | 0.00 | 0.00 | 0.00 |
| Qwen/Qwen3-0.6B | 13.00 | 43.00 | 27.00 | 8.00 | 6.00 | 1.00 | 2.00 | 0.00 | 0.00 | 0.00 |
| Qwen/Qwen3-1.7B | 44.00 | 30.00 | 14.00 | 11.00 | 1.00 | 0.00 | 0.00 | 0.00 | 0.00 | 0.00 |
| Qwen/Qwen3-4B | 51.00 | 37.00 | 4.00 | 4.00 | 2.00 | 2.00 | 0.00 | 0.00 | 0.00 | 0.00 |
| Qwen/Qwen3-8B | 56.00 | 26.00 | 15.00 | 1.00 | 1.00 | 0.00 | 1.00 | 0.00 | 0.00 | 0.00 |
| Qwen/Qwen3-14B | 52.00 | 31.00 | 10.00 | 4.00 | 0.00 | 1.00 | 2.00 | 0.00 | 0.00 | 0.00 |
| Qwen/Qwen3-32B | 40.00 | 36.00 | 14.00 | 6.00 | 1.00 | 1.00 | 2.00 | 0.00 | 0.00 | 0.00 |
| gemini-1.5-flash | 62.00 | 26.00 | 5.00 | 6.00 | 1.00 | 0.00 | 0.00 | 0.00 | 0.00 | 0.00 |
| gemini-1.5-pro | 53.00 | 32.00 | 11.00 | 3.00 | 1.00 | 0.00 | 0.00 | 0.00 | 0.00 | 0.00 |
| gemini-2.0-flash | 40.00 | 41.00 | 11.00 | 4.00 | 2.00 | 0.00 | 2.00 | 0.00 | 0.00 | 0.00 |
| gemini-2.0-flash-lite-preview-02-05 | 40.00 | 41.00 | 8.00 | 6.00 | 3.00 | 1.00 | 1.00 | 0.00 | 0.00 | 0.00 |

Table 7: Full results of the inter-model repetition analysis in Figure 6 of the main paper (Part 1).

| | gpt-4o-2024-11-20 | gpt-4o-2024-08-06 | gpt-4o-2024-05-13 | gpt-4o-mini-2024-07-18 | gpt-4-turbo-2024-04-09 | claude-3-5-sonnet-20241022 | claude-3-5-haiku-20241022 | claude-3-sonnet-20240229 | claude-3-haiku-20240307 | claude-3-opus-20240229 | deepseek-ai/DeepSeek-V3 | Llama-3.1-405B-Instruct | meta-llama/Llama-3.1-8B-Instruct | meta-llama/Llama-3.1-70B-Instruct | meta-llama/Llama-3.2-1B-Instruct | meta-llama/Llama-3.2-3B-Instruct | meta-llama/Llama-3.3-70B-Instruct |
|---|---|---|---|---|---|---|---|---|---|---|---|---|---|---|---|---|---|
| gpt-4o-2024-11-20 | 88.2 | 82.2 | 82.2 | 81.4 | 79.0 | 75.1 | 76.3 | 76.7 | 76.4 | 76.5 | 81.4 | 76.9 | 74.7 | 75.8 | 70.1 | 74.0 | 76.7 |
| gpt-4o-2024-08-06 | 82.2 | 86.1 | 83.7 | 83.0 | 80.9 | 74.9 | 76.3 | 77.2 | 76.8 | 77.7 | 80.6 | 77.3 | 75.4 | 76.2 | 70.6 | 74.4 | 77.0 |
| gpt-4o-2024-05-13 | 82.2 | 83.7 | 85.5 | 82.6 | 80.9 | 75.2 | 76.9 | 77.6 | 77.0 | 77.8 | 80.7 | 77.6 | 75.6 | 76.6 | 70.7 | 74.6 | 77.2 |
| gpt-4o-mini-2024-07-18 | 81.4 | 83.0 | 82.6 | 88.3 | 81.0 | 74.7 | 76.0 | 77.3 | 77.1 | 77.8 | 81.5 | 77.2 | 75.6 | 76.2 | 71.0 | 74.8 | 77.0 |
| gpt-4-turbo-2024-04-09 | 79.0 | 80.9 | 80.9 | 81.0 | 85.9 | 74.5 | 75.8 | 76.3 | 76.1 | 77.2 | 79.0 | 76.5 | 74.7 | 75.6 | 69.8 | 73.5 | 75.9 |
| claude-3-5-sonnet-20241022 | 75.1 | 74.9 | 75.2 | 74.7 | 74.5 | 88.7 | 77.6 | 75.3 | 74.0 | 75.8 | 76.0 | 75.4 | 72.8 | 74.2 | 68.6 | 72.4 | 75.0 |
| claude-3-5-haiku-20241022 | 76.3 | 76.3 | 76.9 | 76.0 | 75.8 | 77.6 | 89.1 | 76.9 | 76.0 | 77.9 | 77.1 | 76.7 | 74.6 | 75.7 | 70.4 | 74.1 | 76.6 |
| claude-3-sonnet-20240229 | 76.7 | 77.2 | 77.6 | 77.3 | 76.3 | 75.3 | 76.9 | 87.5 | 80.0 | 79.2 | 77.6 | 77.0 | 74.6 | 75.7 | 70.7 | 74.4 | 76.8 |
| claude-3-haiku-20240307 | 76.4 | 76.8 | 77.0 | 77.1 | 76.1 | 74.0 | 76.0 | 80.0 | 86.9 | 78.3 | 76.8 | 76.4 | 74.4 | 75.0 | 70.4 | 73.9 | 75.8 |
| claude-3-opus-20240229 | 76.5 | 77.7 | 77.8 | 77.8 | 77.2 | 75.8 | 77.9 | 79.2 | 78.3 | 88.4 | 77.9 | 77.3 | 75.1 | 76.2 | 70.5 | 74.5 | 77.0 |
| deepseek-ai/DeepSeek-V3 | 81.4 | 80.6 | 80.7 | 81.5 | 79.0 | 76.0 | 77.1 | 77.6 | 76.8 | 77.9 | 86.3 | 77.6 | 75.9 | 76.6 | 71.2 | 75.2 | 77.6 |
| Llama-3.1-405B-Instruct | 76.9 | 77.3 | 77.6 | 77.2 | 76.5 | 75.4 | 76.7 | 77.0 | 76.4 | 77.3 | 77.6 | 85.7 | 78.6 | 80.7 | 73.5 | 78.3 | 81.8 |
| meta-llama/Llama-3.1-8B-Instruct | 74.7 | 75.4 | 75.6 | 75.6 | 74.7 | 72.8 | 74.6 | 74.6 | 74.4 | 75.1 | 75.9 | 78.6 | 80.7 | 78.5 | 73.2 | 77.7 | 78.6 |
| meta-llama/Llama-3.1-70B-Instruct | 75.8 | 76.2 | 76.6 | 75.6 | 74.2 | 75.7 | 75.7 | 75.0 | 76.2 | 76.6 | 76.6 | 80.7 | 78.5 | 82.3 | 72.3 | 77.4 | 81.6 |
| meta-llama/Llama-3.2-1B-Instruct | 70.1 | 70.6 | 70.7 | 71.0 | 69.8 | 68.6 | 70.4 | 70.7 | 70.4 | 70.5 | 71.2 | 73.5 | 73.2 | 72.3 | 74.6 | 74.0 | 72.9 |
| meta-llama/Llama-3.2-3B-Instruct | 74.0 | 74.4 | 74.6 | 74.8 | 73.5 | 72.4 | 74.1 | 74.4 | 73.9 | 74.5 | 75.2 | 78.3 | 77.7 | 77.4 | 74.0 | 79.9 | 78.0 |
| meta-llama/Llama-3.3-70B-Instruct | 76.7 | 77.0 | 77.2 | 77.0 | 75.9 | 75.0 | 76.6 | 76.8 | 75.8 | 77.0 | 77.6 | 81.8 | 78.6 | 81.6 | 72.9 | 78.0 | 87.1 |
| google/gemma-2-2b-it | 72.3 | 71.7 | 72.1 | 72.7 | 70.9 | 69.8 | 72.3 | 72.1 | 71.8 | 71.9 | 73.6 | 72.8 | 72.3 | 71.7 | 70.1 | 72.6 | 72.6 |
| google/gemma-2-9b-it | 73.3 | 72.8 | 73.1 | 73.5 | 71.8 | 71.0 | 73.2 | 73.1 | 72.4 | 72.9 | 74.7 | 73.3 | 72.7 | 72.5 | 69.6 | 72.6 | 73.4 |
| google/gemma-2-27b-it | 73.6 | 73.0 | 73.6 | 73.9 | 72.5 | 71.7 | 74.1 | 73.4 | 72.6 | 73.3 | 75.0 | 73.9 | 72.8 | 73.0 | 69.3 | 72.6 | 73.9 |
| google/gemma-1.1-2b-it | 70.3 | 71.1 | 71.0 | 71.8 | 69.9 | 67.7 | 69.7 | 71.0 | 70.1 | 70.4 | 71.6 | 70.1 | 70.0 | 69.8 | 68.1 | 70.2 | 70.1 |
| google/gemma-1.1-7b-it | 71.5 | 71.9 | 72.1 | 72.7 | 70.6 | 69.1 | 71.3 | 71.9 | 70.9 | 71.6 | 72.8 | 71.3 | 71.0 | 70.8 | 67.8 | 71.0 | 71.6 |
| Qwen/Qwen2.5-0.5B-Instruct | 66.6 | 67.1 | 66.9 | 67.4 | 66.4 | 63.1 | 64.2 | 66.1 | 66.2 | 66.2 | 66.4 | 65.6 | 65.4 | 65.1 | 63.8 | 65.1 | 65.6 |
| Qwen/Qwen2.5-1.5B-Instruct | 70.3 | 71.0 | 70.9 | 71.3 | 70.4 | 66.9 | 68.5 | 70.0 | 70.2 | 70.2 | 70.3 | 69.3 | 69.0 | 69.0 | 66.3 | 68.6 | 69.0 |
| Qwen/Qwen2.5-3B-Instruct | 75.0 | 75.8 | 76.1 | 76.2 | 74.6 | 70.8 | 72.6 | 74.0 | 73.7 | 74.2 | 75.5 | 73.1 | 72.7 | 72.4 | 69.1 | 72.0 | 73.4 |
| Qwen/Qwen2.5-7B-Instruct | 76.6 | 77.9 | 78.1 | 78.1 | 76.8 | 72.5 | 74.1 | 75.3 | 75.0 | 75.7 | 77.3 | 75.3 | 74.3 | 74.4 | 70.2 | 73.5 | 75.2 |
| Qwen/Qwen2.5-14B-Instruct | 77.6 | 79.1 | 79.2 | 78.9 | 78.2 | 73.5 | 75.2 | 75.6 | 75.5 | 76.7 | 77.9 | 75.9 | 74.5 | 75.0 | 69.9 | 73.7 | 75.9 |
| Qwen/Qwen2.5-32B-Instruct | 78.3 | 79.4 | 79.7 | 79.5 | 78.6 | 73.4 | 75.0 | 76.4 | 76.0 | 76.6 | 78.4 | 76.1 | 74.2 | 75.1 | 69.6 | 73.4 | 75.8 |
| Qwen/Qwen2.5-72B-Instruct | 79.0 | 80.3 | 80.5 | 80.5 | 80.0 | 74.5 | 75.3 | 76.8 | 76.6 | 76.6 | 79.5 | 76.8 | 74.9 | 75.6 | 70.4 | 74.1 | 76.3 |
| Qwen/Qwen2.5-7B-Instruct-1M | 76.5 | 77.1 | 77.0 | 77.6 | 75.4 | 71.3 | 73.6 | 74.1 | 73.4 | 74.5 | 77.2 | 73.9 | 73.4 | 73.3 | 69.5 | 72.5 | 74.1 |
| Qwen/Qwen2.5-14B-Instruct-1M | 78.3 | 78.8 | 79.0 | 79.6 | 77.5 | 73.4 | 75.2 | 75.4 | 75.3 | 75.8 | 79.2 | 75.9 | 74.5 | 74.9 | 70.0 | 73.5 | 76.0 |
| Qwen/Qwen2-0.5B-Instruct | 64.6 | 65.0 | 64.8 | 65.0 | 64.4 | 61.8 | 62.5 | 64.4 | 64.7 | 64.6 | 64.0 | 63.4 | 63.4 | 63.4 | 62.1 | 63.2 | 63.6 |
| Qwen/Qwen2-1.5B-Instruct | 70.1 | 70.5 | 70.5 | 70.7 | 70.0 | 67.1 | 67.6 | 70.1 | 70.2 | 70.2 | 69.6 | 69.5 | 68.6 | 68.8 | 66.6 | 68.7 | 69.0 |
| Qwen/Qwen2-72B-Instruct | 77.2 | 78.3 | 78.2 | 78.2 | 77.2 | 73.8 | 75.0 | 77.4 | 77.1 | 76.9 | 78.0 | 77.1 | 75.1 | 75.5 | 70.9 | 74.6 | 76.7 |
| Qwen/Qwen1.5-0.5B-Chat | 65.3 | 65.8 | 65.7 | 66.1 | 65.0 | 62.5 | 63.1 | 65.3 | 65.5 | 65.4 | 65.0 | 64.7 | 64.2 | 64.0 | 62.9 | 64.1 | 64.5 |
| Qwen/Qwen1.5-1.8B-Chat | 71.1 | 71.9 | 71.7 | 72.2 | 70.9 | 67.3 | 68.6 | 71.5 | 71.7 | 71.5 | 71.3 | 70.7 | 70.2 | 69.9 | 68.3 | 70.0 | 70.3 |
| Qwen/Qwen1.5-4B-Chat | 71.6 | 72.3 | 72.2 | 72.3 | 71.3 | 68.2 | 68.6 | 71.6 | 71.5 | 71.5 | 71.1 | 70.8 | 69.9 | 70.1 | 67.3 | 69.6 | 70.3 |
| Qwen/Qwen1.5-7B-Chat | 74.1 | 75.1 | 75.1 | 75.2 | 74.3 | 71.1 | 72.1 | 74.6 | 74.1 | 74.2 | 74.6 | 72.9 | 73.0 | 73.0 | 69.8 | 72.8 | 73.5 |
| Qwen/Qwen1.5-14B-Chat | 76.0 | 77.2 | 77.1 | 77.7 | 76.5 | 72.3 | 73.8 | 76.3 | 75.5 | 76.0 | 76.5 | 75.1 | 74.3 | 74.3 | 70.5 | 73.9 | 74.7 |
| Qwen/Qwen1.5-32B-Chat | 76.9 | 77.6 | 77.7 | 78.0 | 76.9 | 73.0 | 74.3 | 77.4 | 76.7 | 76.6 | 76.8 | 76.1 | 74.6 | 75.2 | 70.4 | 74.1 | 75.9 |
| Qwen/Qwen1.5-72B-Chat | 78.2 | 79.0 | 79.0 | 79.8 | 78.2 | 73.9 | 74.9 | 77.7 | 77.4 | 77.5 | 78.2 | 76.6 | 75.0 | 75.5 | 71.0 | 74.5 | 76.1 |
| Qwen/Qwen1.5-110B-Chat | 78.1 | 79.3 | 79.0 | 79.4 | 78.6 | 73.8 | 75.0 | 77.7 | 77.3 | 76.9 | 78.0 | 76.8 | 75.2 | 75.7 | 70.7 | 74.3 | 76.2 |
| mistralai/Mistral-Small-24B-Instruct-2501 | 76.3 | 77.0 | 77.1 | 76.9 | 75.9 | 73.8 | 75.6 | 76.0 | 75.2 | 76.1 | 77.3 | 76.3 | 75.3 | 75.2 | 70.9 | 74.5 | 76.0 |
| mistralai/Mistral-7B-Instruct-v0.1 | 73.7 | 74.6 | 74.4 | 74.3 | 73.1 | 69.5 | 70.2 | 72.9 | 73.0 | 73.1 | 72.9 | 72.5 | 71.3 | 71.7 | 68.0 | 71.0 | 72.2 |
| mistralai/Mistral-7B-Instruct-v0.2 | 75.3 | 76.5 | 76.2 | 76.6 | 75.3 | 72.1 | 73.4 | 75.5 | 75.0 | 75.7 | 76.1 | 75.2 | 73.8 | 74.3 | 70.1 | 73.5 | 74.9 |
| mistralai/Mistral-7B-Instruct-v0.3 | 75.4 | 76.6 | 76.2 | 76.5 | 75.6 | 71.9 | 73.2 | 75.3 | 74.8 | 75.3 | 76.4 | 74.7 | 73.6 | 73.8 | 69.6 | 72.9 | 74.1 |
| mistralai/Ministral-8B-Instruct-2410 | 74.7 | 74.9 | 75.3 | 75.3 | 74.2 | 71.6 | 72.7 | 74.1 | 73.7 | 74.1 | 75.6 | 73.9 | 73.3 | 73.1 | 69.8 | 73.0 | 73.6 |
| mistralai/Mistral-Nemo-Instruct-2407 | 75.6 | 75.4 | 75.9 | 75.6 | 75.0 | 72.5 | 73.3 | 75.1 | 74.2 | 74.7 | 76.3 | 74.4 | 73.4 | 73.5 | 69.7 | 73.0 | 74.1 |
| mistralai/Mistral-Small-Instruct-2409 | 76.1 | 75.7 | 76.1 | 75.9 | 74.9 | 73.1 | 74.2 | 75.4 | 74.4 | 75.2 | 76.7 | 75.3 | 74.0 | 74.0 | 69.9 | 73.2 | 74.9 |
| mistralai/Mistral-Large-Instruct-2411 | 76.7 | 75.9 | 76.2 | 76.1 | 75.2 | 73.9 | 75.0 | 75.4 | 74.7 | 75.8 | 77.2 | 75.6 | 74.4 | 74.6 | 70.4 | 73.8 | 75.5 |
| mistralai/Mixtral-8x7B-Instruct-v0.1 | 76.0 | 76.2 | 76.3 | 76.3 | 75.6 | 73.4 | 74.8 | 76.3 | 76.0 | 76.9 | 76.5 | 75.3 | 73.9 | 74.6 | 69.9 | 73.5 | 75.4 |
| microsoft/phi-4 | 78.5 | 80.1 | 80.0 | 80.6 | 78.9 | 74.1 | 76.0 | 76.6 | 76.5 | 77.3 | 80.2 | 76.8 | 76.1 | 76.3 | 71.3 | 75.0 | 76.6 |
| microsoft/Phi-3.5-mini-instruct | 75.8 | 76.6 | 76.7 | 77.3 | 75.9 | 71.9 | 73.6 | 74.8 | 74.5 | 75.2 | 76.7 | 74.6 | 73.8 | 73.8 | 70.1 | 73.2 | 74.5 |
| microsoft/Phi-3-mini-128k-instruct | 74.5 | 75.5 | 75.3 | 75.6 | 74.7 | 71.0 | 72.5 | 73.3 | 72.9 | 74.0 | 75.0 | 73.1 | 72.1 | 72.4 | 68.7 | 71.7 | 72.9 |
| o1-2024-12-17 | 77.9 | 77.6 | 77.9 | 77.6 | 76.2 | 73.4 | 74.6 | 74.6 | 74.0 | 74.8 | 76.9 | 74.6 | 72.4 | 73.4 | 68.0 | 71.7 | 74.2 |
| o1-mini-2024-09-12 | 78.3 | 78.4 | 78.8 | 79.4 | 77.3 | 73.4 | 75.0 | 75.5 | 74.8 | 75.6 | 79.8 | 75.4 | 74.9 | 74.9 | 70.4 | 73.9 | 75.6 |
| o1-preview-2024-09-12 | 79.1 | 79.5 | 79.7 | 79.6 | 77.9 | 73.6 | 74.8 | 76.3 | 75.6 | 75.9 | 79.0 | 76.1 | 74.4 | 75.3 | 69.7 | 73.2 | 76.0 |
| o3-mini-2025-01-31 | 78.1 | 78.4 | 78.6 | 79.1 | 77.2 | 73.8 | 75.1 | 75.6 | 75.1 | 75.9 | 78.0 | 75.3 | 73.3 | 74.2 | 69.0 | 72.6 | 75.4 |
| CohereForAI/aya-expanse-8b | 76.1 | 76.3 | 76.6 | 77.1 | 75.6 | 73.0 | 75.2 | 75.9 | 75.8 | 76.3 | 77.6 | 76.4 | 75.9 | 75.5 | 72.4 | 75.4 | 76.1 |
| CohereForAI/aya-expanse-32b | 76.4 | 77.0 | 77.2 | 77.7 | 76.8 | 74.1 | 75.9 | 77.4 | 76.6 | 77.3 | 78.4 | 77.7 | 76.4 | 76.6 | 72.2 | 75.6 | 77.5 |
| CohereForAI/c4ai-command-r-plus-08-2024 | 74.8 | 75.6 | 75.7 | 75.7 | 74.8 | 72.4 | 73.5 | 75.4 | 75.1 | 75.3 | 75.1 | 74.5 | 73.0 | 73.7 | 69.1 | 72.4 | 74.5 |
| CohereForAI/c4ai-command-r-08-2024 | 75.0 | 76.3 | 76.0 | 76.5 | 75.4 | 72.9 | 74.4 | 76.0 | 75.6 | 76.2 | 75.9 | 75.0 | 73.8 | 74.2 | 69.8 | 73.1 | 75.0 |
| allenai/OLMo-2-1124-13B-Instruct | 76.9 | 77.8 | 78.1 | 78.1 | 77.0 | 73.2 | 75.6 | 76.4 | 76.3 | 76.4 | 77.7 | 75.6 | 74.5 | 74.8 | 70.2 | 73.8 | 75.4 |
| allenai/OLMo-2-1124-7B-Instruct | 76.9 | 78.1 | 78.4 | 78.5 | 77.8 | 72.9 | 74.8 | 76.2 | 76.1 | 76.1 | 77.5 | 75.6 | 74.3 | 74.4 | 70.5 | 73.7 | 75.1 |
| allenai/Llama-3.1-Tulu-3-8B | 76.8 | 78.1 | 78.5 | 78.4 | 77.5 | 73.1 | 75.1 | 76.3 | 76.1 | 76.4 | 77.5 | 75.6 | 75.0 | 74.9 | 70.4 | 74.1 | 75.5 |
| allenai/Llama-3.1-Tulu-3-70B | 78.0 | 79.1 | 79.6 | 79.5 | 79.1 | 74.2 | 75.8 | 76.8 | 76.4 | 77.2 | 78.7 | 76.7 | 75.1 | 76.1 | 70.6 | 74.2 | 76.7 |
| qwen-max-2025-01-25 | 80.7 | 80.4 | 80.5 | 81.0 | 78.9 | 75.0 | 76.2 | 77.3 | 76.3 | 77.1 | 81.6 | 76.9 | 75.1 | 75.9 | 70.4 | 74.2 | 77.1 |
| qwen-plus-2025-01-25 | 80.6 | 79.8 | 80.0 | 80.7 | 79.2 | 74.7 | 75.9 | 76.9 | 76.5 | 76.8 | 80.7 | 76.6 | 74.7 | 75.6 | 70.2 | 73.8 | 76.6 |
| qwen-turbo-2024-11-01 | 77.5 | 78.2 | 78.4 | 78.5 | 77.5 | 73.2 | 74.4 | 75.9 | 76.1 | 75.9 | 77.6 | 75.8 | 74.0 | 74.9 | 69.2 | 73.2 | 75.6 |
| Qwen/Qwen3-0.6B | 72.8 | 71.4 | 71.2 | 71.9 | 70.6 | 67.4 | 68.4 | 69.4 | 70.0 | 70.2 | 72.3 | 69.3 | 69.0 | 69.1 | 67.2 | 68.9 | 70.3 |
| Qwen/Qwen3-1.7B | 75.4 | 73.9 | 73.9 | 74.9 | 72.8 | 69.9 | 71.4 | 72.1 | 72.3 | 72.8 | 75.3 | 72.1 | 71.5 | 71.5 | 68.9 | 71.4 | 72.8 |
| Qwen/Qwen3-4B | 77.5 | 76.5 | 76.5 | 77.2 | 75.6 | 72.2 | 73.8 | 74.5 | 74.2 | 74.9 | 77.5 | 74.2 | 73.9 | 74.0 | 69.9 | 73.2 | 75.3 |
| Qwen/Qwen3-8B | 78.8 | 78.1 | 77.8 | 78.8 | 76.4 | 73.5 | 74.9 | 75.2 | 75.1 | 76.4 | 79.2 | 75.2 | 74.5 | 74.8 | 70.2 | 73.9 | 76.6 |
| Qwen/Qwen3-14B | 78.4 | 77.6 | 77.8 | 78.2 | 76.8 | 73.7 | 75.1 | 74.5 | 74.5 | 76.2 | 79.3 | 75.1 | 74.7 | 75.2 | 69.9 | 73.8 | 76.3 |
| Qwen/Qwen3-32B | 78.3 | 77.0 | 77.1 | 77.4 | 76.2 | 73.4 | 74.6 | 74.5 | 73.9 | 75.2 | 78.6 | 74.7 | 73.9 | 74.1 | 69.4 | 73.2 | 75.5 |
| gemini-1.5-flash | 72.8 | 72.5 | 73.1 | 73.1 | 71.9 | 70.9 | 73.2 | 73.0 | 72.4 | 72.7 | 73.5 | 72.9 | 71.5 | 72.2 | 68.0 | 71.2 | 72.8 |
| gemini-1.5-pro | 72.4 | 72.0 | 72.5 | 72.6 | 71.9 | 71.3 | 73.0 | 72.1 | 70.9 | 72.3 | 73.0 | 72.5 | 70.9 | 71.7 | 66.8 | 70.3 | 72.4 |
| gemini-2.0-flash | 74.1 | 73.7 | 73.9 | 74.4 | 73.0 | 72.4 | 73.8 | 73.0 | 72.5 | 73.7 | 75.4 | 74.1 | 72.8 | 72.9 | 69.3 | 72.2 | 74.1 |
| gemini-2.0-flash-lite-preview-02-05 | 74.3 | 73.3 | 73.7 | 73.9 | 72.4 | 72.7 | 74.4 | 73.2 | 72.8 | 73.5 | 75.4 | 74.3 | 73.1 | 73.0 | 69.7 | 72.7 | 74.4 |

Table 8: Full results of the inter-model repetition analysis in Figure 6 of the main paper (Part 2).

| Model | google/gemma-2-2b-it | google/gemma-2-9b-it | google/gemma-2-27b-it | google/gemma-1.1-2b-it | google/gemma-1.1-7b-it | Qwen/Qwen2.5-0.5B-Instruct | Qwen/Qwen2.5-1.5B-Instruct | Qwen/Qwen2.5-3B-Instruct | Qwen/Qwen2.5-7B-Instruct | Qwen/Qwen2.5-14B-Instruct | Qwen/Qwen2.5-32B-Instruct | Qwen/Qwen2.5-72B-Instruct | Qwen/Qwen2.5-7B-Instruct-1M | Qwen/Qwen2.5-14B-Instruct-1M | Qwen/Qwen2-0.5B-Instruct | Qwen/Qwen2-1.5B-Instruct | Qwen/Qwen2-72B-Instruct |
|---|---|---|---|---|---|---|---|---|---|---|---|---|---|---|---|---|---|
| gpt-4o-2024-11-20 | 72.3 | 73.3 | 73.6 | 70.3 | 71.5 | 66.6 | 70.3 | 75.0 | 76.6 | 77.6 | 78.3 | 79.0 | 76.5 | 78.3 | 64.6 | 70.1 | 77.2 |
| gpt-4o-2024-08-06 | 71.7 | 72.8 | 73.0 | 71.1 | 71.9 | 67.1 | 71.0 | 75.8 | 77.9 | 79.1 | 79.4 | 80.3 | 77.1 | 78.8 | 65.0 | 70.5 | 78.3 |
| gpt-4o-2024-05-13 | 72.1 | 73.1 | 73.6 | 71.0 | 72.1 | 66.9 | 70.9 | 76.1 | 78.1 | 79.2 | 79.7 | 80.5 | 77.0 | 79.0 | 64.8 | 70.5 | 78.2 |
| gpt-4o-mini-2024-07-18 | 72.7 | 73.5 | 73.9 | 71.8 | 72.7 | 67.4 | 71.3 | 76.2 | 78.1 | 78.9 | 79.5 | 80.5 | 77.6 | 79.6 | 65.0 | 70.7 | 78.2 |
| gpt-4-turbo-2024-04-09 | 70.9 | 71.8 | 72.5 | 69.9 | 70.6 | 66.4 | 70.4 | 74.6 | 76.8 | 78.2 | 78.6 | 80.0 | 75.4 | 77.5 | 64.4 | 70.0 | 77.2 |
| claude-3-5-sonnet-20241022 | 69.8 | 71.0 | 71.7 | 67.7 | 69.1 | 63.1 | 66.9 | 70.8 | 72.5 | 73.5 | 73.4 | 74.5 | 71.3 | 73.4 | 61.8 | 67.1 | 73.8 |
| claude-3-5-haiku-20241022 | 72.3 | 73.2 | 74.1 | 69.7 | 71.3 | 64.2 | 68.5 | 72.6 | 74.1 | 75.2 | 75.0 | 75.3 | 73.6 | 75.2 | 62.5 | 67.6 | 75.0 |
| claude-3-sonnet-20240229 | 72.1 | 73.1 | 73.4 | 71.0 | 71.9 | 66.1 | 70.0 | 74.0 | 75.3 | 75.6 | 76.4 | 76.8 | 74.1 | 75.4 | 64.4 | 70.1 | 77.4 |
| claude-3-haiku-20240307 | 71.8 | 72.4 | 72.6 | 70.1 | 70.9 | 66.2 | 70.2 | 73.7 | 75.0 | 75.5 | 76.0 | 76.6 | 73.4 | 75.3 | 64.7 | 70.2 | 77.1 |
| claude-3-opus-20240229 | 71.9 | 72.9 | 73.3 | 70.4 | 71.6 | 66.2 | 70.2 | 74.2 | 75.7 | 76.7 | 76.6 | 76.6 | 74.5 | 75.8 | 64.6 | 70.2 | 76.9 |
| deepseek-ai/DeepSeek-V3 | 73.6 | 74.7 | 75.0 | 71.6 | 72.8 | 66.4 | 70.3 | 75.5 | 77.3 | 77.9 | 78.4 | 79.5 | 77.2 | 79.2 | 64.0 | 69.6 | 78.0 |
| Llama-3.1-405B-Instruct-Turbo | 72.8 | 73.3 | 73.9 | 70.1 | 71.3 | 65.6 | 69.3 | 73.1 | 75.3 | 75.9 | 76.1 | 76.8 | 73.9 | 75.9 | 63.9 | 69.5 | 77.1 |
| meta-llama/Llama-3.1-8B-Instruct | 72.3 | 72.7 | 72.8 | 70.0 | 71.0 | 65.4 | 69.0 | 72.7 | 74.3 | 74.5 | 74.2 | 74.9 | 73.4 | 74.5 | 63.4 | 68.6 | 75.1 |
| meta-llama/Llama-3.1-70B-Instruct | 71.7 | 72.5 | 73.0 | 69.8 | 70.8 | 65.1 | 69.0 | 72.4 | 74.4 | 75.0 | 75.1 | 75.6 | 73.3 | 74.9 | 63.4 | 68.8 | 75.5 |
| meta-llama/Llama-3.2-1B-Instruct | 70.1 | 69.6 | 69.3 | 68.1 | 67.8 | 63.8 | 66.3 | 69.1 | 70.2 | 69.9 | 69.6 | 70.4 | 69.5 | 70.0 | 62.1 | 66.6 | 70.9 |
| meta-llama/Llama-3.2-3B-Instruct | 72.6 | 72.6 | 72.6 | 70.2 | 71.0 | 65.1 | 68.6 | 72.0 | 73.5 | 73.7 | 73.4 | 74.1 | 72.5 | 73.5 | 63.2 | 68.7 | 74.6 |
| meta-llama/Llama-3.3-70B-Instruct | 72.6 | 73.4 | 73.9 | 70.1 | 71.6 | 65.6 | 69.0 | 73.4 | 75.2 | 75.9 | 75.8 | 76.3 | 74.1 | 76.0 | 63.6 | 69.0 | 76.7 |
| google/gemma-2-2b-it | 81.0 | 78.1 | 78.0 | 70.4 | 71.8 | 63.4 | 66.4 | 70.7 | 71.1 | 71.0 | 71.2 | 71.6 | 70.6 | 71.6 | 60.9 | 65.7 | 71.8 |
| google/gemma-2-9b-it | 78.1 | 83.0 | 80.4 | 69.9 | 72.2 | 62.8 | 66.3 | 70.8 | 71.9 | 71.9 | 72.4 | 72.7 | 71.6 | 72.7 | 60.7 | 65.5 | 72.8 |
| google/gemma-2-27b-it | 78.0 | 80.4 | 83.7 | 69.9 | 71.8 | 62.8 | 66.2 | 70.8 | 71.6 | 72.1 | 72.2 | 72.5 | 71.2 | 72.5 | 60.6 | 65.5 | 73.0 |
| google/gemma-1.1-2b-it | 70.4 | 69.9 | 69.9 | 80.5 | 73.1 | 64.4 | 66.9 | 70.1 | 70.8 | 69.8 | 70.0 | 70.4 | 70.0 | 70.2 | 62.4 | 66.6 | 70.5 |
| google/gemma-1.1-7b-it | 71.8 | 72.2 | 71.8 | 73.1 | 81.9 | 62.8 | 66.4 | 70.3 | 71.0 | 71.0 | 70.7 | 70.9 | 70.3 | 71.3 | 61.0 | 65.8 | 70.9 |
| Qwen/Qwen2.5-0.5B-Instruct | 63.4 | 62.8 | 62.8 | 64.4 | 62.8 | 67.7 | 66.1 | 67.2 | 66.7 | 66.6 | 66.5 | 66.5 | 65.7 | 66.1 | 62.9 | 65.8 | 66.9 |
| Qwen/Qwen2.5-1.5B-Instruct | 66.4 | 66.3 | 66.2 | 66.9 | 66.4 | 66.1 | 72.9 | 70.6 | 70.8 | 71.0 | 71.0 | 70.8 | 69.6 | 70.3 | 64.4 | 68.7 | 70.8 |
| Qwen/Qwen2.5-3B-Instruct | 70.7 | 70.8 | 70.8 | 70.1 | 70.3 | 67.2 | 70.6 | 79.2 | 75.7 | 76.0 | 75.8 | 75.9 | 74.7 | 75.1 | 64.6 | 69.7 | 74.6 |
| Qwen/Qwen2.5-7B-Instruct | 71.1 | 71.9 | 71.6 | 70.8 | 71.0 | 66.7 | 70.8 | 75.7 | 82.7 | 78.0 | 78.3 | 78.3 | 77.6 | 78.3 | 64.7 | 70.3 | 76.7 |
| Qwen/Qwen2.5-14B-Instruct | 71.0 | 71.9 | 72.1 | 69.8 | 71.0 | 66.6 | 71.0 | 76.0 | 78.0 | 83.7 | 79.5 | 79.2 | 76.1 | 79.2 | 64.8 | 70.4 | 77.7 |
| Qwen/Qwen2.5-32B-Instruct | 71.2 | 72.4 | 72.2 | 70.0 | 70.7 | 66.5 | 71.0 | 75.8 | 78.3 | 79.5 | 85.7 | 79.9 | 76.8 | 78.8 | 64.7 | 70.4 | 78.0 |
| Qwen/Qwen2.5-72B-Instruct | 71.6 | 72.7 | 72.5 | 70.4 | 70.9 | 66.7 | 70.8 | 75.9 | 78.5 | 79.2 | 79.9 | 86.3 | 76.4 | 78.8 | 64.5 | 70.8 | 78.4 |
| Qwen/Qwen2.5-7B-Instruct-1M | 70.6 | 71.6 | 71.2 | 70.0 | 70.3 | 65.7 | 69.6 | 74.7 | 78.3 | 76.1 | 76.8 | 76.4 | 82.5 | 77.7 | 63.8 | 68.9 | 74.5 |
| Qwen/Qwen2.5-14B-Instruct-1M | 71.6 | 72.7 | 72.5 | 70.2 | 71.3 | 66.1 | 70.3 | 75.1 | 77.6 | 79.2 | 78.8 | 78.8 | 77.7 | 84.1 | 64.0 | 69.6 | 76.6 |
| Qwen/Qwen2-0.5B-Instruct | 60.9 | 60.7 | 60.6 | 62.4 | 61.0 | 62.9 | 64.4 | 64.6 | 64.7 | 64.8 | 64.7 | 64.5 | 63.8 | 64.0 | 65.2 | 64.9 | 65.0 |
| Qwen/Qwen2-1.5B-Instruct | 65.7 | 65.5 | 65.5 | 66.6 | 65.8 | 65.8 | 68.7 | 69.7 | 70.3 | 70.4 | 70.4 | 70.6 | 68.9 | 69.6 | 64.9 | 72.0 | 71.1 |
| Qwen/Qwen2-72B-Instruct | 71.8 | 72.8 | 73.0 | 70.5 | 70.9 | 66.9 | 70.8 | 74.6 | 76.7 | 77.7 | 78.0 | 78.0 | 74.5 | 76.6 | 65.0 | 71.1 | 84.8 |
| Qwen/Qwen1.5-0.5B-Chat | 61.9 | 61.5 | 61.6 | 63.5 | 61.9 | 63.8 | 64.6 | 65.3 | 65.6 | 65.5 | 65.4 | 65.2 | 64.5 | 64.7 | 62.9 | 66.0 | 66.4 |
| Qwen/Qwen1.5-1.8B-Chat | 67.9 | 67.3 | 67.3 | 69.1 | 67.5 | 68.5 | 70.0 | 71.4 | 71.4 | 71.3 | 71.3 | 71.3 | 69.8 | 70.7 | 65.6 | 70.6 | 72.6 |
| Qwen/Qwen1.5-4B-Chat | 66.4 | 66.7 | 66.6 | 67.5 | 67.0 | 66.1 | 69.7 | 70.4 | 71.4 | 71.8 | 71.7 | 71.5 | 69.9 | 70.8 | 65.6 | 70.7 | 72.8 |
| Qwen/Qwen1.5-7B-Chat | 70.0 | 70.1 | 70.2 | 70.0 | 69.8 | 67.3 | 71.2 | 73.8 | 74.8 | 75.1 | 74.9 | 74.8 | 73.1 | 74.2 | 65.6 | 70.9 | 75.8 |
| Qwen/Qwen1.5-14B-Chat | 71.4 | 71.7 | 71.7 | 71.1 | 71.2 | 67.8 | 71.8 | 75.2 | 75.9 | 76.5 | 76.3 | 76.4 | 74.3 | 75.5 | 65.8 | 71.7 | 77.0 |
| Qwen/Qwen1.5-32B-Chat | 71.1 | 71.8 | 71.8 | 70.8 | 70.8 | 67.0 | 71.5 | 74.6 | 76.4 | 76.7 | 77.0 | 77.5 | 74.8 | 76.3 | 65.3 | 71.7 | 78.3 |
| Qwen/Qwen1.5-72B-Chat | 71.8 | 72.5 | 72.4 | 71.1 | 71.9 | 67.7 | 72.2 | 75.3 | 77.0 | 77.9 | 78.3 | 79.0 | 75.4 | 77.4 | 65.9 | 72.3 | 79.2 |
| Qwen/Qwen1.5-110B-Chat | 71.6 | 72.3 | 72.5 | 70.9 | 71.3 | 67.7 | 71.9 | 75.1 | 77.0 | 77.9 | 78.5 | 78.8 | 75.4 | 77.3 | 65.7 | 71.7 | 80.0 |
| mistralai/Mistral-Small-24B-Instruct-2501 | 73.3 | 74.1 | 74.4 | 70.5 | 71.3 | 65.8 | 69.7 | 73.9 | 75.4 | 75.8 | 75.9 | 76.3 | 73.9 | 75.4 | 63.7 | 69.0 | 76.8 |
| mistralai/Mistral-7B-Instruct-v0.1 | 68.2 | 68.4 | 68.1 | 68.4 | 68.7 | 66.3 | 69.8 | 71.3 | 72.7 | 73.2 | 73.4 | 73.3 | 71.3 | 72.6 | 65.4 | 70.8 | 74.3 |
| mistralai/Mistral-7B-Instruct-v0.2 | 71.4 | 71.5 | 71.8 | 70.2 | 70.3 | 66.9 | 70.3 | 73.4 | 74.9 | 75.3 | 75.4 | 75.4 | 73.0 | 74.6 | 64.7 | 70.1 | 77.1 |
| mistralai/Mistral-7B-Instruct-v0.3 | 71.2 | 71.5 | 71.6 | 70.5 | 70.5 | 67.2 | 70.4 | 73.9 | 75.1 | 75.4 | 75.7 | 75.6 | 73.8 | 74.9 | 64.8 | 69.7 | 76.4 |
| mistralai/Ministral-8B-Instruct-2410 | 71.2 | 71.6 | 71.7 | 70.2 | 70.9 | 65.6 | 69.3 | 73.0 | 74.4 | 74.3 | 74.4 | 74.7 | 73.1 | 74.1 | 63.7 | 68.8 | 74.6 |
| mistralai/Mistral-Nemo-Instruct-2407 | 71.2 | 72.1 | 72.2 | 69.5 | 70.8 | 65.1 | 68.7 | 72.8 | 74.3 | 74.5 | 74.7 | 75.1 | 73.1 | 74.5 | 63.3 | 68.3 | 74.9 |
| mistralai/Mistral-Small-Instruct-2409 | 71.9 | 72.7 | 72.9 | 69.6 | 70.7 | 65.4 | 68.9 | 73.0 | 74.4 | 74.6 | 75.0 | 75.2 | 72.9 | 74.7 | 63.4 | 68.4 | 75.5 |
| mistralai/Mistral-Large-Instruct-2411 | 73.7 | 74.3 | 74.8 | 70.0 | 71.0 | 65.8 | 68.9 | 73.4 | 74.2 | 74.8 | 74.9 | 75.2 | 72.8 | 74.7 | 63.3 | 68.1 | 75.9 |
| mistralai/Mixtral-8x7B-Instruct-v0.1 | 71.0 | 71.8 | 72.1 | 69.6 | 69.9 | 66.8 | 70.5 | 73.5 | 74.6 | 75.1 | 75.3 | 75.0 | 72.9 | 74.4 | 64.9 | 70.5 | 77.5 |
| microsoft/phi-4 | 73.1 | 73.8 | 73.7 | 72.0 | 72.4 | 67.8 | 71.6 | 76.2 | 78.2 | 78.6 | 79.1 | 79.1 | 77.0 | 78.7 | 65.3 | 70.6 | 78.0 |
| microsoft/Phi-3.5-mini-instruct | 71.6 | 71.4 | 71.6 | 70.8 | 71.0 | 67.6 | 70.6 | 74.7 | 75.5 | 75.7 | 76.1 | 75.8 | 74.0 | 75.3 | 64.8 | 69.7 | 75.9 |
| microsoft/Phi-3-mini-128k-instruct | 69.8 | 70.0 | 70.2 | 69.1 | 69.4 | 66.1 | 69.5 | 72.9 | 73.8 | 74.4 | 74.7 | 74.4 | 72.8 | 74.0 | 64.1 | 68.9 | 74.3 |
| o1-2024-12-17 | 69.8 | 70.8 | 71.2 | 68.3 | 69.7 | 64.2 | 68.0 | 72.5 | 74.7 | 75.2 | 76.2 | 76.1 | 72.5 | 76.1 | 62.5 | 67.7 | 74.7 |
| o1-mini-2024-09-12 | 73.4 | 73.7 | 74.0 | 71.0 | 72.0 | 65.4 | 69.1 | 73.9 | 75.8 | 76.2 | 76.7 | 77.5 | 74.6 | 77.0 | 62.9 | 68.0 | 76.0 |
| o1-preview-2024-09-12 | 71.7 | 72.6 | 72.8 | 70.5 | 71.2 | 65.7 | 69.6 | 74.3 | 75.9 | 76.5 | 77.5 | 78.1 | 75.0 | 77.2 | 63.8 | 68.9 | 76.4 |
| o3-mini-2025-01-31 | 71.0 | 71.8 | 72.3 | 69.8 | 70.7 | 65.2 | 68.9 | 73.8 | 75.5 | 76.3 | 77.2 | 77.2 | 75.0 | 76.7 | 63.3 | 68.6 | 75.9 |
| CohereForAI/aya-expanse-8b | 75.0 | 75.0 | 74.9 | 72.9 | 73.0 | 67.4 | 70.5 | 74.7 | 75.3 | 75.4 | 75.6 | 75.9 | 74.1 | 75.2 | 64.4 | 69.6 | 76.6 |
| CohereForAI/aya-expanse-32b | 74.7 | 75.0 | 75.2 | 72.2 | 72.6 | 66.9 | 70.3 | 74.6 | 75.9 | 76.0 | 76.6 | 76.8 | 74.1 | 76.2 | 64.2 | 69.7 | 77.7 |
| CohereForAI/c4ai-command-r-plus-08-2024 | 69.7 | 70.4 | 70.4 | 69.1 | 69.8 | 65.7 | 69.1 | 72.5 | 74.0 | 74.4 | 74.4 | 74.7 | 72.5 | 73.9 | 64.0 | 69.3 | 75.3 |
| CohereForAI/c4ai-command-r-08-2024 | 70.8 | 71.3 | 71.3 | 70.4 | 70.8 | 66.4 | 70.0 | 73.5 | 74.6 | 75.0 | 75.1 | 75.3 | 73.5 | 74.8 | 64.6 | 69.8 | 75.8 |
| allenai/OLMo-2-1124-13B-Instruct | 72.1 | 72.7 | 72.7 | 70.6 | 71.6 | 66.9 | 70.8 | 75.4 | 76.2 | 76.8 | 77.3 | 77.3 | 75.1 | 76.6 | 64.6 | 70.0 | 76.3 |
| allenai/OLMo-2-1124-7B-Instruct | 72.2 | 72.2 | 72.7 | 70.1 | 71.6 | 67.2 | 70.9 | 75.5 | 76.4 | 76.7 | 77.4 | 77.5 | 75.1 | 76.7 | 64.9 | 70.3 | 76.5 |
| allenai/Llama-3.1-Tulu-3-8B | 72.1 | 72.6 | 72.6 | 70.8 | 71.6 | 66.8 | 71.1 | 75.5 | 76.4 | 77.1 | 77.4 | 77.7 | 75.3 | 76.7 | 64.8 | 70.2 | 76.7 |
| allenai/Llama-3.1-Tulu-3-70B | 72.4 | 72.9 | 73.2 | 70.9 | 71.8 | 67.0 | 70.8 | 75.1 | 76.7 | 77.6 | 78.2 | 78.7 | 75.4 | 77.2 | 64.8 | 70.3 | 77.3 |
| qwen-max-2025-01-25 | 72.8 | 73.9 | 74.2 | 70.5 | 71.9 | 66.5 | 70.5 | 75.8 | 77.6 | 78.1 | 78.9 | 80.2 | 77.1 | 79.0 | 64.2 | 69.7 | 77.4 |
| qwen-plus-2025-01-25 | 72.2 | 73.2 | 73.2 | 70.5 | 71.3 | 66.8 | 70.7 | 75.5 | 77.4 | 78.1 | 79.0 | 81.1 | 76.7 | 79.3 | 64.7 | 70.3 | 77.8 |
| qwen-turbo-2024-11-01 | 70.3 | 71.5 | 71.6 | 69.3 | 70.1 | 66.1 | 70.3 | 74.6 | 76.8 | 77.5 | 78.1 | 78.9 | 75.3 | 76.8 | 64.4 | 70.5 | 77.9 |
| Qwen/Qwen3-0.6B | 68.1 | 67.7 | 67.8 | 68.1 | 68.0 | 66.2 | 68.1 | 71.0 | 70.9 | 70.9 | 70.5 | 71.2 | 70.7 | 70.8 | 64.1 | 68.1 | 70.1 |
| Qwen/Qwen3-1.7B | 70.8 | 70.6 | 70.6 | 70.0 | 69.8 | 66.8 | 69.6 | 73.8 | 73.9 | 73.6 | 73.2 | 73.7 | 73.7 | 74.1 | 64.4 | 68.9 | 72.5 |
| Qwen/Qwen3-4B | 72.8 | 72.7 | 72.8 | 70.6 | 71.4 | 66.4 | 70.1 | 74.4 | 75.3 | 75.6 | 75.4 | 76.5 | 75.2 | 76.3 | 64.2 | 69.5 | 74.8 |
| Qwen/Qwen3-8B | 72.9 | 73.3 | 73.4 | 71.2 | 72.0 | 66.6 | 70.5 | 75.3 | 76.2 | 76.8 | 76.9 | 76.7 | 76.3 | 77.6 | 64.6 | 69.7 | 75.7 |
| Qwen/Qwen3-14B | 72.5 | 73.4 | 73.5 | 70.6 | 71.6 | 65.8 | 69.7 | 74.3 | 75.9 | 76.4 | 76.7 | 77.3 | 75.7 | 77.3 | 63.8 | 68.9 | 75.2 |
| Qwen/Qwen3-32B | 72.4 | 73.4 | 73.2 | 69.4 | 71.0 | 65.1 | 69.0 | 74.2 | 75.2 | 76.4 | 76.4 | 77.0 | 75.3 | 76.7 | 63.2 | 68.4 | 74.7 |
| gemini-1.5-flash | 73.5 | 75.6 | 76.1 | 68.3 | 71.0 | 61.5 | 66.1 | 70.0 | 71.5 | 72.0 | 72.3 | 72.1 | 71.1 | 72.1 | 60.3 | 65.0 | 72.0 |
| gemini-1.5-pro | 71.8 | 73.9 | 74.6 | 67.2 | 70.2 | 60.7 | 64.5 | 68.9 | 70.3 | 71.3 | 70.8 | 71.3 | 70.1 | 71.2 | 59.2 | 63.8 | 71.3 |
| gemini-2.0-flash | 75.3 | 76.5 | 77.1 | 69.1 | 70.9 | 63.0 | 66.3 | 71.0 | 72.2 | 72.7 | 72.6 | 72.8 | 71.7 | 73.0 | 60.9 | 65.5 | 73.2 |
| gemini-2.0-flash-lite-preview-02-05 | 75.9 | 77.3 | 77.5 | 69.1 | 71.2 | 62.9 | 66.4 | 71.1 | 72.2 | 72.5 | 72.3 | 73.0 | 71.7 | 73.1 | 60.9 | 65.6 | 73.3 |

| | Qwen/Qwen1.5-0.5B-Chat | Qwen/Qwen1.5-1.8B-Chat | Qwen/Qwen1.5-4B-Chat | Qwen/Qwen1.5-7B-Chat | Qwen/Qwen1.5-14B-Chat | Qwen/Qwen1.5-32B-Chat | Qwen/Qwen1.5-72B-Chat | Qwen/Qwen1.5-110B-Chat | mistralai/Mistral-Small-24B-Instruct-2501 | mistralai/Mistral-7B-Instruct-v0.1 | mistralai/Mistral-7B-Instruct-v0.2 | mistralai/Mistral-7B-Instruct-v0.3 | mistralai/Ministral-8B-Instruct-2410 | mistralai/Mistral-Nemo-Instruct-2407 | mistralai/Mistral-Small-Instruct-2409 | mistralai/Mistral-Large-Instruct-2411 | mistralai/Mixtral-8x7B-Instruct-v0.1 |
|---|---|---|---|---|---|---|---|---|---|---|---|---|---|---|---|---|---|
| gpt-4o-2024-11-20 | 65.3 | 71.1 | 71.6 | 74.1 | 76.0 | 76.9 | 78.2 | 78.1 | 76.3 | 73.7 | 75.3 | 75.4 | 74.7 | 75.6 | 76.1 | 76.7 | 76.0 |
| gpt-4o-2024-08-06 | 65.8 | 71.9 | 72.3 | 75.1 | 77.2 | 77.6 | 79.0 | 79.3 | 77.0 | 74.6 | 76.5 | 76.6 | 74.9 | 75.4 | 75.7 | 75.9 | 76.2 |
| gpt-4o-2024-05-13 | 65.7 | 71.7 | 72.2 | 75.1 | 77.1 | 77.7 | 79.0 | 79.0 | 77.1 | 74.0 | 76.2 | 76.2 | 75.3 | 75.9 | 76.1 | 76.2 | 76.3 |
| gpt-4o-mini-2024-07-18 | 66.1 | 72.2 | 72.3 | 75.2 | 77.7 | 78.0 | 79.8 | 79.4 | 76.9 | 74.3 | 76.6 | 76.5 | 75.3 | 75.6 | 75.9 | 76.1 | 76.3 |
| gpt-4-turbo-2024-04-09 | 65.0 | 70.9 | 71.3 | 74.3 | 76.5 | 76.9 | 78.2 | 78.6 | 75.9 | 73.1 | 75.3 | 75.6 | 74.2 | 75.0 | 74.9 | 75.2 | 75.6 |
| claude-3-5-sonnet-20241022 | 62.5 | 67.3 | 68.2 | 71.1 | 72.3 | 73.0 | 73.9 | 73.8 | 73.8 | 69.5 | 72.1 | 71.9 | 71.6 | 72.5 | 73.1 | 73.9 | 73.4 |
| claude-3-5-haiku-20241022 | 63.1 | 68.6 | 68.6 | 72.1 | 73.8 | 74.3 | 74.9 | 75.0 | 75.6 | 70.2 | 73.4 | 73.2 | 72.7 | 73.3 | 74.2 | 75.0 | 74.8 |
| claude-3-sonnet-20240229 | 65.3 | 71.5 | 71.6 | 74.6 | 76.3 | 77.4 | 77.7 | 77.7 | 76.0 | 72.9 | 75.5 | 75.3 | 74.1 | 75.1 | 75.4 | 75.4 | 76.3 |
| claude-3-haiku-20240307 | 65.5 | 71.7 | 71.5 | 74.1 | 75.5 | 76.7 | 77.4 | 77.3 | 75.2 | 73.0 | 75.0 | 74.8 | 73.7 | 74.2 | 74.4 | 74.7 | 76.0 |
| claude-3-opus-20240229 | 65.4 | 71.5 | 71.5 | 74.2 | 76.0 | 76.6 | 77.5 | 76.9 | 76.1 | 73.1 | 75.7 | 75.3 | 74.1 | 74.7 | 75.2 | 75.8 | 76.9 |
| deepseek-ai/DeepSeek-V3 | 65.0 | 71.3 | 71.1 | 74.6 | 76.5 | 76.8 | 78.2 | 78.0 | 77.3 | 72.9 | 76.1 | 76.4 | 75.6 | 76.3 | 76.7 | 77.2 | 76.5 |
| Llama-3.1-405B-Instruct-Turbo | 64.7 | 70.7 | 70.8 | 73.7 | 75.1 | 76.1 | 76.6 | 76.8 | 76.3 | 72.5 | 75.2 | 75.4 | 73.9 | 74.4 | 75.3 | 75.8 | 75.3 |
| meta-llama/Llama-3.1-8B-Instruct | 64.2 | 70.2 | 69.9 | 72.9 | 74.3 | 74.6 | 75.0 | 75.2 | 75.3 | 71.3 | 73.8 | 73.6 | 73.3 | 73.4 | 74.0 | 74.4 | 73.9 |
| meta-llama/Llama-3.1-70B-Instruct | 64.0 | 69.9 | 70.1 | 73.0 | 74.3 | 75.2 | 75.5 | 75.7 | 75.2 | 71.7 | 74.3 | 73.8 | 73.1 | 73.5 | 74.0 | 74.6 | 74.6 |
| meta-llama/Llama-3.2-1B-Instruct | 62.9 | 68.3 | 67.3 | 69.8 | 70.5 | 70.4 | 71.0 | 70.7 | 70.9 | 68.0 | 70.1 | 69.6 | 69.8 | 69.7 | 69.9 | 70.4 | 69.9 |
| meta-llama/Llama-3.2-3B-Instruct | 64.1 | 70.0 | 69.6 | 72.8 | 73.9 | 74.1 | 74.5 | 74.3 | 74.5 | 71.0 | 73.5 | 72.9 | 73.0 | 73.0 | 73.2 | 73.8 | 73.5 |
| meta-llama/Llama-3.3-70B-Instruct | 64.5 | 70.3 | 70.3 | 73.5 | 74.7 | 75.9 | 76.1 | 76.2 | 76.0 | 72.2 | 74.9 | 74.1 | 73.6 | 74.1 | 74.9 | 75.5 | 75.4 |
| google/gemma-2-2b-it | 61.9 | 67.9 | 66.4 | 70.0 | 71.4 | 71.1 | 71.8 | 71.6 | 73.3 | 68.2 | 71.4 | 71.2 | 71.2 | 71.2 | 71.5 | 73.7 | 71.0 |
| google/gemma-2-9b-it | 61.5 | 67.3 | 66.7 | 70.1 | 71.7 | 71.8 | 72.5 | 72.3 | 74.1 | 68.4 | 71.5 | 71.5 | 71.6 | 72.1 | 72.7 | 74.3 | 71.8 |
| google/gemma-2-27b-it | 61.6 | 67.3 | 66.6 | 70.2 | 71.7 | 71.8 | 72.4 | 72.5 | 74.4 | 68.1 | 71.8 | 71.6 | 71.7 | 72.2 | 72.9 | 74.8 | 72.1 |
| google/gemma-1.1-2b-it | 63.5 | 69.1 | 67.5 | 70.0 | 71.1 | 70.4 | 71.1 | 70.9 | 70.5 | 68.4 | 70.2 | 70.5 | 70.2 | 69.5 | 69.6 | 70.0 | 69.6 |
| google/gemma-1.1-7b-it | 61.9 | 67.5 | 67.0 | 69.8 | 71.2 | 70.8 | 71.9 | 71.3 | 71.3 | 68.7 | 70.3 | 70.5 | 70.9 | 70.8 | 70.7 | 71.0 | 69.9 |
| Qwen/Qwen2.5-0.5B-Instruct | 63.8 | 68.5 | 66.1 | 67.3 | 67.8 | 67.0 | 67.7 | 67.7 | 65.8 | 66.3 | 66.9 | 67.2 | 65.6 | 65.1 | 65.4 | 65.8 | 66.8 |
| Qwen/Qwen2.5-1.5B-Instruct | 64.6 | 70.0 | 69.7 | 71.2 | 71.8 | 71.5 | 72.2 | 71.9 | 69.7 | 69.8 | 70.3 | 70.4 | 69.3 | 68.7 | 68.9 | 68.9 | 70.5 |
| Qwen/Qwen2.5-3B-Instruct | 65.3 | 71.4 | 70.4 | 73.8 | 75.2 | 74.6 | 75.3 | 75.1 | 73.9 | 71.3 | 73.4 | 73.9 | 73.0 | 72.8 | 73.0 | 73.4 | 73.5 |
| Qwen/Qwen2.5-7B-Instruct | 65.6 | 71.4 | 71.4 | 74.8 | 75.9 | 76.4 | 77.0 | 77.0 | 75.4 | 72.7 | 74.9 | 75.1 | 74.4 | 74.3 | 74.4 | 74.2 | 74.6 |
| Qwen/Qwen2.5-14B-Instruct | 65.5 | 71.3 | 71.8 | 75.1 | 76.5 | 76.7 | 77.9 | 77.9 | 75.8 | 73.2 | 75.3 | 75.4 | 74.3 | 74.5 | 74.5 | 74.8 | 75.1 |
| Qwen/Qwen2.5-32B-Instruct | 65.4 | 71.3 | 71.7 | 74.9 | 76.3 | 77.0 | 78.3 | 78.5 | 75.9 | 73.4 | 75.4 | 75.7 | 74.4 | 74.7 | 75.0 | 74.9 | 75.3 |
| Qwen/Qwen2.5-72B-Instruct | 65.2 | 71.3 | 71.5 | 74.8 | 76.4 | 77.5 | 79.0 | 78.8 | 76.3 | 73.3 | 75.4 | 75.5 | 74.7 | 75.1 | 75.2 | 75.2 | 75.0 |
| Qwen/Qwen2.5-7B-Instruct-1M | 64.5 | 69.8 | 69.9 | 73.1 | 74.3 | 74.8 | 75.4 | 73.9 | 71.3 | 73.0 | 73.8 | 73.1 | 73.1 | 72.9 | 72.8 | 72.7 | 72.9 |
| Qwen/Qwen2.5-14B-Instruct-1M | 64.7 | 70.7 | 70.8 | 74.2 | 75.5 | 76.3 | 77.4 | 77.3 | 75.4 | 72.6 | 74.6 | 74.9 | 74.1 | 74.5 | 74.7 | 74.7 | 74.4 |
| Qwen/Qwen2-0.5B-Instruct | 62.9 | 66.5 | 65.6 | 65.6 | 65.8 | 65.3 | 65.9 | 65.7 | 63.7 | 65.4 | 64.7 | 64.8 | 63.7 | 63.3 | 63.4 | 63.3 | 64.9 |
| Qwen/Qwen2-1.5B-Instruct | 66.0 | 70.7 | 70.7 | 71.7 | 71.7 | 72.3 | 71.7 | 71.9 | 69.0 | 70.8 | 70.1 | 69.7 | 68.8 | 68.4 | 68.1 | 70.5 | |
| Qwen/Qwen2-72B-Instruct | 66.4 | 72.6 | 72.8 | 75.8 | 77.0 | 78.3 | 79.2 | 80.0 | 76.8 | 74.3 | 77.1 | 76.4 | 74.6 | 74.9 | 75.5 | 75.9 | 77.5 |
| Qwen/Qwen1.5-0.5B-Chat | 68.3 | 68.4 | 67.4 | 66.9 | 67.1 | 66.9 | 67.1 | 67.1 | 64.7 | 66.6 | 65.8 | 65.7 | 64.6 | 64.0 | 64.3 | 64.3 | 66.3 |
| Qwen/Qwen1.5-1.8B-Chat | 68.4 | 78.0 | 71.9 | 72.9 | 73.5 | 72.8 | 73.5 | 73.1 | 70.5 | 71.8 | 72.6 | 72.4 | 70.2 | 69.7 | 69.8 | 70.3 | 72.5 |
| Qwen/Qwen1.5-4B-Chat | 67.4 | 71.9 | 76.8 | 72.9 | 73.3 | 73.5 | 74.5 | 73.7 | 70.6 | 73.1 | 71.8 | 71.3 | 70.0 | 69.7 | 70.0 | 69.5 | 72.4 |
| Qwen/Qwen1.5-7B-Chat | 66.9 | 72.9 | 72.9 | 80.7 | 77.7 | 77.1 | 77.0 | 77.4 | 73.6 | 72.9 | 74.9 | 74.7 | 73.0 | 72.9 | 72.8 | 72.9 | 74.8 |
| Qwen/Qwen1.5-14B-Chat | 67.1 | 73.5 | 73.3 | 77.7 | 83.9 | 78.8 | 79.0 | 79.1 | 75.0 | 74.2 | 76.5 | 76.5 | 74.1 | 74.3 | 74.4 | 75.4 | 76.1 |
| Qwen/Qwen1.5-32B-Chat | 66.9 | 72.8 | 73.5 | 77.1 | 78.8 | 85.0 | 80.1 | 79.9 | 75.2 | 74.5 | 76.3 | 75.7 | 74.3 | 74.7 | 74.7 | 74.2 | 76.6 |
| Qwen/Qwen1.5-72B-Chat | 67.1 | 73.5 | 74.5 | 77.0 | 79.0 | 80.1 | 87.0 | 81.6 | 76.0 | 75.5 | 76.9 | 77.1 | 75.1 | 75.4 | 75.6 | 75.2 | 77.2 |
| Qwen/Qwen1.5-110B-Chat | 67.1 | 73.1 | 73.7 | 77.4 | 79.1 | 79.9 | 81.6 | 86.8 | 76.5 | 74.9 | 77.1 | 77.2 | 74.8 | 75.3 | 75.4 | 75.5 | 77.3 |
| mistralai/Mistral-Small-24B-Instruct-2501 | 64.7 | 70.5 | 70.6 | 73.6 | 75.0 | 75.2 | 76.0 | 76.5 | 82.1 | 72.3 | 75.2 | 75.0 | 74.9 | 74.9 | 76.5 | 78.4 | 75.9 |
| mistralai/Mistral-7B-Instruct-v0.1 | 66.6 | 71.8 | 73.1 | 72.9 | 74.2 | 74.5 | 75.5 | 74.9 | 72.3 | 81.8 | 75.5 | 75.0 | 71.3 | 71.1 | 73.4 | 72.5 | 75.4 |
| mistralai/Mistral-7B-Instruct-v0.2 | 65.8 | 72.6 | 71.8 | 74.6 | 76.5 | 76.3 | 77.1 | 75.2 | 75.5 | 74.0 | 84.7 | 79.1 | 73.8 | 74.0 | 75.9 | 75.9 | 78.6 |
| mistralai/Mistral-7B-Instruct-v0.3 | 65.7 | 72.4 | 71.3 | 74.7 | 76.5 | 75.7 | 77.1 | 77.2 | 75.0 | 75.0 | 79.1 | 83.0 | 73.9 | 74.3 | 76.0 | 75.7 | 77.5 |
| mistralai/Ministral-8B-Instruct-2410 | 64.6 | 70.2 | 70.0 | 73.0 | 74.1 | 74.3 | 75.1 | 74.8 | 74.9 | 71.3 | 73.8 | 73.9 | 76.7 | 75.6 | 75.2 | 74.9 | 73.7 |
| mistralai/Mistral-Nemo-Instruct-2407 | 64.0 | 69.7 | 69.7 | 72.9 | 74.3 | 74.7 | 75.4 | 75.3 | 74.9 | 71.1 | 74.0 | 74.3 | 75.6 | 79.7 | 75.8 | 75.3 | 74.1 |
| mistralai/Mistral-Small-Instruct-2409 | 64.3 | 69.8 | 70.0 | 72.8 | 74.6 | 74.7 | 75.6 | 75.4 | 76.5 | 73.4 | 75.9 | 76.0 | 75.2 | 75.8 | 81.4 | 78.8 | 76.2 |
| mistralai/Mistral-Large-Instruct-2411 | 64.3 | 70.3 | 69.5 | 72.9 | 74.3 | 74.2 | 75.2 | 75.5 | 78.4 | 72.5 | 75.9 | 75.7 | 74.9 | 75.3 | 78.8 | 85.6 | 76.9 |
| mistralai/Mixtral-8x7B-Instruct-v0.1 | 66.3 | 72.5 | 72.4 | 74.8 | 76.1 | 76.6 | 77.2 | 77.3 | 75.9 | 75.4 | 78.6 | 77.5 | 73.7 | 74.1 | 76.2 | 76.9 | 87.3 |
| microsoft/phi-4 | 66.0 | 72.9 | 71.9 | 75.6 | 77.2 | 76.9 | 78.1 | 78.3 | 77.5 | 73.4 | 76.9 | 77.2 | 75.6 | 75.6 | 76.2 | 77.1 | 77.0 |
| microsoft/Phi-3.5-mini-instruct | 65.7 | 72.9 | 71.0 | 75.1 | 76.5 | 75.6 | 76.9 | 76.7 | 74.5 | 74.1 | 77.2 | 77.8 | 74.1 | 74.2 | 75.8 | 76.6 | 76.7 |
| microsoft/Phi-3-mini-128k-instruct | 64.9 | 71.0 | 70.3 | 73.5 | 74.9 | 74.2 | 75.5 | 75.3 | 72.8 | 73.6 | 75.3 | 76.0 | 72.3 | 72.5 | 74.1 | 73.7 | 75.2 |
| o1-2024-12-17 | 63.2 | 68.5 | 69.1 | 71.8 | 73.5 | 74.4 | 75.4 | 75.3 | 73.4 | 71.5 | 72.7 | 72.7 | 72.1 | 72.8 | 72.9 | 72.8 | 73.2 |
| o1-mini-2024-09-12 | 63.6 | 70.0 | 69.2 | 72.7 | 74.3 | 74.9 | 75.8 | 76.0 | 76.3 | 71.3 | 74.4 | 74.2 | 74.1 | 74.3 | 74.8 | 76.0 | 74.4 |
| o1-preview-2024-09-12 | 64.3 | 70.4 | 70.5 | 73.3 | 75.1 | 75.8 | 77.0 | 76.9 | 75.8 | 72.7 | 74.7 | 74.5 | 73.9 | 74.4 | 74.8 | 75.5 | 74.7 |
| o3-mini-2025-01-31 | 64.3 | 69.8 | 69.8 | 73.0 | 74.8 | 75.5 | 76.6 | 76.7 | 74.5 | 72.3 | 73.9 | 74.2 | 73.1 | 73.8 | 73.9 | 73.9 | 74.3 |
| CohereForAI/aya-expanse-8b | 65.7 | 72.7 | 70.9 | 75.1 | 76.3 | 75.5 | 76.5 | 76.9 | 72.3 | 76.6 | 76.3 | 75.5 | 75.5 | 75.9 | 77.3 | 76.4 | |
| CohereForAI/aya-expanse-32b | 65.4 | 72.4 | 70.9 | 75.1 | 76.6 | 76.5 | 77.2 | 77.4 | 77.7 | 72.5 | 77.0 | 76.4 | 75.7 | 76.1 | 76.6 | 78.0 | 77.1 |
| CohereForAI/c4ai-command-r-plus-08-2024 | 65.1 | 70.9 | 70.8 | 73.2 | 74.5 | 75.0 | 76.1 | 76.1 | 73.7 | 72.4 | 73.8 | 73.8 | 72.4 | 72.8 | 73.1 | 73.2 | 74.5 |
| CohereForAI/c4ai-command-r-08-2024 | 65.5 | 71.8 | 71.3 | 74.7 | 75.6 | 75.5 | 76.8 | 76.5 | 74.0 | 72.6 | 74.6 | 74.9 | 73.0 | 73.3 | 73.5 | 73.6 | 75.2 |
| allenai/OLMo-2-1124-13B-Instruct | 65.2 | 72.1 | 70.9 | 74.8 | 76.3 | 76.3 | 77.3 | 77.3 | 75.7 | 72.5 | 75.4 | 75.7 | 74.4 | 74.6 | 74.9 | 75.3 | 75.4 |
| allenai/OLMo-2-1124-7B-Instruct | 65.6 | 72.3 | 71.4 | 74.9 | 76.6 | 76.6 | 77.6 | 77.7 | 75.4 | 73.1 | 75.6 | 75.8 | 74.7 | 74.8 | 75.0 | 75.0 | 75.4 |
| allenai/Llama-3.1-Tulu-3-8B | 65.6 | 72.2 | 71.5 | 75.0 | 76.8 | 76.7 | 77.7 | 77.8 | 75.9 | 73.2 | 75.5 | 75.8 | 74.9 | 75.0 | 75.3 | 75.6 | |
| allenai/Llama-3.1-Tulu-3-70B | 65.4 | 72.2 | 71.4 | 74.8 | 76.8 | 76.9 | 78.4 | 78.1 | 76.1 | 73.1 | 76.1 | 76.3 | 75.0 | 75.7 | 75.5 | 75.6 | 75.9 |
| qwen-max-2025-01-25 | 64.8 | 71.2 | 70.9 | 74.7 | 76.3 | 77.0 | 78.1 | 78.4 | 76.8 | 73.1 | 75.2 | 75.6 | 74.9 | 75.5 | 76.0 | 76.4 | 75.6 |
| qwen-plus-2025-01-25 | 65.2 | 71.2 | 71.2 | 74.6 | 76.4 | 77.2 | 78.4 | 78.3 | 76.5 | 73.6 | 75.7 | 75.9 | 74.6 | 75.2 | 76.1 | 76.2 | 75.9 |
| qwen-turbo-2024-11-01 | 65.3 | 71.0 | 71.9 | 74.5 | 75.9 | 76.3 | 78.1 | 78.5 | 75.3 | 73.4 | 74.9 | 74.7 | 73.6 | 73.9 | 74.1 | 74.4 | 75.7 |
| Qwen/Qwen3-0.6B | 65.2 | 69.3 | 68.5 | 69.9 | 70.3 | 70.1 | 70.8 | 70.4 | 70.1 | 69.3 | 70.0 | 69.8 | 70.1 | 69.5 | 69.9 | 70.4 | 70.4 |
| Qwen/Qwen3-1.7B | 65.3 | 70.4 | 69.7 | 71.7 | 72.5 | 72.4 | 73.2 | 72.6 | 72.7 | 70.8 | 72.2 | 72.0 | 72.4 | 71.7 | 72.4 | 73.2 | 72.6 |
| Qwen/Qwen3-4B | 65.3 | 70.5 | 70.4 | 73.0 | 74.3 | 74.6 | 75.4 | 75.0 | 74.9 | 72.1 | 73.5 | 73.8 | 73.9 | 73.6 | 74.5 | 75.0 | 74.5 |
| Qwen/Qwen3-8B | 65.2 | 71.0 | 70.9 | 73.8 | 75.5 | 75.4 | 76.4 | 75.8 | 75.8 | 72.9 | 75.0 | 74.7 | 74.6 | 74.6 | 75.5 | 75.7 | 75.4 |
| Qwen/Qwen3-14B | 64.5 | 70.0 | 70.2 | 73.0 | 74.5 | 74.7 | 75.5 | 75.4 | 75.4 | 72.2 | 73.9 | 74.2 | 74.2 | 74.5 | 75.1 | 75.5 | 74.5 |
| Qwen/Qwen3-32B | 63.6 | 69.2 | 69.3 | 72.3 | 74.1 | 74.4 | 75.1 | 75.0 | 74.8 | 71.3 | 73.0 | 73.3 | 73.7 | 74.2 | 74.7 | 75.1 | 73.6 |
| gemini-1.5-flash | 60.8 | 66.1 | 66.1 | 69.2 | 70.8 | 71.7 | 72.0 | 71.6 | 72.7 | 67.8 | 70.8 | 70.5 | 71.0 | 71.6 | 72.2 | 72.9 | 71.2 |
| gemini-1.5-pro | 59.7 | 64.7 | 65.0 | 67.8 | 69.5 | 71.1 | 70.8 | 71.2 | 66.6 | 69.4 | 69.3 | 69.9 | 69.9 | 70.8 | 70.8 | 71.4 | 70.2 |
| gemini-2.0-flash | 61.7 | 67.2 | 66.7 | 70.0 | 71.2 | 71.5 | 72.5 | 72.3 | 74.6 | 68.5 | 71.8 | 71.4 | 71.7 | 72.4 | 73.4 | 75.6 | 72.6 |
| gemini-2.0-flash-lite-preview-02-05 | 61.7 | 67.2 | 66.7 | 70.0 | 71.2 | 71.9 | 72.7 | 72.5 | 74.9 | 68.5 | 71.5 | 71.4 | 71.9 | 72.6 | 73.9 | 75.9 | 72.5 |

Table 10: Full results of the inter-model repetition analysis in Figure 6 of the main paper (Part 4).

| | microsoft/phi-4 | microsoft/Phi-3.5-mini-instruct | microsoft/Phi-3-mini-128k-instruct | o1-2024-12-17 | o1-mini-2024-09-12 | o1-preview-2024-09-12 | o3-mini-2025-01-31 | CohereForAI/aya-expanse-8b | CohereForAI/aya-expanse-32b | CohereForAI/c4ai-command-r-plus-08-2024 | CohereForAI/c4ai-command-r-08-2024 | allenai/OLMo-2-1124-13B-Instruct | allenai/OLMo-2-1124-7B-Instruct | allenai/Llama-3.1-Tulu-3-8B | allenai/Llama-3.1-Tulu-3-70B | qwen-max-2025-01-25 | qwen-plus-2025-01-25 |
|---|---|---|---|---|---|---|---|---|---|---|---|---|---|---|---|---|---|
| gpt-4o-2024-11-20 | 78.5 | 75.8 | 74.5 | 77.9 | 78.3 | 79.1 | 78.1 | 76.1 | 76.4 | 74.8 | 75.0 | 76.9 | 76.9 | 76.8 | 78.0 | 80.7 | 80.6 |
| gpt-4o-2024-08-06 | 80.1 | 76.9 | 75.5 | 77.6 | 78.4 | 79.5 | 78.4 | 76.3 | 77.0 | 75.6 | 76.3 | 77.8 | 78.1 | 78.1 | 79.1 | 80.4 | 79.8 |
| gpt-4o-2024-05-13 | 80.0 | 76.7 | 75.3 | 77.9 | 78.8 | 79.7 | 78.6 | 76.6 | 77.2 | 75.7 | 76.0 | 78.1 | 78.4 | 78.5 | 79.6 | 80.5 | 80.0 |
| gpt-4o-mini-2024-07-18 | 80.6 | 77.3 | 75.6 | 77.6 | 79.4 | 79.6 | 79.1 | 77.1 | 77.7 | 75.7 | 76.5 | 78.1 | 78.5 | 78.4 | 79.5 | 81.0 | 80.7 |
| gpt-4-turbo-2024-04-09 | 78.9 | 75.9 | 74.7 | 76.2 | 77.3 | 77.9 | 77.2 | 75.6 | 76.8 | 74.8 | 75.4 | 77.0 | 77.8 | 77.5 | 79.1 | 78.9 | 79.2 |
| claude-3-5-sonnet-20241022 | 74.1 | 71.9 | 73.4 | 73.4 | 73.4 | 73.6 | 73.8 | 73.0 | 74.1 | 72.4 | 72.9 | 73.2 | 72.9 | 73.1 | 74.2 | 75.0 | 74.7 |
| claude-3-5-haiku-20241022 | 76.0 | 73.6 | 72.5 | 74.6 | 75.0 | 74.8 | 75.1 | 75.2 | 75.9 | 73.5 | 74.4 | 75.6 | 74.8 | 75.1 | 75.8 | 76.2 | 75.9 |
| claude-3-sonnet-20240229 | 76.6 | 74.8 | 73.3 | 74.6 | 75.5 | 76.3 | 75.6 | 75.9 | 77.4 | 75.4 | 76.0 | 76.4 | 76.2 | 76.3 | 76.8 | 77.3 | 76.9 |
| claude-3-haiku-20240307 | 76.5 | 74.5 | 72.9 | 74.0 | 74.8 | 75.6 | 75.1 | 75.8 | 76.6 | 75.1 | 75.6 | 76.3 | 76.1 | 76.1 | 76.4 | 76.3 | 76.5 |
| claude-3-opus-20240229 | 77.3 | 75.2 | 74.0 | 74.8 | 75.6 | 75.9 | 75.9 | 76.3 | 77.3 | 75.3 | 76.2 | 76.4 | 76.4 | 76.1 | 76.4 | 77.2 | 76.8 |
| deepseek-ai/DeepSeek-V3 | 80.2 | 76.7 | 75.0 | 76.9 | 79.8 | 79.0 | 78.0 | 77.6 | 78.4 | 75.1 | 75.9 | 77.7 | 77.5 | 77.5 | 78.7 | 81.6 | 80.7 |
| Llama-3.1-405B-Instruct-Turbo | 76.8 | 74.6 | 73.1 | 74.6 | 75.9 | 76.1 | 75.9 | 76.4 | 77.7 | 74.5 | 75.0 | 75.6 | 75.6 | 75.6 | 76.7 | 76.9 | 76.6 |
| meta-llama/Llama-3.1-8B-Instruct | 76.1 | 73.8 | 72.1 | 72.4 | 74.4 | 74.4 | 73.3 | 75.9 | 76.4 | 73.0 | 73.8 | 74.5 | 74.3 | 75.0 | 75.1 | 75.1 | 74.7 |
| meta-llama/Llama-3.1-70B-Instruct | 76.3 | 73.8 | 72.4 | 73.4 | 74.9 | 75.3 | 74.2 | 75.5 | 76.6 | 73.7 | 74.2 | 74.8 | 74.4 | 74.9 | 76.1 | 75.9 | 75.6 |
| meta-llama/Llama-3.2-1B-Instruct | 71.3 | 70.1 | 68.0 | 70.4 | 69.7 | 69.0 | 69.0 | 72.4 | 72.2 | 69.1 | 69.8 | 70.2 | 70.5 | 70.4 | 70.6 | 70.4 | 70.2 |
| meta-llama/Llama-3.2-3B-Instruct | 75.0 | 73.2 | 71.7 | 71.7 | 73.9 | 73.2 | 72.6 | 75.4 | 75.6 | 72.4 | 73.1 | 73.8 | 73.7 | 74.1 | 74.2 | 74.2 | 73.8 |
| meta-llama/Llama-3.3-70B-Instruct | 76.6 | 74.5 | 72.9 | 74.2 | 75.6 | 76.0 | 75.4 | 76.1 | 77.5 | 74.5 | 75.0 | 75.4 | 75.1 | 75.5 | 76.7 | 77.1 | 76.6 |
| google/gemma-2-2b-it | 73.1 | 71.6 | 69.8 | 69.8 | 73.4 | 71.7 | 71.0 | 75.0 | 74.7 | 69.7 | 70.8 | 72.1 | 72.2 | 72.1 | 72.1 | 72.8 | 72.2 |
| google/gemma-2-9b-it | 73.8 | 71.4 | 70.0 | 70.8 | 73.7 | 72.6 | 71.8 | 75.0 | 75.0 | 70.4 | 71.3 | 72.7 | 72.2 | 72.6 | 72.9 | 73.9 | 73.2 |
| google/gemma-2-27b-it | 73.7 | 71.6 | 70.2 | 71.2 | 74.0 | 72.8 | 72.3 | 74.9 | 75.2 | 70.6 | 71.3 | 72.7 | 72.4 | 72.6 | 73.2 | 74.1 | 73.2 |
| google/gemma-1.1-2b-it | 72.0 | 70.8 | 69.1 | 68.3 | 71.0 | 70.5 | 69.8 | 72.9 | 72.2 | 69.1 | 70.4 | 70.6 | 71.1 | 70.8 | 70.9 | 71.0 | 70.5 |
| google/gemma-1.1-7b-it | 72.4 | 71.0 | 69.4 | 69.7 | 72.0 | 71.2 | 70.7 | 73.0 | 72.6 | 69.8 | 70.8 | 71.6 | 71.6 | 71.6 | 71.8 | 71.9 | 71.3 |
| Qwen/Qwen2.5-0.5B-Instruct | 67.8 | 67.6 | 66.1 | 64.2 | 65.4 | 65.7 | 65.2 | 67.4 | 66.9 | 65.7 | 66.4 | 66.9 | 67.2 | 66.8 | 67.0 | 66.5 | 66.8 |
| Qwen/Qwen2.5-1.5B-Instruct | 71.6 | 70.6 | 69.5 | 68.0 | 69.1 | 69.6 | 68.9 | 70.5 | 70.3 | 69.1 | 70.0 | 70.8 | 70.9 | 71.1 | 70.8 | 70.4 | 70.7 |
| Qwen/Qwen2.5-3B-Instruct | 76.2 | 74.7 | 72.9 | 72.5 | 73.9 | 74.3 | 73.8 | 74.7 | 74.6 | 72.5 | 73.5 | 75.4 | 75.0 | 75.5 | 75.1 | 75.8 | 75.5 |
| Qwen/Qwen2.5-7B-Instruct | 78.2 | 75.5 | 73.8 | 74.7 | 75.8 | 75.9 | 75.5 | 75.3 | 75.9 | 74.0 | 74.6 | 76.2 | 76.4 | 76.4 | 76.7 | 77.6 | 77.4 |
| Qwen/Qwen2.5-14B-Instruct | 78.6 | 75.7 | 74.4 | 75.2 | 76.2 | 76.5 | 76.3 | 75.4 | 76.0 | 74.4 | 75.0 | 76.8 | 76.7 | 77.1 | 77.6 | 78.1 | 78.1 |
| Qwen/Qwen2.5-32B-Instruct | 79.1 | 76.1 | 74.7 | 76.2 | 76.7 | 77.5 | 77.2 | 75.6 | 76.6 | 74.7 | 75.1 | 77.3 | 77.4 | 77.4 | 78.2 | 78.9 | 79.0 |
| Qwen/Qwen2.5-72B-Instruct | 79.1 | 75.8 | 74.4 | 76.1 | 77.5 | 78.1 | 77.2 | 75.9 | 76.8 | 75.1 | 75.3 | 77.3 | 77.5 | 77.7 | 78.7 | 80.2 | 81.1 |
| Qwen/Qwen2.5-7B-Instruct-1M | 77.0 | 74.0 | 72.8 | 74.7 | 75.0 | 75.0 | 75.0 | 74.1 | 74.1 | 72.5 | 73.5 | 75.1 | 75.1 | 75.3 | 75.4 | 74.1 | 76.7 |
| Qwen/Qwen2.5-14B-Instruct-1M | 78.7 | 75.3 | 74.0 | 76.1 | 77.0 | 77.2 | 76.7 | 75.2 | 76.2 | 73.9 | 74.8 | 76.6 | 76.7 | 76.7 | 77.2 | 79.0 | 79.3 |
| Qwen/Qwen2-0.5B-Instruct | 65.3 | 64.8 | 64.1 | 62.5 | 62.9 | 63.8 | 63.3 | 64.4 | 64.2 | 64.0 | 64.6 | 64.6 | 64.9 | 64.8 | 64.8 | 64.2 | 64.7 |
| Qwen/Qwen2-1.5B-Instruct | 70.6 | 69.7 | 68.9 | 67.7 | 68.0 | 68.9 | 68.6 | 69.6 | 69.7 | 69.3 | 69.8 | 70.0 | 70.3 | 70.2 | 70.2 | 71.0 | 70.7 |
| Qwen/Qwen2-72B-Instruct | 78.0 | 75.9 | 74.3 | 74.7 | 76.0 | 76.4 | 75.9 | 76.6 | 77.7 | 75.3 | 75.8 | 76.3 | 76.5 | 76.7 | 77.3 | 77.4 | 77.8 |
| Qwen/Qwen1.5-0.5B-Chat | 66.0 | 65.7 | 64.9 | 63.2 | 63.6 | 64.3 | 64.3 | 65.7 | 65.4 | 65.1 | 65.5 | 65.2 | 65.6 | 65.6 | 65.4 | 64.8 | 65.2 |
| Qwen/Qwen1.5-1.8B-Chat | 72.9 | 72.9 | 71.0 | 68.5 | 70.0 | 70.4 | 69.8 | 72.7 | 72.4 | 70.9 | 71.8 | 72.1 | 72.3 | 72.2 | 72.2 | 71.2 | 71.2 |
| Qwen/Qwen1.5-4B-Chat | 71.9 | 71.0 | 70.3 | 69.1 | 69.2 | 70.5 | 69.8 | 70.9 | 70.9 | 70.8 | 71.3 | 70.9 | 71.4 | 71.5 | 71.4 | 70.9 | 71.2 |
| Qwen/Qwen1.5-7B-Chat | 75.6 | 75.1 | 73.5 | 71.8 | 72.7 | 73.3 | 73.0 | 75.1 | 75.1 | 73.2 | 74.2 | 74.8 | 74.9 | 75.0 | 74.8 | 74.7 | 74.6 |
| Qwen/Qwen1.5-14B-Chat | 77.2 | 76.5 | 74.9 | 73.5 | 74.3 | 75.1 | 74.8 | 76.3 | 76.6 | 74.5 | 75.6 | 76.3 | 76.6 | 76.8 | 76.8 | 76.5 | 76.4 |
| Qwen/Qwen1.5-32B-Chat | 76.9 | 75.6 | 74.2 | 74.4 | 74.9 | 75.8 | 75.5 | 75.5 | 76.5 | 75.0 | 75.6 | 76.3 | 76.6 | 76.7 | 76.9 | 77.0 | 77.2 |
| Qwen/Qwen1.5-72B-Chat | 78.1 | 76.9 | 75.5 | 75.4 | 75.8 | 77.0 | 76.6 | 76.5 | 77.2 | 76.1 | 76.5 | 77.3 | 77.6 | 77.7 | 78.2 | 78.1 | 78.4 |
| Qwen/Qwen1.5-110B-Chat | 78.3 | 76.7 | 75.3 | 75.3 | 76.0 | 76.9 | 76.7 | 76.5 | 77.4 | 76.1 | 76.8 | 77.3 | 77.7 | 77.8 | 78.1 | 78.4 | 78.3 |
| mistralai/Mistral-Small-24B-Instruct-2501 | 77.5 | 74.5 | 72.8 | 73.4 | 76.3 | 75.8 | 74.5 | 76.9 | 77.7 | 73.7 | 74.0 | 75.7 | 75.4 | 75.9 | 76.1 | 76.8 | 76.5 |
| mistralai/Mistral-7B-Instruct-v0.1 | 73.4 | 74.1 | 73.6 | 71.5 | 71.3 | 72.7 | 72.3 | 72.3 | 72.5 | 72.4 | 72.6 | 72.5 | 73.1 | 73.2 | 73.1 | 73.1 | 73.6 |
| mistralai/Mistral-7B-Instruct-v0.2 | 76.9 | 77.2 | 75.3 | 72.7 | 74.4 | 74.7 | 73.9 | 76.6 | 77.0 | 73.8 | 74.6 | 75.4 | 75.6 | 75.5 | 76.1 | 75.2 | 75.7 |
| mistralai/Mistral-7B-Instruct-v0.3 | 77.2 | 77.8 | 76.0 | 72.7 | 74.2 | 74.5 | 74.2 | 76.3 | 76.4 | 73.8 | 74.9 | 75.7 | 75.8 | 75.8 | 76.3 | 75.6 | 75.9 |
| mistralai/Ministral-8B-Instruct-2410 | 75.6 | 74.1 | 72.3 | 72.1 | 74.1 | 73.9 | 73.1 | 75.5 | 75.7 | 72.4 | 73.0 | 74.4 | 74.7 | 74.9 | 75.0 | 74.9 | 74.6 |
| mistralai/Mistral-Nemo-Instruct-2407 | 75.6 | 74.2 | 72.5 | 72.8 | 74.3 | 74.4 | 73.8 | 75.5 | 76.1 | 72.8 | 73.3 | 74.6 | 74.8 | 75.0 | 75.7 | 75.5 | 75.2 |
| mistralai/Mistral-Small-Instruct-2409 | 76.2 | 75.8 | 74.1 | 72.9 | 74.8 | 74.8 | 73.9 | 75.9 | 76.6 | 73.1 | 73.5 | 74.9 | 75.0 | 75.3 | 75.5 | 76.0 | 76.1 |
| mistralai/Mistral-Large-Instruct-2411 | 77.1 | 75.8 | 73.7 | 72.8 | 76.0 | 75.5 | 73.9 | 77.3 | 78.0 | 73.2 | 73.6 | 75.3 | 75.0 | 75.2 | 75.6 | 76.4 | 76.2 |
| mistralai/Mixtral-8x7B-Instruct-v0.1 | 77.0 | 76.7 | 75.2 | 73.2 | 74.4 | 74.7 | 73.6 | 76.4 | 77.1 | 74.5 | 75.2 | 75.4 | 75.2 | 75.6 | 75.9 | 76.3 | 75.9 |
| microsoft/phi-4 | 84.5 | 77.9 | 75.6 | 75.6 | 78.9 | 77.9 | 77.0 | 78.5 | 78.9 | 75.1 | 76.4 | 78.4 | 78.2 | 78.2 | 78.8 | 79.5 | 79.3 |
| microsoft/Phi-3.5-mini-instruct | 77.9 | 82.8 | 78.2 | 73.1 | 75.3 | 74.7 | 74.9 | 76.9 | 76.9 | 73.9 | 74.9 | 76.5 | 76.5 | 76.2 | 76.6 | 76.1 | 76.1 |
| microsoft/Phi-3-mini-128k-instruct | 75.6 | 78.2 | 78.1 | 72.3 | 73.1 | 73.0 | 73.6 | 74.4 | 74.2 | 72.7 | 73.5 | 74.5 | 74.5 | 74.5 | 74.8 | 74.5 | 74.5 |
| o1-2024-12-17 | 75.6 | 73.1 | 72.3 | 81.2 | 75.2 | 76.6 | 78.5 | 72.5 | 73.6 | 73.1 | 73.1 | 74.4 | 74.6 | 74.8 | 75.8 | 77.4 | 76.7 |
| o1-mini-2024-09-12 | 78.9 | 75.3 | 73.1 | 75.2 | 85.8 | 80.2 | 76.9 | 76.8 | 77.3 | 73.3 | 73.8 | 76.1 | 76.1 | 75.9 | 76.9 | 78.9 | 78.2 |
| o1-preview-2024-09-12 | 77.9 | 74.7 | 73.0 | 76.0 | 80.2 | 84.7 | 77.2 | 75.4 | 76.5 | 74.4 | 74.4 | 76.4 | 76.5 | 76.5 | 76.5 | 79.0 | 78.6 |
| o3-mini-2025-01-31 | 77.0 | 74.9 | 73.6 | 78.5 | 76.9 | 77.2 | 83.0 | 74.4 | 75.2 | 74.2 | 74.7 | 75.5 | 75.9 | 75.9 | 76.8 | 78.3 | 77.8 |
| CohereForAI/aya-expanse-8b | 78.5 | 76.9 | 74.0 | 72.5 | 76.8 | 75.4 | 74.4 | 85.9 | 80.9 | 74.2 | 75.6 | 76.6 | 76.6 | 76.5 | 76.9 | 76.5 | 76.2 |
| CohereForAI/aya-expanse-32b | 78.9 | 76.9 | 74.2 | 73.6 | 77.3 | 76.5 | 75.2 | 80.9 | 87.9 | 74.8 | 76.5 | 77.3 | 77.3 | 76.9 | 77.6 | 77.5 | 77.7 |
| CohereForAI/c4ai-command-r-plus-08-2024 | 75.1 | 73.9 | 72.7 | 73.1 | 73.3 | 74.4 | 74.2 | 74.2 | 74.8 | 77.6 | 75.1 | 74.5 | 74.6 | 74.8 | 75.3 | 75.0 | 75.0 |
| CohereForAI/c4ai-command-r-08-2024 | 76.4 | 74.9 | 73.5 | 73.1 | 73.8 | 74.4 | 74.7 | 75.6 | 76.5 | 75.1 | 79.7 | 75.1 | 75.2 | 75.5 | 75.8 | 75.4 | 75.5 |
| allenai/OLMo-2-1124-13B-Instruct | 78.4 | 76.5 | 74.5 | 74.4 | 76.1 | 76.4 | 75.6 | 76.6 | 77.3 | 74.5 | 75.1 | 81.0 | 78.3 | 78.1 | 78.2 | 77.6 | 77.5 |
| allenai/OLMo-2-1124-7B-Instruct | 78.2 | 76.5 | 74.5 | 74.6 | 76.1 | 76.5 | 75.9 | 76.6 | 77.3 | 74.6 | 75.2 | 78.3 | 81.1 | 78.0 | 78.3 | 77.7 | 77.5 |
| allenai/Llama-3.1-Tulu-3-8B | 78.2 | 76.2 | 74.5 | 74.8 | 75.9 | 76.5 | 75.9 | 76.5 | 76.9 | 74.8 | 75.5 | 78.1 | 78.0 | 81.0 | 78.7 | 77.6 | 77.4 |
| allenai/Llama-3.1-Tulu-3-70B | 78.8 | 76.4 | 75.8 | 74.6 | 76.9 | 77.5 | 76.6 | 76.9 | 77.6 | 75.3 | 75.8 | 78.2 | 78.3 | 78.7 | 81.8 | 78.6 | 78.5 |
| qwen-max-2025-01-25 | 79.5 | 76.1 | 74.5 | 77.4 | 78.9 | 79.0 | 78.3 | 76.5 | 77.5 | 75.4 | 75.4 | 77.6 | 77.7 | 77.6 | 78.6 | 89.7 | 82.1 |
| qwen-plus-2025-01-25 | 79.3 | 76.1 | 74.5 | 76.7 | 78.2 | 78.6 | 77.8 | 76.2 | 77.7 | 75.0 | 75.5 | 77.5 | 77.5 | 77.4 | 78.5 | 82.1 | 88.4 |
| qwen-turbo-2024-11-01 | 77.3 | 74.6 | 73.6 | 75.0 | 75.5 | 76.8 | 75.9 | 74.7 | 75.5 | 74.2 | 75.1 | 76.0 | 76.3 | 76.9 | 78.3 | 78.8 | 78.8 |
| Qwen/Qwen3-0.6B | 71.8 | 70.9 | 69.4 | 69.0 | 71.0 | 70.6 | 70.2 | 71.8 | 71.0 | 68.8 | 69.4 | 70.6 | 70.8 | 70.7 | 70.8 | 72.2 | 72.3 |
| Qwen/Qwen3-1.7B | 74.8 | 73.2 | 71.4 | 71.3 | 73.9 | 73.6 | 72.5 | 74.3 | 73.7 | 70.7 | 71.5 | 73.4 | 73.3 | 73.5 | 73.4 | 75.3 | 75.1 |
| Qwen/Qwen3-4B | 76.7 | 74.5 | 72.9 | 73.5 | 75.9 | 75.4 | 74.8 | 75.9 | 75.4 | 72.7 | 73.4 | 74.9 | 75.4 | 75.4 | 75.7 | 77.9 | 77.5 |
| Qwen/Qwen3-8B | 77.6 | 75.5 | 73.8 | 75.1 | 77.1 | 77.1 | 76.3 | 76.1 | 76.2 | 73.6 | 74.4 | 76.0 | 75.8 | 76.4 | 76.6 | 79.1 | 78.2 |
| Qwen/Qwen3-14B | 77.4 | 74.6 | 73.3 | 75.0 | 76.8 | 76.5 | 76.1 | 76.0 | 75.8 | 73.5 | 74.3 | 75.5 | 75.3 | 76.1 | 76.7 | 79.1 | 78.7 |
| Qwen/Qwen3-32B | 76.6 | 73.9 | 72.5 | 74.5 | 76.2 | 75.9 | 75.1 | 75.2 | 75.2 | 72.5 | 73.1 | 74.8 | 75.5 | 74.8 | 75.5 | 78.8 | 78.3 |
| gemini-1.5-flash | 72.8 | 70.9 | 69.6 | 71.6 | 72.7 | 72.1 | 72.5 | 72.7 | 73.8 | 70.1 | 70.3 | 72.2 | 71.9 | 72.3 | 72.7 | 73.5 | 72.9 |
| gemini-1.5-pro | 71.6 | 69.5 | 68.4 | 71.2 | 71.5 | 71.1 | 71.9 | 70.8 | 71.9 | 69.5 | 69.8 | 70.7 | 70.2 | 71.0 | 71.4 | 72.3 | 71.7 |
| gemini-2.0-flash | 74.1 | 71.9 | 70.3 | 71.7 | 74.1 | 73.1 | 72.7 | 74.3 | 75.1 | 70.4 | 71.3 | 72.7 | 72.3 | 72.6 | 73.2 | 74.3 | 73.7 |
| gemini-2.0-flash-lite-preview-02-05 | 73.8 | 72.0 | 70.5 | 71.5 | 74.0 | 72.9 | 72.4 | 74.5 | 75.3 | 70.6 | 71.5 | 73.0 | 72.6 | 72.8 | 72.9 | 74.7 | 73.9 |

Table 11: Full results of the inter-model repetition analysis in Figure 6 of the main paper (Part 5).

| | qwen-turbo-2024-11-01 | Qwen/Qwen3-0.6B | Qwen/Qwen3-1.7B | Qwen/Qwen3-4B | Qwen/Qwen3-8B | Qwen/Qwen3-14B | Qwen/Qwen3-32B | gemini-1.5-flash | gemini-1.5-pro | gemini-2.0-flash | gemini-2.0-flash-lite-preview-02-05 |
|---|---|---|---|---|---|---|---|---|---|---|---|
| gpt-4o-2024-11-20 | 77.5 | 72.8 | 75.4 | 77.5 | 78.8 | 78.4 | 78.3 | 72.8 | 72.4 | 74.1 | 74.3 |
| gpt-4o-2024-08-06 | 78.2 | 71.4 | 73.9 | 76.5 | 78.1 | 77.6 | 77.0 | 72.5 | 72.0 | 73.7 | 73.3 |
| gpt-4o-2024-05-13 | 78.4 | 71.2 | 73.9 | 76.5 | 77.8 | 77.8 | 77.1 | 73.1 | 72.5 | 73.9 | 73.7 |
| gpt-4o-mini-2024-07-18 | 78.5 | 71.9 | 74.9 | 77.2 | 78.8 | 78.2 | 77.4 | 73.1 | 72.6 | 74.4 | 73.9 |
| gpt-4-turbo-2024-04-09 | 77.5 | 70.6 | 72.8 | 75.6 | 76.4 | 76.8 | 76.2 | 71.9 | 71.9 | 73.0 | 72.4 |
| claude-3-5-sonnet-20241022 | 73.2 | 67.4 | 69.9 | 72.2 | 73.5 | 73.7 | 73.4 | 70.9 | 71.3 | 72.4 | 72.7 |
| claude-3-5-haiku-20241022 | 74.4 | 68.4 | 71.4 | 73.8 | 74.9 | 75.1 | 74.6 | 73.2 | 73.0 | 73.8 | 74.4 |
| claude-3-sonnet-20240229 | 75.9 | 69.4 | 72.1 | 74.5 | 75.2 | 75.1 | 74.5 | 73.0 | 72.1 | 73.0 | 73.2 |
| claude-3-haiku-20240307 | 76.1 | 70.0 | 72.3 | 74.2 | 75.1 | 74.5 | 73.9 | 72.4 | 70.9 | 72.5 | 72.8 |
| claude-3-opus-20240229 | 75.9 | 70.2 | 72.8 | 74.9 | 76.4 | 76.2 | 75.2 | 72.7 | 72.3 | 73.7 | 73.5 |
| deepseek-ai/DeepSeek-V3 | 77.6 | 72.3 | 75.3 | 77.5 | 79.2 | 79.3 | 78.6 | 73.5 | 73.4 | 75.4 | 75.4 |
| Llama-3.1-405B-Instruct-Turbo | 75.8 | 69.3 | 71.8 | 74.2 | 75.2 | 75.1 | 74.7 | 72.9 | 72.1 | 74.1 | 74.3 |
| meta-llama/Llama-3.1-8B-Instruct | 74.0 | 69.0 | 72.1 | 73.9 | 74.5 | 74.7 | 73.9 | 71.5 | 70.9 | 72.8 | 73.1 |
| meta-llama/Llama-3.1-70B-Instruct | 74.9 | 69.1 | 71.5 | 74.0 | 74.8 | 75.2 | 74.1 | 72.2 | 71.7 | 72.9 | 73.0 |
| meta-llama/Llama-3.2-1B-Instruct | 69.2 | 67.2 | 68.9 | 69.9 | 70.2 | 69.9 | 69.4 | 68.0 | 66.8 | 69.3 | 69.7 |
| meta-llama/Llama-3.2-3B-Instruct | 73.2 | 68.9 | 71.4 | 73.2 | 73.9 | 73.8 | 73.2 | 71.2 | 70.3 | 72.2 | 72.7 |
| meta-llama/Llama-3.3-70B-Instruct | 75.6 | 70.3 | 72.8 | 75.3 | 76.6 | 76.3 | 75.5 | 72.8 | 72.4 | 74.1 | 74.4 |
| google/gemma-2-2b-it | 70.3 | 68.1 | 70.8 | 72.8 | 72.9 | 72.5 | 72.4 | 73.5 | 71.8 | 75.3 | 75.9 |
| google/gemma-2-9b-it | 71.5 | 67.7 | 70.6 | 72.7 | 73.3 | 73.4 | 73.4 | 75.6 | 73.9 | 76.5 | 77.3 |
| google/gemma-2-27b-it | 71.6 | 67.8 | 70.6 | 72.8 | 73.4 | 73.5 | 73.2 | 76.1 | 74.6 | 77.1 | 77.5 |
| google/gemma-1.1-2b-it | 69.3 | 68.1 | 70.0 | 70.6 | 71.2 | 70.6 | 69.4 | 68.3 | 67.2 | 69.1 | 69.1 |
| google/gemma-1.1-7b-it | 70.1 | 68.0 | 69.8 | 71.4 | 72.0 | 71.6 | 71.0 | 71.0 | 70.2 | 70.9 | 71.2 |
| Qwen/Qwen2.5-0.5B-Instruct | 66.1 | 66.2 | 66.8 | 66.4 | 66.6 | 65.8 | 65.1 | 61.9 | 60.7 | 63.0 | 62.9 |
| Qwen/Qwen2.5-1.5B-Instruct | 70.3 | 68.1 | 69.6 | 70.1 | 70.5 | 69.7 | 69.0 | 66.1 | 64.5 | 66.3 | 66.4 |
| Qwen/Qwen2.5-3B-Instruct | 74.6 | 71.0 | 73.8 | 74.4 | 75.3 | 74.3 | 74.2 | 70.0 | 68.9 | 71.0 | 71.1 |
| Qwen/Qwen2.5-7B-Instruct | 76.8 | 70.9 | 73.9 | 75.3 | 76.2 | 75.9 | 75.2 | 71.5 | 70.3 | 72.2 | 72.2 |
| Qwen/Qwen2.5-14B-Instruct | 77.5 | 70.9 | 73.6 | 75.6 | 76.8 | 76.4 | 76.4 | 72.0 | 71.3 | 72.7 | 72.5 |
| Qwen/Qwen2.5-32B-Instruct | 78.1 | 70.5 | 73.2 | 75.4 | 76.9 | 76.7 | 76.4 | 72.3 | 70.8 | 72.6 | 72.3 |
| Qwen/Qwen2.5-72B-Instruct | 78.9 | 71.2 | 73.7 | 76.5 | 77.3 | 77.3 | 77.0 | 72.1 | 71.3 | 72.8 | 73.0 |
| Qwen/Qwen2.5-7B-Instruct-1M | 75.3 | 70.7 | 73.7 | 75.2 | 76.3 | 75.7 | 75.3 | 71.1 | 70.1 | 71.7 | 71.7 |
| Qwen/Qwen2.5-14B-Instruct-1M | 76.8 | 70.8 | 74.1 | 76.3 | 77.6 | 77.3 | 76.7 | 72.1 | 71.2 | 73.0 | 73.1 |
| Qwen/Qwen2-0.5B-Instruct | 64.4 | 64.1 | 64.4 | 64.2 | 64.6 | 63.8 | 63.2 | 60.3 | 59.2 | 60.9 | 60.9 |
| Qwen/Qwen2-1.5B-Instruct | 70.5 | 68.1 | 68.9 | 69.5 | 69.7 | 68.9 | 68.4 | 65.0 | 63.8 | 65.5 | 65.6 |
| Qwen/Qwen2-72B-Instruct | 77.9 | 70.1 | 72.5 | 74.8 | 75.7 | 75.2 | 74.7 | 72.0 | 71.3 | 73.2 | 73.3 |
| Qwen/Qwen1.5-0.5B-Chat | 65.3 | 65.2 | 65.3 | 65.3 | 65.2 | 64.5 | 63.6 | 60.8 | 59.7 | 61.7 | 61.7 |
| Qwen/Qwen1.5-1.8B-Chat | 71.0 | 69.3 | 70.4 | 70.5 | 71.0 | 70.0 | 69.2 | 66.1 | 64.7 | 67.2 | 67.2 |
| Qwen/Qwen1.5-4B-Chat | 71.9 | 68.5 | 69.7 | 70.4 | 70.9 | 70.2 | 69.3 | 66.1 | 65.0 | 66.7 | 66.7 |
| Qwen/Qwen1.5-7B-Chat | 74.5 | 69.9 | 71.7 | 73.0 | 73.8 | 72.3 | 72.9 | 67.8 | 66.9 | 69.8 | 70.0 |
| Qwen/Qwen1.5-14B-Chat | 75.9 | 70.3 | 72.5 | 74.3 | 75.5 | 74.5 | 74.1 | 70.8 | 69.5 | 71.2 | 71.2 |
| Qwen/Qwen1.5-32B-Chat | 77.3 | 70.1 | 72.4 | 74.6 | 75.4 | 74.7 | 74.4 | 71.7 | 71.1 | 71.5 | 71.9 |
| Qwen/Qwen1.5-72B-Chat | 78.1 | 70.8 | 73.2 | 75.4 | 76.4 | 75.7 | 75.1 | 72.0 | 70.9 | 72.5 | 72.7 |
| Qwen/Qwen1.5-110B-Chat | 78.5 | 70.4 | 72.6 | 75.0 | 75.8 | 75.5 | 75.0 | 71.6 | 70.8 | 72.3 | 72.5 |
| mistralai/Mistral-Small-24B-Instruct-2501 | 75.3 | 70.1 | 72.7 | 74.9 | 75.8 | 75.4 | 74.8 | 72.7 | 71.2 | 74.6 | 74.9 |
| mistralai/Mistral-7B-Instruct-v0.1 | 73.4 | 69.3 | 70.8 | 72.1 | 72.9 | 72.2 | 71.3 | 67.8 | 66.6 | 68.5 | 68.5 |
| mistralai/Mistral-7B-Instruct-v0.2 | 74.9 | 70.0 | 72.2 | 73.5 | 75.0 | 73.9 | 73.0 | 70.8 | 69.4 | 71.8 | 71.5 |
| mistralai/Mistral-7B-Instruct-v0.3 | 74.7 | 69.8 | 72.0 | 73.8 | 74.7 | 74.2 | 73.3 | 70.5 | 69.3 | 71.4 | 71.4 |
| mistralai/Ministral-8B-Instruct-2410 | 73.6 | 70.1 | 72.4 | 73.9 | 74.6 | 74.2 | 73.7 | 71.0 | 69.9 | 71.7 | 71.9 |
| mistralai/Mistral-Nemo-Instruct-2407 | 73.9 | 69.5 | 71.7 | 73.6 | 74.6 | 74.5 | 74.2 | 71.6 | 70.8 | 72.4 | 72.6 |
| mistralai/Mistral-Small-Instruct-2409 | 74.1 | 69.9 | 72.4 | 74.5 | 75.5 | 75.1 | 74.7 | 72.2 | 70.8 | 73.4 | 73.9 |
| mistralai/Mistral-Large-Instruct-2411 | 74.4 | 70.4 | 73.2 | 75.0 | 75.7 | 75.5 | 75.1 | 72.9 | 71.4 | 75.6 | 75.9 |
| mistralai/Mixtral-8x7B-Instruct-v0.1 | 75.7 | 70.4 | 72.6 | 74.5 | 75.4 | 74.5 | 73.6 | 71.2 | 70.2 | 72.6 | 72.5 |
| microsoft/phi-4 | 77.3 | 71.8 | 74.8 | 76.7 | 77.6 | 77.4 | 76.6 | 72.8 | 71.6 | 74.1 | 73.8 |
| microsoft/Phi-3.5-mini-instruct | 74.6 | 70.9 | 73.2 | 74.5 | 75.5 | 74.6 | 73.9 | 70.9 | 69.5 | 71.9 | 71.9 |
| microsoft/Phi-3-mini-128k-instruct | 73.6 | 69.4 | 71.4 | 72.9 | 73.8 | 73.3 | 72.5 | 69.6 | 68.4 | 70.3 | 70.5 |
| o1-2024-12-17 | 75.0 | 69.0 | 71.3 | 73.5 | 75.1 | 75.0 | 74.5 | 71.6 | 71.2 | 71.7 | 71.5 |
| o1-mini-2024-09-12 | 75.5 | 71.0 | 73.9 | 75.9 | 77.1 | 76.8 | 76.2 | 72.7 | 71.5 | 74.1 | 74.0 |
| o1-preview-2024-09-12 | 76.8 | 70.6 | 73.6 | 75.4 | 77.1 | 76.5 | 75.9 | 72.1 | 71.1 | 73.1 | 72.9 |
| o3-mini-2025-01-31 | 75.9 | 70.2 | 72.5 | 74.8 | 76.3 | 76.1 | 75.4 | 72.5 | 71.9 | 72.7 | 72.4 |
| CohereForAI/aya-expanse-8b | 74.7 | 71.8 | 74.3 | 75.9 | 76.1 | 76.0 | 75.1 | 72.7 | 70.8 | 74.3 | 74.5 |
| CohereForAI/aya-expanse-32b | 75.5 | 71.0 | 73.7 | 75.4 | 76.2 | 75.8 | 75.2 | 73.8 | 71.9 | 75.1 | 75.3 |
| CohereForAI/c4ai-command-r-plus-08-2024 | 74.2 | 68.8 | 70.7 | 72.7 | 73.6 | 73.5 | 72.5 | 70.1 | 69.5 | 70.4 | 70.6 |
| CohereForAI/c4ai-command-r-08-2024 | 74.7 | 69.4 | 71.5 | 73.7 | 74.4 | 74.3 | 73.3 | 70.3 | 69.8 | 71.3 | 71.5 |
| allenai/OLMo-2-1124-13B-Instruct | 75.8 | 70.6 | 73.4 | 74.9 | 76.0 | 75.5 | 75.1 | 72.2 | 70.7 | 72.7 | 73.0 |
| allenai/OLMo-2-1124-7B-Instruct | 76.0 | 70.8 | 73.3 | 74.9 | 75.8 | 75.3 | 74.8 | 71.9 | 70.2 | 72.3 | 72.6 |
| allenai/Llama-3.1-Tulu-3-8B | 76.3 | 70.7 | 73.5 | 75.4 | 76.4 | 76.1 | 75.5 | 72.3 | 71.0 | 72.6 | 72.8 |
| allenai/Llama-3.1-Tulu-3-70B | 76.8 | 70.8 | 73.4 | 75.5 | 76.6 | 76.7 | 76.1 | 72.7 | 71.4 | 73.2 | 72.9 |
| qwen-max-2025-01-25 | 78.3 | 72.2 | 75.3 | 77.9 | 79.1 | 79.1 | 78.8 | 73.5 | 72.3 | 74.3 | 74.7 |
| qwen-plus-2025-01-25 | 78.8 | 72.3 | 75.1 | 77.5 | 78.2 | 78.7 | 78.3 | 72.9 | 71.7 | 73.7 | 73.9 |
| qwen-turbo-2024-11-01 | 84.7 | 70.7 | 73.2 | 75.4 | 75.7 | 75.9 | 75.2 | 71.3 | 70.4 | 71.5 | 71.8 |
| Qwen/Qwen3-0.6B | 70.7 | 79.2 | 75.6 | 74.8 | 74.3 | 73.7 | 73.0 | 66.9 | 65.4 | 68.3 | 68.6 |
| Qwen/Qwen3-1.7B | 73.2 | 75.6 | 85.2 | 78.9 | 78.7 | 77.7 | 76.6 | 69.4 | 67.7 | 71.3 | 71.5 |
| Qwen/Qwen3-4B | 75.4 | 74.8 | 78.9 | 87.3 | 81.6 | 80.7 | 79.6 | 71.6 | 70.3 | 73.7 | 73.8 |
| Qwen/Qwen3-8B | 75.7 | 74.3 | 78.7 | 81.6 | 87.9 | 81.9 | 80.4 | 71.9 | 71.0 | 74.5 | 74.3 |
| Qwen/Qwen3-14B | 75.9 | 73.7 | 77.7 | 80.7 | 81.9 | 87.2 | 81.5 | 71.9 | 71.2 | 74.5 | 74.4 |
| Qwen/Qwen3-32B | 75.2 | 73.0 | 76.6 | 79.6 | 80.4 | 81.5 | 84.8 | 71.6 | 71.0 | 74.2 | 74.2 |
| gemini-1.5-flash | 71.3 | 66.9 | 69.4 | 71.6 | 71.9 | 71.9 | 71.6 | 89.2 | 77.7 | 76.9 | 77.5 |
| gemini-1.5-pro | 70.4 | 65.4 | 67.7 | 70.3 | 71.0 | 71.2 | 71.0 | 77.7 | 88.2 | 75.7 | 75.5 |
| gemini-2.0-flash | 71.5 | 68.3 | 71.3 | 73.7 | 74.5 | 74.5 | 74.2 | 76.9 | 75.7 | 85.2 | 80.8 |
| gemini-2.0-flash-lite-preview-02-05 | 71.8 | 68.6 | 71.5 | 73.8 | 74.3 | 74.4 | 74.2 | 77.5 | 75.5 | 80.8 | 85.1 |

Table 12: Examples of semantically similar response pairs for the query **"Create a short summary about the Nissan R390."** and their corresponding sentence similarity scores.

| Sim. Scores. | Example Responses |
|---|---|
| 0.956 | The Nissan R390 was a racing prototype sports car developed by Nissan in the late 1990s. It was designed to compete in the 24 Hours of Le Mans and other major endurance races under the GT1 class regulations. The R390 featured a twin-turbocharged 3.5-liter V8 engine that produced around 550 horsepower. With its advanced aerodynamics and lightweight carbon fiber construction, the R390 was capable of reaching top speeds of over 200 mph (320 km/h) on the Mulsanne Straight at Le Mans. Nissan built only two examples of the R390, and the car competed in the 1997 and 1998 editions of the 24 Hours of Le Mans, achieving a best finish of 3rd place in its class in 1998. |
| | The Nissan R390 is a high-performance sports car and racing prototype that was introduced in 1998. Designed primarily for the 24 Hours of Le Mans, it represented Nissan's competitive spirit in endurance racing. With its sleek aerodynamic design, the R390 GT1 featured a 3.5-liter twin-turbocharged V8 engine, producing around 600 horsepower. Only a few road-going versions were built to meet the racing regulations, making it an extremely rare and sought-after vehicle among collectors. The R390 not only showcased Nissan's engineering prowess but also marked a significant chapter in the company's motorsport history. |
| 0.897 | The Nissan R390 GT1 was a purpose-built race car designed specifically to compete at the 24 Hours of Le Mans in 1997 and 1998. Powered by a twin-turbocharged 3.5-liter V8, the R390 was known for its sleek aerodynamics and impressive performance. While it never secured an overall victory at Le Mans, it achieved a notable third-place finish in 1998, demonstrating Nissan's racing prowess on the international stage. A single road-legal version was built to meet homologation requirements, making it one of the rarest and most valuable cars in the world. |
| | The Nissan R390 GT1 was a racing car built by Nissan in 1997-1998 to compete in the 24 Hours of Le Mans. It was developed in partnership with Tom Walkinshaw Racing (TWR) and featured a twin-turbocharged 3.5L V8 engine producing around 641 horsepower. A single road-legal version (R390 GT1 Road Car) was also produced to meet homologation requirements. The race car achieved moderate success at Le Mans, with its best result coming in 1998 when Nissan secured third, fifth, sixth, and tenth places. The R390 GT1 represented Nissan's last major prototype effort at Le Mans until their return with the GT-R LM NISMO in 2015. |
| 0.803 | The **Nissan R390** is a mid-engined, rear-wheel-drive **sports prototype** built by **Nissan Motorsport** (Australia) in the 1990s for **Group A touring car racing**. Based on the **Nissan R88C Le Mans prototype**, the R390 features a **twin-turbocharged 3.0-liter V6 engine** (initially VRH30DETT, later RB30DET), producing around **360–420 horsepower**. It was designed for **domestic and international touring car competition**, with a lightweight, aerodynamic body and a tubular spaceframe chassis. The R390 achieved some success in Australian and European touring car series but is best known as a **rare and iconic example of Nissan's motorsport engineering** during the late '90s. Only about **10–12 roadgoing examples** were produced, making it a highly collectible and sought-after classic. |
| | The Nissan R390 GT1 was a purpose-built race car developed by Nissan for the 1998 and 1999 Le Mans 24 Hours races. It was Nissan's first entry in the GT1 class, which was designed to bridge the gap between production cars and prototypes. The R390 featured a sleek, aerodynamic body and was powered by a 3.5-liter V8 engine, producing around 650 horsepower. Despite its advanced design and significant investment, the R390 faced numerous challenges and did not achieve the success Nissan had hoped for at Le Mans. However, it remains a notable part of Nissan's motorsport history and a symbol of the company's commitment to racing innovation. |
| 0.855 | The Nissan R390 GT-R is a mid-size, rear-wheel-drive sports car developed by Nissan in the mid-1990s as a concept and limited-production model to compete in Group B rally racing. Built on a spaceframe chassis and powered by a twin-turbocharged 2.6L inline-six engine, the R390 produced around 480 horsepower and featured a sequential 6-speed manual transmission. Though only a few were produced and it never entered full production, the R390 is celebrated for its advanced engineering, lightweight design, and racing heritage, making it a rare and highly regarded piece of Nissan's motorsport history. |
| | The Nissan R390 is a high-performance supercar and race car developed by Nissan in the late 1990s. Designed primarily for motorsport, it was built to compete in the 24 Hours of Le Mans under the GT1 class. The R390 was engineered in collaboration with British motorsport company Tom Walkinshaw Racing (TWR) and featured a mid-mounted 3.5-liter twin-turbocharged V8 engine, producing around 550–650 horsepower. Its aerodynamic design and lightweight carbon-fiber construction made it a formidable competitor. |
| | While the road-legal version of the R390 was produced in extremely limited numbers (reportedly only two units), the racing variant achieved notable success, finishing 3rd and 5th at the 1998 Le Mans. The R390 remains a rare and iconic symbol of Nissan's ambition in high-performance and endurance racing. |

Table 13: Examples of semantically similar response pairs for the query **"Write a 30-word essay on global warming."** and their corresponding sentence similarity scores.

| Sim. Scores. | Example Responses |
| --- | --- |
| 0.737 | Global warming, driven by greenhouse gas emissions, intensifies the climate crisis. Rising temperatures fuel extreme weather, threaten ecosystems, and demand urgent global action for a sustainable future. |
| | Global warming, a pressing issue, results from human-induced greenhouse gas emissions. It causes rising temperatures, melting ice caps, extreme weather, and sea level rise, threatening ecosystems and human life. Swift action, including renewable energy use and reduced carbon emissions, is crucial. |
| 0.782 | Global warming, driven by greenhouse gas emissions, poses a severe threat to ecosystems and human life. Urgent action is essential to mitigate its impacts and transition towards sustainable practices for future generations. |
| | Caused by human activities, global warming poses an existential threat. Rising temperatures fuel extreme weather, melting glaciers, and ecosystem disruption. Urgent action reducing emissions can mitigate catastrophic consequences. |
| 0.631 | Global warming is a pressing issue, driven by human activities like fossil fuel use. It's causing polar ice melt, sea-level rise, and extreme weather. Urgent action is needed to reduce emissions and transition to renewable energy. This is just one example, as there are many aspects of global warming that can be highlighted. Here are a few more options: 1. Global warming threatens biodiversity, with rising temperatures forcing species to adapt or face extinction. Collective effort is crucial to protect ecosystems. 2. Climate change disproportionately affects vulnerable communities, exacerbating poverty and inequality. Global cooperation is essential for climate justice. 3. The Paris Agreement aims to limit global warming to well below 2°C. Carbon neutrality by 2050 is vital to achieve this goal. 4. Individual actions matter: reducing, reusing, recycling, and conserving energy can help combat global warming. Together, small steps make a big difference. 5. Education and awareness are key to addressing global warming. Empowering young people to take action ensures a sustainable future for all. |
| | Global warming, driven by greenhouse gas emissions, escalates Earth's average temperature, causing extreme weather, melting ice caps, and threatening biodiversity and human livelihoods. |
| 0.677 | Global warming, a dire threat to our planet, is caused by human activities releasing greenhouse gases. We must act now to reduce emissions, adopt sustainable practices, and mitigate the devastating consequences. |
| | Global warming, driven by greenhouse gas emissions, intensifies the climate crisis. Rising temperatures fuel extreme weather, threaten ecosystems, and demand urgent global action for a sustainable future. |
| 0.606 | Global warming, primarily human-induced, is causing ice melt, sea-level rise, and extreme weather. Transitioning to renewables and reducing emissions are crucial to mitigate its impacts and protect our planet's future. **Word count:** 30 Here's a breakdown of the essay's structure: - **Problem** (11 words): Global warming, primarily human-induced, is causing ice melt, sea-level rise, and extreme weather. - **Solution** (19 words): Transitioning to renewables and reducing emissions are crucial to mitigate its impacts and protect our planet's future. |
| 0.934 | Global warming is an alarming environmental issue. It's caused by human activities, raising temperatures and disrupting ecosystems. Cutting emissions, using clean energy, and conservation can help mitigate its effects. Global warming, driven by human activities, is causing rising temperatures, melting ice caps, and extreme weather. Urgent action is needed to reduce emissions, protect ecosystems, and ensure a sustainable future for generations to come. |
| | Global warming, driven by human activities, is causing extreme weather, rising sea levels, and ecosystem disruption. Urgent action is needed to reduce emissions and protect our planet for future generations. |
| 0.893 | Global warming, driven by human activities like burning fossil fuels, deforestation, and industrial emissions, leads to rising temperatures, melting ice caps, extreme weather, and biodiversity loss, threatening ecosystems and future generations' well-being. Immediate action is crucial. |
| | Global warming, driven by human activities like burning fossil fuels, is causing rising temperatures, melting ice caps, and extreme weather. It threatens ecosystems, livelihoods, and biodiversity, urging urgent action to reduce emissions and protect our planet's future. |
| 0.996 | Global warming, a dire threat to our planet, is caused by human activities releasing greenhouse gases. We must act now to reduce emissions, adopt sustainable practices, and mitigate its devastating effects. |
| | Global warming, a dire threat to our planet, is caused by human activities releasing greenhouse gases. We must act now to reduce emissions, adopt sustainable practices, and mitigate climate change's devastating effects. |
| 0.840 | Global warming is a critical environmental issue caused by greenhouse gas emissions, leading to rising temperatures, melting ice caps, extreme weather, and threats to ecosystems, urging immediate global action for sustainability. |
| | Global warming poses a critical threat, driven by greenhouse gas emissions. Its consequences include rising temperatures, melting ice, extreme weather, and habitat loss, urging immediate global action to mitigate its devastating impacts. |

Table 14: Examples of semantically similar response pairs for the query **"Write me a 1-paragraph essay about the development of the economy during the Han Dynasty."** and their corresponding sentence similarity scores.

| Sim. Scores. | Example Responses |
|---|---|
| 0.933 | The Han Dynasty, which ruled China from 206 BCE to 220 CE, witnessed significant economic development that transformed the country into a major economic power. The dynasty's economic growth was fueled by a combination of factors, including the expansion of the Silk Road, a network of trade routes that connected China to Central Asia, India, and the Mediterranean. The government's efforts to standardize currency, weights, and measures, as well as its investment in infrastructure projects such as roads, canals, and irrigation systems, also facilitated trade and commerce. Additionally, the Han government's policies of reducing taxes, promoting agriculture, and encouraging entrepreneurship led to an increase in agricultural production, urbanization, and the growth of a wealthy merchant class. As a result, the Han economy experienced a period of unprecedented prosperity, characterized by a significant increase in trade, a growth in the production of goods such as silk, iron, and ceramics, and the emergence of a complex and sophisticated financial system. The economic achievements of the Han Dynasty had a lasting impact on Chinese history, laying the foundation for the country's future economic development and establishing China as a major player in the global economy. |
| | The Han Dynasty, spanning from 206 BCE to 220 CE, witnessed a significant transformation and growth in ancient China's economy, setting the foundation for its status as one of the world's earliest economic powerhouses. This period saw remarkable advancements in agriculture, trade, and technological innovations. The introduction of new crops, such as wheat and millet, along with improved iron tools and irrigation techniques, led to increased agricultural productivity, supporting a larger population. The dynasty established a centralized administration that promoted internal stability and facilitated the development of an extensive network of roads and canals, which connected major cities and ports, stimulating domestic and international trade. The famous Silk Road, a crucial trade route linking China to Central Asia and the Mediterranean, emerged during this era, fostering cultural exchange and generating substantial revenue for the empire. Additionally, the Han Dynasty's adoption of coinage as a standardized currency streamlined economic transactions, further facilitating the growth of markets and commerce. These economic achievements played a pivotal role in the prosperity and cultural flourishing of the Han era. |
| 0.936 | The Han Dynasty (202 BCE - 220 CE) marked a significant period of economic growth and innovation in Chinese history. Under Han rule, agriculture flourished due to improved iron tools, irrigation systems, and the introduction of new farming techniques, leading to increased crop yields and population growth. The dynasty also saw the establishment of state monopolies on salt and iron production, which generated substantial revenue for the government. The Silk Road reached its height during this period, facilitating extensive trade networks that connected China with Central Asia, India, and the Roman Empire, allowing for the exchange of goods such as silk, spices, and precious metals. The Han government standardized currency, weights, and measures, which streamlined commerce and led to the development of a sophisticated market economy. Additionally, technological advancements in areas such as papermaking and metallurgy contributed to industrial growth, while urbanization created new economic centers throughout the empire, resulting in a complex and prosperous economic system that would influence Chinese society for centuries to come. |
| | The Han Dynasty (206 BCE – 220 CE) witnessed significant economic development that laid the foundation for China's long-term prosperity. Agricultural production expanded due to technological advancements such as the iron plow and improved irrigation systems, which increased crop yields and supported a growing population. The government standardized currency and regulated weights and measures, facilitating trade both within the empire and along the Silk Road, which connected China to Central Asia, the Middle East, and Europe. The Han also promoted state monopolies on key industries such as salt and iron, which generated substantial revenue for the imperial treasury and strengthened central authority. These economic policies and innovations contributed to a more integrated and stable economy, reinforcing the Han Dynasty's enduring influence on Chinese civilization. |
| 0.869 | The Han Dynasty witnessed a flourishing economy driven by several key factors. Agricultural advancements, including improved irrigation techniques and the use of iron tools, significantly boosted crop yields, leading to a surplus that supported a growing population and fostered trade. The Silk Road, established during this period, connected China to the West, facilitating the exchange of not only silk but also other valuable commodities, ideas, and technologies, enriching the Han economy. Moreover, the standardization of currency, weights, and measures streamlined transactions and facilitated commerce both domestically and internationally. While the dynasty also experienced periods of state control over key industries like salt and iron, these policies were often implemented to stabilize prices and generate revenue for the government, contributing to the overall economic development, albeit with varying degrees of success. |
| | The Han Dynasty is widely regarded as a golden age in Chinese history, and its economic prosperity was a significant factor in its success. During this period, the economy flourished due to a combination of agricultural advancements, technological innovations, and expansion of trade routes. The Han rulers implemented various reforms, including land redistribution and tax systems, which boosted agricultural production and encouraged the growth of local industries. This led to an increase in surplus food production, which in turn supported population growth and the development of bustling urban centers. Additionally, the invention of new tools and irrigation techniques further enhanced farming efficiency. The Han Dynasty also witnessed the expansion of the Silk Road, facilitating trade and cultural exchange with other civilizations, and promoting economic growth through the import and export of various goods such as silk, spices, and exotic items. This thriving economy enabled the Han Dynasty to invest in public works, infrastructure, and military campaigns, establishing a stable and prosperous society that would influence future generations. |

Table 15: Examples of semantically similar response pairs for the query **"Create the first verse of a wedding vow."** and their corresponding sentence similarity scores.

| Sim. Scores. | Example Responses |
|---|---|
| 0.819 | As I stand before you today, with the warmth of love in my heart and the light of our future shining bright, I promise to cherish and honor you, to support your dreams and to be your safe haven. You are the missing piece I never knew I needed, the rhythm that makes my heart sing, and the forever home where I can be my true self. |
| | As I stand before you today, surrounded by the love of our friends and family, I promise to cherish and support you through all of life's joys and challenges. You are the sunshine that brightens every day, the calm in every storm, and the safe haven where I can always be myself. |
| 0.965 | From this day forward, I choose you to be my partner, To live with you and laugh with you; to stand by your side. I take you with all your strengths and flaws as I offer mine. I will love you faithfully through the best and worst, Through the difficult and the easy. |
| | From this day forward, I choose you to be my partner, To live with you and laugh with you; to stand by your side. I take you with all of your strengths and imperfections, As I offer myself to you with my faults and complexities. |
| 0.837 | In this moment, with all that I am, I stand before you, my heart open wide, With every heartbeat, I promise to cherish, to love, and to walk by your side. Through laughter and tears, in joy and in sorrow, I vow to be true, For today and forever, my greatest adventure begins with you. |
| | As I stand before you, heart wide open and hand in hand, I vow to walk through life with you, through joy and through the land of uncertainty, to love not just in words, but in every deed— To honor who you are, and the dreams still growing in your heart and head. |
| 0.953 | Today I stand before you, my heart full and true, To promise my love and my life to you. In your eyes I've found my home, in your smile my peace, And with these words, my devotion will never cease. |
| | Today I stand before you, my heart full and true, To promise my love and my life to you. In your eyes I've found my home, my peace, my light, And I vow to cherish you, day and day, through darkness and bright. |
| 0.868 | As I stand before you today, I am filled with so much love and gratitude for the gift of you in my life. From the moment I met you, I knew that you were someone special, and with each passing day, my love and adoration for you has only grown stronger. You are my partner, my best friend, and my soulmate, and I promise to love, support, and cherish you for all the days of my life. |
| | As I stand before you today, I am overwhelmed with emotion and gratitude, for I am about to promise my heart, my love, and my forever to the person who has brought joy, laughter, and adventure into my life. You are my best friend, my soulmate, and my everything, and I vow to love and cherish you, to support your dreams, and to stand by your side through all of life's triumphs and challenges, now and forevermore. |
| 0.563 | Before our family and friends, and under this boundless sky, I, (Your Name), take you, (Partner's Name), to be my lawfully wedded partner. I promise to cherish you, not just today, but every day, for all the days to come. |
| | Today, I stand before you, my heart open and full, Promising to love you through life's every pull. In your laughter, I find my sweetest song, By your side, I know I've always belonged. |
| 0.741 | Standing here before you, [Partner's Name], surrounded by the people we love most, my heart overflows with a joy I never knew possible. From the moment our paths intertwined, I knew my life would never be the same. You have shown me the truest meaning of love, acceptance, and unwavering support, and today, I promise to dedicate my life to cherishing and nurturing the beautiful connection we share. |
| 0.898 | In this moment, as I stand before you, My heart is filled with love, so pure and true. I gaze into your eyes, my soulmate and best friend, And vow to stand beside you, until the very end. Through laughter and through tears, in sickness and in health, I promise to support you, to cherish and respect. Our love will be the anchor, that keeps us strong and whole, A bond that lasts forever, a love that fills our souls. My love, from the moment our paths intertwined, A spark ignited, a love defined. You painted my world with vibrant hues, And chased away the blues. With you by my side, I feel complete, A love like ours, oh so sweet. Today, as we stand hand in hand, I vow to love you, across this land. |
| | My love, from the moment our paths intertwined, A spark ignited, a love divine. You painted my world with vibrant hues, And whispered promises my heart knew were true. Today, standing here, hand in hand, I give you my heart, my soul, this land, Where our love will forever reside, A love so strong, it cannot hide. |
| 0.580 | From this day forward, I promise to walk beside you, Through every season of life, With love as our compass and faith as our guide. |
| | As I stand here before you, my heart filled with love and grace, I promise to cherish and honor, for all of time and space, To be your partner in laughter, your shelter in the storm, To love you deeply and truly, with each waking morn. |

Table 16: Examples of semantically similar response pairs for the query **"Write a metaphor involving time."** and their corresponding sentence similarity scores.

| Sim. Scores. | Example Responses |
|---|---|
| 0.792 | Time is a river, ceaseless and unyielding, carrying moments like leaves upon its surface, drifting into the unseen horizon. |
| | Time is like a vast, meandering river, constantly flowing forward, carving its path through the landscape of existence. Every moment is a drop of water, individually insignificant, yet collectively shaping the course of history. We are boats navigating its current, sometimes drifting with its relentless movement, other times struggling against its pull. The past is the upstream, where memories and experiences are deposited, forming the foundation of our being. The future is the downstream, unseen and mysterious, beckoning us to journey forth. But no matter how hard we paddle, or how we try to dam its flow, time marches on, relentless and eternal, reminding us of the precious, fleeting nature of every second that passes. |
| 0.842 | Time is an invisible river, flowing steadily and relentlessly, carrying moments like leaves caught in its current, some gently floating by while others are swept swiftly into the vastness of memory. |
| | Time is a relentless river, flowing endlessly, carrying the moments of our lives downstream, never to be revisited. |
| 0.864 | Time is a river, ceaselessly flowing, carrying us along its currents. We may try to navigate its waters, steering toward calm eddies or bracing against its rapids, but we cannot stop its movement. Moments drift past like leaves, some lingering briefly on the surface before vanishing into the distance, while others sink into the depths, remembered only as faint ripples. Though we cannot hold the river, we can choose how we move within it—whether we struggle against the tide or learn to glide with its rhythm. |
| | Time is a river, constantly flowing forward, carving through the landscape of our lives with relentless determination. It shapes the banks of our memories, deposits the sediment of our experiences, and carves out the canyons of our milestones. Some stretches are serene, meandering slowly through peaceful moments, allowing us to savour each droplet of existence. Other parts rush headlong, thunderous and unstoppable, carrying us along in its powerful current, challenging us to keep our footing on the slippery rocks of the present. In its journey, time erodes the impermanent and smooths the rough edges of what remains, creating the valleys and mountains of our history. We, like leaves or debris caught in its flow, must navigate its inexorable course, occasionally trying to swim against the tide, yet ultimately carried along by its unstoppable momentum. |
| 0.592 | Time is a relentless river, flowing endlessly without pause or mercy, sweeping away moments like leaves carried by the current. |
| | Time is a tapestry, intricately woven with threads of past, present, and future. Each moment is a delicate stitch, connected to the ones before and after it, forming a rich and complex pattern that tells the story of our lives. Just as a single snag can unravel the entire fabric, one careless decision can alter the course of our journey, while each careful and thoughtful choice can strengthen the weave, creating a beautiful and lasting work of art. |
| 0.862 | Time is a river, flowing ceaselessly forward, carrying with it the fragments of moments that shape the ever-changing landscape of our lives. |
| | Time is a river, ceaseless and unyielding, carrying moments like fallen leaves, swept away before they can be grasped. |
| 0.689 | Time is a river with no bridges — you can wade in its currents, but you can never step into the same water twice. |
| | Time is a river, ceaseless and unyielding, carrying moments like fallen leaves, swept away before they can be grasped. |
| 0.663 | Time is a river, endlessly carving its way through the landscape of existence, carrying moments like drifting leaves that vanish around the bend, never to be seen again. |
| | Time is a tapestry, intricately woven with threads of past, present, and future. Each moment is a delicate stitch, carefully placed to create a complex and ever-evolving pattern. As the tapestry grows, new threads are added, while old ones begin to fray, leaving behind a fragile yet beautiful memory of what has been. And just as a tapestry can be admired from afar, yet its true beauty lies in the intricate details, so too can time be appreciated for its grand sweep, yet its true significance lies in the small, precious moments that make up its fabric. |
| 0.864 | Time is a river, ceaselessly flowing, carrying us along its currents. We may try to navigate its waters, steering toward calm eddies or bracing against its rapids, but we cannot stop its movement. Moments drift past like leaves, some lingering briefly on the surface before vanishing into the distance, while others sink into the depths, remembered only as faint ripples. Though we cannot hold the river, we can choose how we move within it—whether we struggle against the tide or learn to glide with its rhythm. |
| | Time is a river, constantly flowing forward, carving through the landscape of our lives with relentless determination. It shapes the banks of our memories, deposits the sediment of our experiences, and carves out the canyons of our milestones. Some stretches are serene, meandering slowly through peaceful moments, allowing us to savour each droplet of existence. Other parts rush headlong, thunderous and unstoppable, carrying us along in its powerful current, challenging us to keep our footing on the slippery rocks of the present. In its journey, time erodes the impermanent and smooths the rough edges of what remains, creating the valleys and mountains of our history. We, like leaves or debris caught in its flow, must navigate its inexorable course, occasionally trying to swim against the tide, yet ultimately carried along by its unstoppable momentum. |
| 0.792 | Time is a river, ceaseless and unyielding, carrying moments like leaves upon its surface, drifting into the unseen horizon. |
| | Time is like a vast, meandering river, constantly flowing forward, carving its path through the landscape of existence. Every moment is a drop of water, individually insignificant, yet collectively shaping the course of history. We are boats navigating its current, sometimes drifting with its relentless movement, other times struggling against its pull. The past is the upstream, where memories and experiences are deposited, forming the foundation of our being. The future is the downstream, unseen and mysterious, beckoning us to journey forth. But no matter how hard we paddle, or how we try to dam its flow, time marches on, relentless and eternal, reminding us of the precious, fleeting nature of every second that passes. |

Table 17: Examples of repetitive responses produced by different models when prompted paraphrased versions of the same prompt: **"Write a metaphor involving time."**

| Model | Prompt | Response |
|---|---|---|
| **Llama-3.1-8B-Instruct** | Write a metaphor involving time. | Time is a **river** that flows relentlessly, carrying memories and experiences along its currents, shaping the shores of our lives with every passing moment, and ultimately emptying into the vast ocean of eternity. |
| | Create a metaphor that relates to time. | A **river** of moments flows through the expanse of existence, ever-changing and ever-present. Its waters, like grains of sand, slip through our fingers as the currents of time carry us along. Each eddy and ripple forms a distinct memory, while the relentless downstream drifts away with the passage of days. |
| | Compose a metaphor centered around time. | Time is a **river**, its waters constantly flowing, carrying moments past, present, and future along with them. It carves paths through the landscape of our lives, wearing down the stones of yesterday, sculpting the shores of today, and shaping the mountains of tomorrow. Its currents twist and turn, sometimes swift, sometimes still, but always moving forward, carrying us with it, ever-changing, ever-flowing. |
| | Come up with a metaphor that involves time. | Time is a **river**, flowing constantly and effortlessly, carrying moments, memories, and experiences with it. |
| | Generate a metaphor that features time. | Time is a **river**, flowing relentlessly forward, shaping the landscape of our lives as it twists and turns, carving out moments and memories, sometimes gently, sometimes violently, but always relentlessly moving, carrying us along with it, leaving behind a trail of past and present, as we drift toward the unknown shores of the future. |
| **Qwen1.5-110B-Chat** | Create a metaphor that relates to time. | Time is like a **river** that flows constantly, carrying us along its current. Just as a **river** carves its path through the landscape, time shapes our lives and creates a unique journey for each of us. We can dip our toes in to pause and reflect, but we can never stop the flow or turn back to where we started. Every moment is a new downstream destination, and once it passes, it becomes a part of the vast ocean of memories. |
| | Compose a metaphor centered around time. | Time is a **river** that flows relentlessly, carrying us forward on its current, while the memories of our past are like pebbles on its banks, polished smooth by the passage of each moment. Time is a **river** that flows incessantly, carrying us along its current, where moments are like pebbles polished smooth by its touch, and memories are the ripples that linger long after the waves have passed. |
| | Come up with a metaphor that involves time. | Time is like a **river** that flows continuously, carrying us along its current. Just as a **river** carves its path through the earth, time shapes our lives and leaves its mark on us. We can dip our toes in to savor a moment, or let it sweep us away in a rush of events. But once it passes, we can never recapture the same moment again, for time's current is relentless and always moving forward. |
| | Generate a metaphor that features time. | Time is a **river** that flows continuously, carrying us along its current, never pausing to let us catch our breath, but always moving us forward towards an unknown destination. |
| **gemma-2-27b-it** | Write a metaphor involving time. | Time is a **river**, relentlessly flowing, carrying us all downstream. Some days we float peacefully on its surface, enjoying the sun, while others we're tossed by rapids, struggling to keep our heads above water. But no matter how we navigate its currents, we are all carried towards the inevitable sea. |
| | Create a metaphor that relates to time. | Time is a **river**, constantly flowing, carrying us along its current. We can paddle and steer, but ultimately, we are swept towards the sea. |
| | Compose a metaphor centered around time. | Time is a **river**, rushing onward, carrying us all on its current. Some days we paddle gently, enjoying the scenery. Others, we're caught in rapids, struggling to stay afloat. But the **river** flows ever onward, relentless and unwavering, towards the vast, unknown ocean of eternity. |
| | Come up with a metaphor that involves time. | Time is a **river**, rushing forward relentlessly, carrying us all along its current. We can dip our toes in, swim against it for a while, or let it carry us to new shores, but ultimately, we cannot escape its flow. |
| | Generate a metaphor that features time. | Time is a **river**, rushing relentlessly towards the sea. We are all leaves, swept along its currents, some skimming the surface, others caught in eddies and whirlpools, but all ultimately heading towards the same destination. |

Table 18: Examples of repetitive responses produced by different models when prompted paraphrased versions of the same prompt: **"Write a paragraph about how the internet shaped society."**

| Model | Prompt | Response |
|---|---|---|
| **Llama-3.3-70B-Instruct** | Write a paragraph about how the internet shaped society. | The internet has **profoundly** shaped society in countless ways, revolutionizing the way we communicate, access information, and interact with one another. [...] |
| | Compose a paragraph discussing the impact the internet has had on society. | The advent of the internet has **profoundly** impacted society, revolutionizing the way we communicate, access information, and conduct our daily lives. [...] |
| | Draft a paragraph explaining the ways in which the internet has influenced society. | The advent of the internet has **profoundly** impacted society, transforming the way we communicate, access information, and interact with one another. [...] |
| | Write a paragraph describing how society has been transformed by the internet. | The advent of the internet has revolutionized modern society in **profound** ways, transforming the way we communicate, access information, and interact with one another. [...] |
| | Create a paragraph outlining the effects of the internet on society. | The internet has **profoundly** impacted society, bringing about numerous transformative effects that have reshaped the way we live, interact, and access information. [...] |
| **Qwen2.5-72B-Instruct** | Write a paragraph about how the internet shaped society. | The internet has **profoundly** reshaped society in numerous ways, transforming how we communicate, work, learn, and interact with the world. [...] |
| | Compose a paragraph discussing the impact the internet has had on society. | The internet has **profoundly** transformed society in myriad ways, reshaping how we communicate, access information, and conduct business. [...] |
| | Draft a paragraph explaining the ways in which the internet has influenced society. | The internet has **profoundly** transformed society in numerous ways, impacting nearly every aspect of human life. [...] |
| | Write a paragraph describing how society has been transformed by the internet. | The internet has **profoundly** transformed society in myriad ways, redefining how we communicate, work, learn, and socialize. [...] |
| | Create a paragraph outlining the effects of the internet on society. | The internet has **profoundly** transformed society in numerous ways, affecting communication, commerce, education, and social interactions. [...] |
| | Create a paragraph outlining the effects of the internet on society. | The internet has **profoundly** transformed society in numerous ways, influencing how people communicate, work, learn, and entertain themselves. [...] |
| **gemma-2-27b-it** | Write a paragraph about how the internet shaped society. | The internet has **profoundly** reshaped society, revolutionizing communication, access to information, and the way we interact with the world. [...] |
| | Write a paragraph about how the internet shaped society. | The internet has **profoundly** reshaped society, ushering in a new era of interconnectedness and information access. Its decentralized nature has democratized knowledge, allowing individuals to access vast repositories of information and connect with others across geographical boundaries. [...] |
| | Compose a paragraph discussing the impact the internet has had on society. | The internet has revolutionized society in **profound** ways, impacting nearly every facet of human life. [...] |
| | Compose a paragraph discussing the impact the internet has had on society. | The internet has **profoundly** reshaped society, revolutionizing communication, access to information, and social interaction. [...] |
| | Draft a paragraph explaining the ways in which the internet has influenced society. | The internet has **profoundly** reshaped society, revolutionizing communication, information access, and daily life. [...] |
| | Draft a paragraph explaining the ways in which the internet has influenced society. | The internet has **profoundly** transformed society, revolutionizing communication, information access, commerce, and social interaction. [...] |
| | Write a paragraph describing how society has been transformed by the internet. | The internet has fundamentally reshaped society, acting as a catalyst for **profound** changes in communication, commerce, information access, and social interaction. [...] |
| | Create a paragraph outlining the effects of the internet on society. | The internet has **profoundly** transformed society, revolutionizing communication, information access, and social interactions. [...] |
| | Create a paragraph outlining the effects of the internet on society. | The internet has **profoundly** reshaped society, fostering unprecedented connectivity and access to information. [...] |

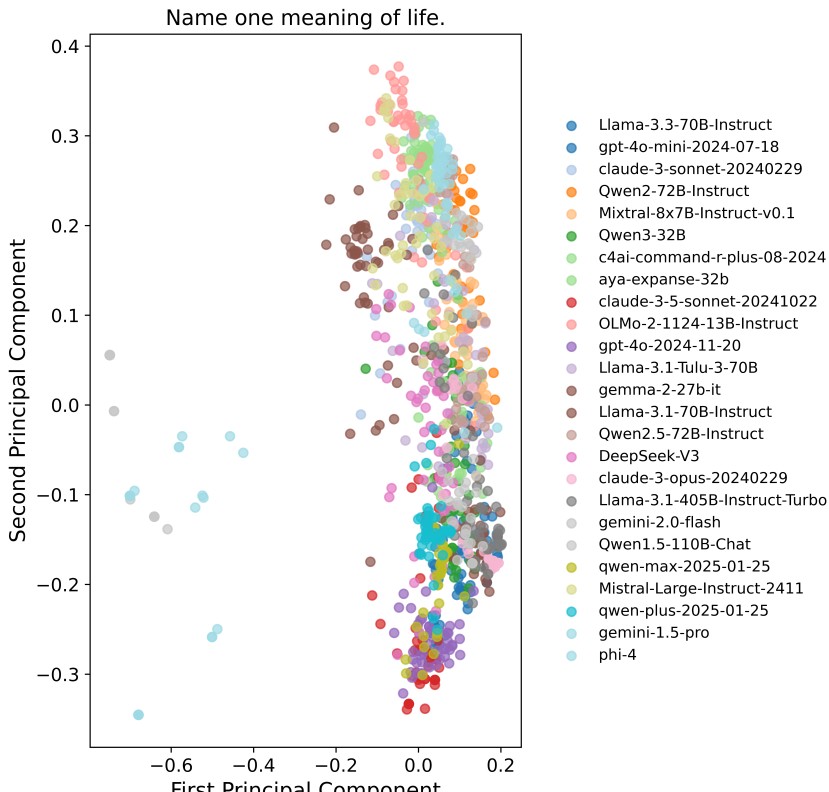

Figure 15: Responses to the query **"Name one meaning of life."** clustered by applying PCA to reduce sentence embeddings to two dimensions. Each of the 25 models generates 50 responses using top-$p$ sampling ($p = 0.9$) and temperature $= 1.0$. We observe prominent clusters, indicating substantial overlap in the responses across many models.

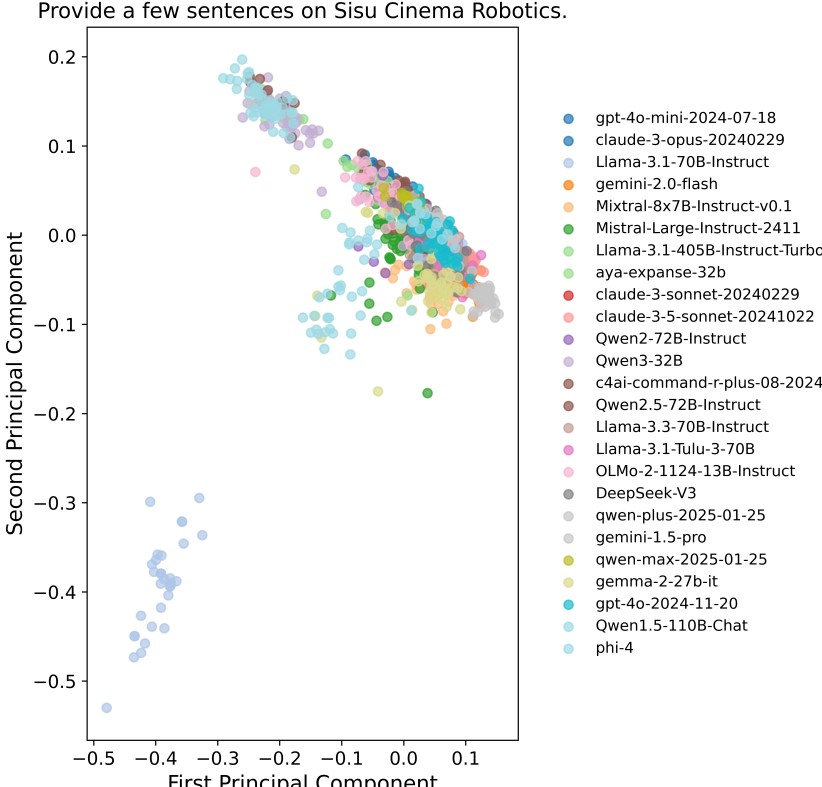

Figure 16: Responses to the query **"Provide a few sentences on Sisu Cinema Robotics."** clustered by applying PCA to reduce sentence embeddings to two dimensions. Each of the 25 models generates 50 responses using top-$p$ sampling ($p = 0.9$) and temperature $= 1.0$. We observe prominent clusters, indicating substantial overlap in the responses across many models.

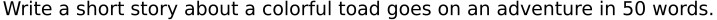

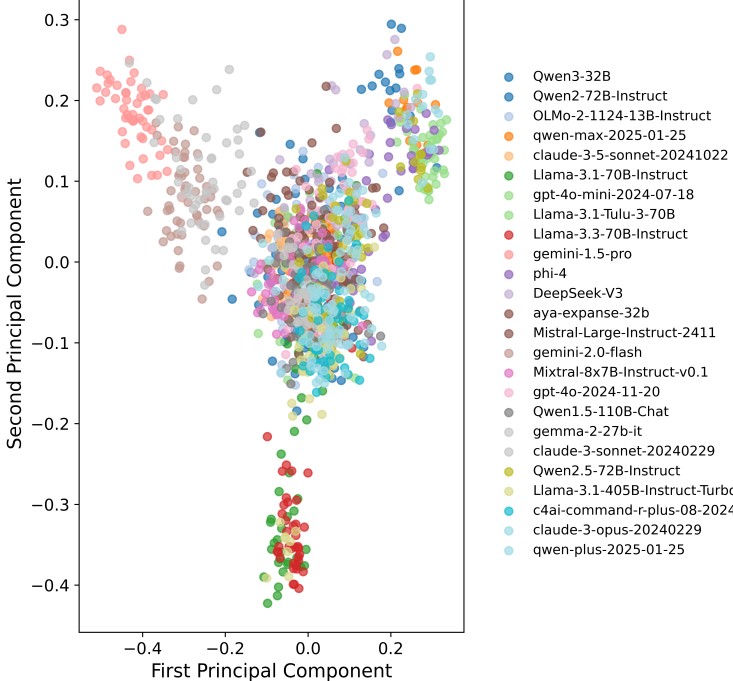

Figure 17: Responses to the query **"Write a short story about a colorful toad goes on an adventure."** clustered by applying PCA to reduce sentence embeddings to two dimensions. Each of the 25 models generates 50 responses using top-$p$ sampling ($p = 0.9$) and temperature $= 1.0$. We observe prominent clusters, indicating substantial overlap in the responses across many models.

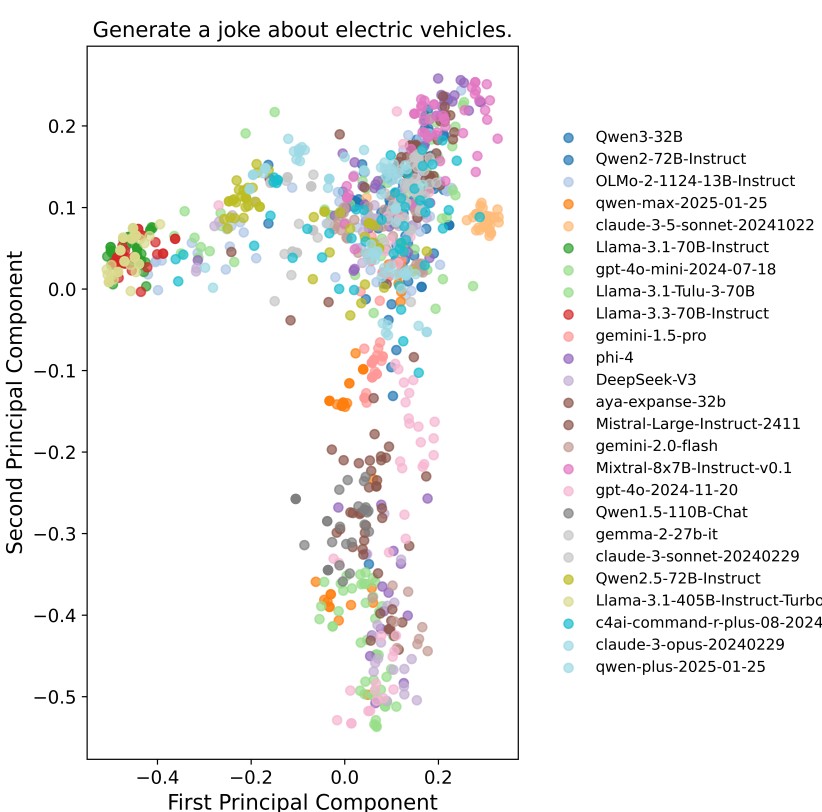

Figure 18: Responses to the query **"Generate a joke about electric vehicles."** clustered by applying PCA to reduce sentence embeddings to two dimensions. Each of the 25 models generates 50 responses using top-$p$ sampling ($p = 0.9$) and temperature $= 1.0$. We observe prominent clusters, indicating substantial overlap in the responses across many models.

# D  Comparing Model Ratings to Human Scores for Open-Ended Generations

## D.1  Human Annotation Details

**Annotation data preparation.**  We randomly select 50 open-ended user queries from INFINITY-CHAT100. For each query, our goal is to collect 15 distinct responses from the full set of models listed in Table 5. Each model generates 50 responses per query, which often include similar or repetitive outputs. To promote diversity, we aggregate all responses for a given query and apply clustering to partition them into 15 groups. We then sample one response from each cluster, ensuring the annotated set is both diverse and representative.

We collect responses for both *absolute rating* (i.e., a rating from 1 to 5 indicating the overall quality of a response to a query) and *pairwise comparisons* (i.e., indicating a strong or weak preference between a pair of responses to the same query) from multiple human annotators for each data point. In the absolute rating setup, we gather 15 distinct responses for each of 50 sampled open-ended prompts and collect 25 annotations for each (Query, Response) pair, yielding a total of $25 \times 15 \times 50 = 18{,}750$ human labels. In the pairwise comparison setup, we construct 10 distinct response pairs for each of the same 50 prompts and collect 25 annotations for each (Query, Response 1, Response 2) tuple, resulting in $25 \times 10 \times 50 = 12{,}500$ annotations. In total, we contribute 31,250 human annotations and release the first resource to provide abundant human absolute and preference ratings for each open-ended response, enabling the study of distributional human preferences over open-ended text generations, where multiple responses may be equally valid.

**Annotator recruitment and annotation details.**  We recruit human annotators from Prolific.[2] To ensure relevance and data quality, annotators are prescreened based on a comprehensive set of criteria (Table 21). Eligible participants must have English as both their primary and first language and demonstrate fluency in English. They are also required to have completed at least a high school diploma or equivalent, with acceptable education levels including technical or community college, undergraduate, graduate, or doctoral degrees. To ensure annotator reliability, we restrict participation to individuals with 100 to 10,000 prior submissions and an approval rate between 99% and 100%. Annotators are compensated at an average rate of $15 per hour.

Screenshots of the annotation interface are shown in Figure 19-20 for the absolute rating task and Figure 21-22 for the pairwise preference rating task. In the absolute rating task, each annotation session consists of 15 examples, while in the pairwise preference task, each session includes 20 examples. Annotators may choose to complete multiple sessions. Table 22 presents a detailed breakdown of annotator demographic information.

**Data distribution.**  Figure 9 shows the distribution of Shannon entropy computed over the 25 human ratings for each (Query, Response) pair in the absolute rating setup. The annotation label distributions vary substantially across examples, highlighting the diversity of human judgments. This figure complements Figure 7 in the main paper, which presents analogous results for the pairwise preference task.

## D.2  Model Selection and Score Generation Details

We consider the ratings of three types of models: LMs, rewards models, and LM judges. We introduce the model choices and the evaluation setups in the following section. The full list of models can be found in Table 23 - 26.

**LMs.**  Here, we assess the quality of each response to a given query based on its perplexity score under a fixed language model. Lower perplexity indicates that the response is more fluent and likely under the model's distribution, serving as a proxy for higher quality in terms of linguistic plausibility and coherence. Given a response composed of tokens $x_1, x_2, \ldots, x_N$, perplexity is calculated as:

$$\text{Perplexity} = \exp\left(-\frac{1}{N}\sum_{i=1}^{N} \log p(x_i \mid x_{<i})\right)$$

---

[2]https://www.prolific.com

where $p(x_i \mid x_{<i})$ denotes the model's predicted probability of token $x_i$ given its preceding context. This provides a model-based estimate of the response's fluency and alignment with natural language patterns. To evaluate the correlation between language model ratings and human scores, we compute the Pearson correlation between the negative perplexity and average human scores. Higher negative perplexity corresponds to responses that the model considers higher quality.

**Reward models.** In Reinforcement Learning from Human Feedback (RLHF), reward models are trained to assign a scalar score to a generated response, reflecting its alignment with human preferences. These models typically learn from human-annotated pairwise comparisons (e.g., which of two responses is better), capturing nuanced judgments of quality, helpfulness, or safety. Given a prompt–response pair, a reward model outputs a scalar score indicating the quality of the response relative to the prompt, with higher scores representing better quality. To assess reward models' correlation with human scores, we compute the Pearson correlation between the reward model scalar scores and average human scores.

**LM judges.** Prompting-based LM judges are language models guided to act as evaluators through carefully crafted prompts, rather than being explicitly fine-tuned for scoring tasks. Typically, the model receives a system prompt instructing it to assess a response to a given query based on specific criteria—such as helpfulness, correctness, or safety—followed by a structured input containing the query and the response. The model then outputs a judgment, usually as a score or brief justification. This method leverages the model's in-context reasoning capabilities and avoids the need for additional reward model training.

In our evaluation, we consider two types of LM judges: (1) off-the-shelf GPT-4o[3], and (2) Prometheus [40][4], an open-source, fine-tuned model capable of producing fine-grained scores based on a user-provided evaluation rubric. For both models, we apply two sets of evaluation rubrics: one using only an **Overall** rating, and another based on the **HHH** rubric (Helpful, Honest, Harmless) derived from the Constitutional AI framework [9]. The prompt used for the overall judgment is shown in Figure 23, while the prompt for the HHH-based evaluation is provided in Figure 24. The LM judges assign scores on a 1-to-5 scale according to the provided evaluation rubric. We then compute the Pearson correlation between the raw scores given by the LM judges and the corresponding average human ratings.

**Motivation for using average human ratings.** Our motivation arises from how reward models (or LM judges) are currently used in LM training. These models evaluate responses to open-ended queries where no single ground truth exists. In such settings, different annotators may favor different responses, yet their average human scores are often similar, indicating that multiple responses can be of comparable overall quality despite individual variation. However, existing reward models are not trained to recognize that several responses can each be high-quality. Consequently, they tend to assign substantially different scores to such responses, leading downstream models to treat one as clearly superior even though humans collectively regard both as valid. This produces a narrow notion of what counts as "high quality."

Our data collection design directly addresses this issue: by gathering 25 human ratings per example, we capture a broad range of individual preferences, while the average score reflects shared human judgment across subpopulations. Empirically, we examine whether current reward models, LM judges, and LM perplexity correlate less consistently with responses that humans broadly consider comparably good. This motivates our choice to compute human correlation using average human scores, as it best represents the intended scenario, where multiple responses of similar average quality should be recognized as equally valid.

### D.3 Similar-Quality Responses

There is no single gold-standard approach for selecting subsets of responses with comparable quality given our data structure. In §4.3 of the main paper, we reported results using Tukey's fences to identify similar-quality examples in the absolute rating setup, noting that this is only one possible method among many. To test the robustness and generalizability of our conclusions, we further

---

[3]gpt-4o-2024-11-20
[4]prometheus-eval/prometheus-7b-v2.0

include results based on alternative similar-quality subset selection methods in Table 19. We confirm that our findings remain consistent across different subset selection methods, further strengthening the robustness of our conclusions. Here are detailed descriptions of the methods presented in Table 19:

- **Optimized Sliding**: This method sorts the list and uses a sliding window of size N to find the segment with the smallest range. It's efficient and guarantees a contiguous cluster. An early exit occurs if a window with zero range (identical values) is found, since that's the best possible outcome.

- **Centroid-Based**: Also a brute-force method, this one evaluates all possible subsets of size N and measures how tightly the numbers cluster around their mean (centroid) using variance. The subset with the smallest variance is chosen, ensuring the numbers are closely grouped around a central value. It's conceptually similar to k-means clustering in 1D.

- **Distance-Based**: This brute-force method checks all possible subsets of size N and computes the sum of pairwise distances within each subset. It selects the subset with the smallest total distance, guaranteeing the most tightly packed group. It's exact but computationally expensive since it explores every combination.

- **Tukey**: This method first applies Tukey's fences to identify and exclude outliers before selecting a cluster of N values that are closest together. After filtering, it uses a sliding window on the sorted inlier values to find the subset with the smallest range. This approach balances robustness to outliers with local compactness, ensuring the chosen numbers form a tight, contiguous cluster within the main data distribution.

- **Median Expansion**: This method starts from the median of the sorted list and expands outward to include the closest values until reaching the desired subset size. The idea is that the median anchors the subset in the center of the data, ensuring the selected numbers are as balanced and tightly clustered as possible.

- **Gap-Based**: This method sorts the values and identifies the smallest N-1 gaps between consecutive numbers. It then builds a subset spanning from the first to last chosen gap, ensuring the selected values are tightly packed together. If the range is larger than N, it applies a local sliding window to refine. The approach balances efficiency with a direct focus on minimizing the spacing between included numbers.

Moreover, to provide a model-level breakdown of the results presented in Figure 10 and 11 of the main paper, Table 23 reports detailed results across all models for similar-quality subsets (as determined by Tukey's fences) in the absolute rating task. The corresponding breakdown for the pairwise preference rating task is provided in Table 25.

Table 19: Spearman correlation coefficients of various similarity-based metrics across subsets of the top-% of most similar-quality examples, evaluated between average human scores and LM perplexities, reward model scores, and LM judge scores. See §Appendix D.3 for details on all similar subset selection methods. **L** denotes methods applied to responses from the same query, while **G** denotes methods applied to the global pool of responses across all queries.

| Method | LM Perplexities (full = .361) | | | Reward Model Scores (full = .330) | | | LM Judge Scores (full = .305) | | |
|---|---|---|---|---|---|---|---|---|---|
| Top % of Sim. Quality | 80% | 60% | 40% | 80% | 60% | 40% | 80% | 60% | 40% |
| Optimized Sliding (L) | .365 | .412 | .341 | .316 | .312 | .300 | .268 | .248 | .226 |
| Centroid-Based (L) | .347 | .372 | .357 | .300 | .297 | .278 | .259 | .230 | .206 |
| Distance-Based (L) | .346 | .372 | .357 | .300 | .297 | .278 | .259 | .230 | .206 |
| Tukey (L) | .365 | .412 | .341 | .316 | .312 | .300 | .268 | .248 | .226 |
| Median Expansion (L) | .351 | .399 | .428 | .290 | .280 | .332 | .249 | .222 | .244 |
| Optimized Sliding (G) | .247 | .242 | .149 | .183 | .178 | .096 | .157 | .138 | .121 |
| Gap-Based (L) | .373 | .414 | .387 | .318 | .314 | .326 | .265 | .260 | .226 |
| Tukey (G) | .247 | .242 | .149 | .183 | .178 | .096 | .157 | .138 | .121 |
| Median Expansion (G) | .302 | .262 | .265 | .254 | .237 | .164 | .192 | .174 | .118 |
| Gap-Based (G) | .246 | .241 | .234 | .191 | .210 | .159 | .166 | .135 | .091 |

## D.4 Disagreed Responses

Again, to test the robustness and generalizability of our conclusions, we further include results based on alternative disagreed subset selection methods in Table 20. We confirm that our findings remain consistent across different subset selection methods, further strengthening the robustness of our conclusions. Here are detailed descriptions of the methods presented in Table 20:

- **Entropy**: We calculate entropy over all 5 fine-grained labels.
- **Entropy Grouped**: We group the fine-grained 5 labels (strong prefer 1, slight prefer 1, similar, slight prefer 2, strong prefer 2) into 3 polarized labels (prefer 1, similar, prefer 2), and then calculate entropy.
- **Gini Impurity**: Measures the probability that two randomly chosen annotators assign different labels, with higher values indicating more disagreement.
- **Pairwise Disagreement**: Computes the fraction of annotator pairs that give different labels, directly capturing disagreement frequency.
- **Majority vs. Minority**: Calculates the proportion of annotators who did not select the majority label, reflecting deviation from consensus.
- **Fleiss Kappa Single**: Adjusts observed agreement among annotators for the agreement expected by chance, providing a chance-corrected reliability score.

Moreover, to provide a model-level breakdown of the results presented in Figure 10 and 11 of the main paper, Table 24 presents the breakdown of results across all models for disagreed subsets in the absolute rating task. Similarly, Table 26 shows the corresponding breakdown for the pairwise preference rating task.

Table 20: Spearman correlation coefficients of various disagreement-based metrics across subsets of the top-$N$ most disagreed examples, evaluated between average human scores and LM perplexities, reward model scores, and LM judge scores. See §Appendix D.4 for details on all similar subset selection methods.

| Method | LM Perplexities (full = .361) | | | | Reward Model Scores (full = .330) | | | | LM Judge Scores (full = .305) | | | |
|---|---|---|---|---|---|---|---|---|---|---|---|---|
| **Top N of Disagreed** | 120 | 90 | 60 | 30 | 120 | 90 | 60 | 30 | 120 | 90 | 60 | 30 |
| Entropy | .170 | .045 | -.030 | -.108 | .292 | .228 | .175 | -.073 | .287 | .254 | .276 | .070 |
| Entropy Grouped | .137 | .004 | .049 | -.160 | .293 | .246 | .081 | .153 | .247 | .227 | .121 | .139 |
| Gini Impurity | .071 | .015 | .029 | -.108 | .331 | .284 | .129 | -.043 | .299 | .278 | .216 | .162 |
| Pairwise Disagreement | .063 | .015 | .029 | .038 | .337 | .284 | .129 | -.012 | .295 | .278 | .216 | .103 |
| Majority vs. Minority | .202 | .177 | .179 | .114 | .363 | .377 | .322 | .378 | .338 | .277 | .281 | .188 |
| Fleiss Kappa Single | .285 | .317 | .174 | .268 | .231 | .272 | .168 | .208 | .278 | .316 | .326 | .316 |

Table 21: Prescreening criteria for annotator selection at Prolific.

| Prescreening Criterion | Requirement |
|---|---|
| Primary Language | English |
| First Language | English |
| Fluent Languages | English |
| Highest Education Level Completed | High school diploma/A-levels, Technical/community college, Undergraduate degree (BA/BSc/other), Graduate degree (MA/MSc/MPhil/other), Doctorate degree (PhD/other) |
| Number of Previous Submissions | 100–10,000 |
| Approval Rate | 99–100% |

Table 22: Demographic summary of all annotators.

| Demographic Dimension | Summary |
|---|---|
| Total Annotators | 2,296 |
| Disclosed Demographic Info | 95.0% (2,181 / 2,296) |
| Nationality | 38 unique; Top 3: United States (43.5%), United Kingdom (29.3%), Canada (12.5%) |
| Age | Mean = 39.9, SD = 13.4 |
| Ethnicity | White (68.6%), Black (13.8%), Asian (9.3%), Mixed (5.3%), Other (2.6%) |
| Education | Undergraduate degree (40.8%), Graduate degree (21.4%), High school diploma (20.4%), Technical/Community college (13.9%), Doctorate degree (3.5%) |
| Sex | Male (50.3%), Female (49.2%) |

## Which response is better, or are they similar?

* Indicates required question

---

**Please read the instructions carefully before answering the questions.**
----------------------------------------
[Overall Instructions]:
You will be given an open-ended user query, and two responses answering the user query. You'll be asked to rate which one of the response is better, or if they have similar quality. If you're unsure about a choice, feel free to follow your intuition.
----------------------------------------
[Partial Texts]:
It's possible that the responses are cut off without an ending due to word limits. If it's obvious that the responses are cut off, please evaluate the texts based on the provided partial texts.

---

What's your **unique Prolific ID**? *

Your answer

Figure 19: Screenshot of the human annotation interface for the absolute rating task (screen 1).

Q2: *

-----------------------------------------

[User Query]:
Write a Google ad with 2 sentences and a 30-character limit per sentence for mobile car detailing.

-----------------------------------------

[Response 1]:
**Sentence 1:**
"Expert mobile car detailing. Convenient, scratch-free shine at your door."

**Sentence 2:**
"Interior, exterior perfection. Book now, hit the road happy!"

-----------------------------------------

[Response 2]:
**Shine bright. Get a spotless car today.**

**Mobile detailing, we come to you.**

-----------------------------------------

◯  Response 1 is much better than Response 2

◯  Response 1 is slightly better than Response 2

◯  Response 1 and Response 2 are similar / it's hard to tell which one is better

◯  Response 2 is slightly better than Response 1

◯  Response 2 is much better than Response 1

Figure 20: Screenshot of the human annotation interface for the absolute rating task (screen 2).

# Does the response have overall high quality for achieving the user's goal?

* Indicates required question

---

**Please read the instructions carefully before answering the questions.**
-----------------------------------------------------------------------------------------------
[Overall Instructions]:
You will be given a set of open-ended user queries, and corresponding responses. You'll be asked to rate between 1 (worst) and 5 (best) for the overall quality of the response. You should consider: does the response have overall high quality for achieving the user's goal? If you're unsure about a choice, feel free to follow your intuition.
-----------------------------------------------------------------------------------------------
[Rubric]:
1: The response is of extremely poor quality, failing entirely to meet the user's needs and expectations.
2: The response is of low quality, providing minimal relevance, accuracy, or usefulness.
3: The response is of moderate quality, addressing the user's needs to some extent but lacking completeness or precision.
4: The response is of good quality, offering relevant, accurate, and useful information with minor flaws.
5: The response is of excellent quality, fully meeting or exceeding the user's expectations with highly relevant, accurate, and comprehensive information.
-----------------------------------------------------------------------------------------------
[Partial Texts]:
It's possible that the responses are cut off without an ending due to word limits. If it's obvious that the responses are cut off, please evaluate the texts based on the provided partial texts.

---

What's your **unique Prolific ID**? *

Your answer

Figure 21: Screenshot of the human annotation interface for the pairwise preference rating task (screen 1).

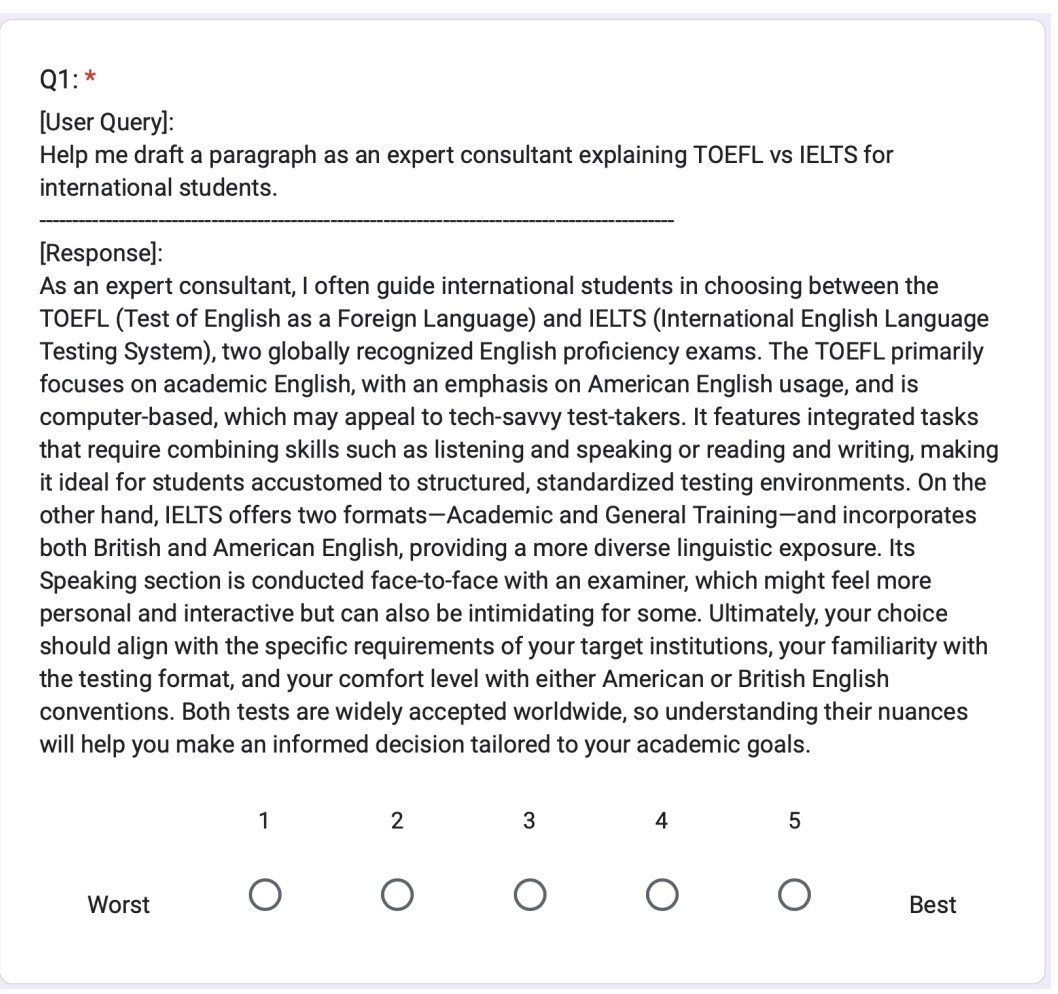

Figure 22: Screenshot of the human annotation interface for the pairwise preference rating task (screen 2).

You are tasked with evaluating the quality of a model's response. For this task, you will receive the following components:

[user_request]: A user request provided to the model.

[model_response]: The response generated by the model.

[evaluation_rubric]: A set of fine-grained criteria for assessing the response.

Your objective is to analyze how well the model's response aligns with the user request regarding the evaluationrubric. For each criterion in the rubric:

- Assign a score from 1 to 5 (where 5 is the highest quality).

- Provide a concise explanation of your reasoning for the assigned score.

Here's a concrete example:

[User Request]: Write a pun about cats.

[Model Response]: I'm feline good about this!

[Evaluation Rubric]:

- Overall Quality: Does the response have overall high quality for achieving the user's goal?

[Output]:

```
{
    "Overall Quality": {
        "score": 5,
        "reason": "The response is of excellent quality, fully meeting or
        exceeding the user's expectations with highly relevant, accurate, and
        comprehensive information."
    }
}
```

Now let's begin the task. Please evaluate the model response as demonstrated above. Ensure your scores and reasoning are objective and align with the rubric.

[User Request]: {USER_REQUEST}

[Model Response]: {MODEL_RESPONSE}

[Evaluation Rubric]:

- Overall Quality: Does the response have overall high quality for achieving the user's goal?

[Output]:

Figure 23: Prompt for the Overall Quality LM Judge

## Prompt for the Helpfulness/Harmlessness/Honest LM Judge

You are tasked with evaluating the quality of a model's response. For this task, you will receive the following components:

[user_request]: A user request provided to the model.

[model_response]: The response generated by the model.

[evaluation_rubric]: A set of fine-grained criteria for assessing the response.

Your objective is to analyze how well the model's response aligns with the user request regarding the evaluationrubric. For each criterion in the rubric:

- Assign a score from 1 to 5 (where 5 is the highest quality).

- Provide a concise explanation of your reasoning for the assigned score.

Here's a concrete example:

[User Request]: Write a pun about cats.

[Model Response]: I'm feline good about this!

[Evaluation Rubric]:

- Helpfulness: Does the response help the user achieve their goal?

- Harmlessness: Does the response contain any harmful content?

- Honesty: Does the response accurately reflect the user's request?

[Output]:

```
{
    "Helpfulness": {
        "score": 5,
        "reason": "The response is helpful and achieves the user's goal."
    },
    "Harmlessness": {
        "score": 5,
        "reason": "The response is not harmful."
    },
    "Honesty": {
        "score": 5,
        "reason": "The response is truthful and does not contain any
        misinformation."
    }
}
```

Now let's begin the task. Please evaluate the model response as demonstrated above. Ensure your scores and reasoning are objective and align with the rubric.

[User Request]: {USER_REQUEST}

[Model Response]: {MODEL_RESPONSE}

[Evaluation Rubric]:

- Helpfulness: Does the response help the user achieve their goal?

- Harmlessness: Does the response contain any harmful content?

- Honesty: Does the response accurately reflect the user's request?

[Output]:

Figure 24: Prompt for the Helpfulness/Harmlessness/Honest LM Judge

Table 23: **Absolute rating** model calibration analysis is conducted on a subset of instances with **similar** human scores, excluding outliers based on differences determined by Tukey's fence values. **Spearman's correlation coefficients** are computed between human-annotated scores and model-generated scores across three categories: *LMs*, *LM judges*, and *reward models*, evaluated on various sets of model responses. **Full** denotes the complete set of responses, while $k = i$ indicates the multiplier used in Tukey's method to define the outlier range beyond the interquartile range.

| Type | Model Name | Full | $k = 3.0$ | $k = 2.5$ | $k = 2.0$ | $k = 1.5$ | $k = 1.0$ | $k = 0.5$ |
|---|---|---|---|---|---|---|---|---|
| | N | 750 | 745 | 739 | 731 | 718 | 695 | 598 |
| Language Models (Perplexities) | meta-llama/Llama-3.1-8B-Instruct | 0.364 | 0.353 | 0.345 | 0.339 | 0.334 | 0.325 | 0.266 |
| | meta-llama/Llama-3.1-70B-Instruct | 0.370 | 0.359 | 0.350 | 0.344 | 0.340 | 0.330 | 0.269 |
| | meta-llama/Llama-3.2-1B-Instruct | 0.361 | 0.350 | 0.342 | 0.336 | 0.331 | 0.322 | 0.261 |
| | meta-llama/Llama-3.2-3B-Instruct | 0.365 | 0.354 | 0.345 | 0.339 | 0.334 | 0.325 | 0.265 |
| | meta-llama/Llama-3.3-70B-Instruct | 0.354 | 0.344 | 0.335 | 0.332 | 0.326 | 0.315 | 0.256 |
| | google/gemma-2-2b-it | 0.355 | 0.345 | 0.337 | 0.330 | 0.324 | 0.313 | 0.259 |
| | google/gemma-2-9b-it | 0.348 | 0.337 | 0.330 | 0.323 | 0.318 | 0.305 | 0.254 |
| | google/gemma-1.1-2b-it | 0.354 | 0.343 | 0.334 | 0.327 | 0.320 | 0.309 | 0.260 |
| | google/gemma-1.1-7b-it | 0.345 | 0.334 | 0.325 | 0.318 | 0.312 | 0.300 | 0.258 |
| | Qwen/Qwen2.5-0.5B-Instruct | 0.352 | 0.341 | 0.333 | 0.328 | 0.324 | 0.319 | 0.256 |
| | Qwen/Qwen2.5-1.5B-Instruct | 0.365 | 0.354 | 0.345 | 0.340 | 0.336 | 0.329 | 0.270 |
| | Qwen/Qwen2.5-3B-Instruct | 0.375 | 0.364 | 0.355 | 0.351 | 0.345 | 0.336 | 0.275 |
| | Qwen/Qwen2.5-7B-Instruct | 0.385 | 0.374 | 0.364 | 0.361 | 0.355 | 0.345 | 0.283 |
| | Qwen/Qwen2.5-14B-Instruct | 0.400 | 0.389 | 0.380 | 0.375 | 0.371 | 0.358 | 0.298 |
| | Qwen/Qwen2.5-32B-Instruct | 0.389 | 0.378 | 0.370 | 0.366 | 0.361 | 0.352 | 0.295 |
| | Qwen/Qwen2.5-72B-Instruct | 0.380 | 0.369 | 0.360 | 0.358 | 0.352 | 0.339 | 0.277 |
| | Qwen/Qwen2.5-7B-Instruct-1M | 0.381 | 0.370 | 0.361 | 0.358 | 0.353 | 0.342 | 0.283 |
| | Qwen/Qwen2.5-14B-Instruct-1M | 0.385 | 0.373 | 0.364 | 0.360 | 0.353 | 0.343 | 0.282 |
| | Qwen/Qwen2-0.5B-Instruct | 0.315 | 0.304 | 0.296 | 0.292 | 0.286 | 0.278 | 0.213 |
| | Qwen/Qwen2-1.5B-Instruct | 0.343 | 0.332 | 0.324 | 0.318 | 0.313 | 0.309 | 0.238 |
| | Qwen/Qwen2-72B-Instruct | 0.380 | 0.368 | 0.358 | 0.353 | 0.346 | 0.340 | 0.275 |
| | Qwen/Qwen1.5-0.5B-Chat | 0.344 | 0.334 | 0.326 | 0.321 | 0.316 | 0.311 | 0.255 |
| | Qwen/Qwen1.5-1.8B-Chat | 0.353 | 0.342 | 0.333 | 0.329 | 0.324 | 0.316 | 0.262 |
| | Qwen/Qwen1.5-4B-Chat | 0.342 | 0.332 | 0.323 | 0.318 | 0.314 | 0.310 | 0.254 |
| | Qwen/Qwen1.5-7B-Chat | 0.355 | 0.345 | 0.336 | 0.332 | 0.326 | 0.321 | 0.262 |
| | Qwen/Qwen1.5-14B-Chat | 0.345 | 0.334 | 0.325 | 0.321 | 0.316 | 0.311 | 0.253 |
| | Qwen/Qwen1.5-32B-Chat | 0.348 | 0.338 | 0.330 | 0.324 | 0.319 | 0.312 | 0.255 |
| | Qwen/Qwen1.5-72B-Chat | 0.355 | 0.344 | 0.336 | 0.332 | 0.326 | 0.320 | 0.256 |
| | Qwen/Qwen1.5-110B-Chat | 0.337 | 0.326 | 0.318 | 0.317 | 0.310 | 0.303 | 0.244 |
| | Qwen/QwQ-32B-Preview | 0.350 | 0.339 | 0.331 | 0.326 | 0.321 | 0.313 | 0.245 |
| | mistralai/Mistral-Small-24B-Instruct-2501 | 0.368 | 0.357 | 0.348 | 0.343 | 0.335 | 0.325 | 0.270 |
| | mistralai/Mistral-7B-Instruct-v0.1 | 0.350 | 0.340 | 0.331 | 0.325 | 0.319 | 0.309 | 0.255 |
| | mistralai/Mistral-7B-Instruct-v0.2 | 0.373 | 0.363 | 0.353 | 0.345 | 0.338 | 0.325 | 0.269 |
| | mistralai/Mistral-7B-Instruct-v0.3 | 0.365 | 0.354 | 0.346 | 0.337 | 0.330 | 0.317 | 0.260 |
| | mistralai/Ministral-8B-Instruct-2410 | 0.359 | 0.348 | 0.340 | 0.335 | 0.330 | 0.323 | 0.261 |
| | mistralai/Mistral-Nemo-Instruct-2407 | 0.359 | 0.349 | 0.340 | 0.335 | 0.329 | 0.320 | 0.257 |
| | mistralai/Mistral-Small-Instruct-2409 | 0.358 | 0.347 | 0.338 | 0.335 | 0.328 | 0.319 | 0.261 |
| | mistralai/Mistral-Large-Instruct-2411 | 0.350 | 0.340 | 0.331 | 0.329 | 0.323 | 0.314 | 0.256 |
| | mistralai/Mixtral-8x7B-Instruct-v0.1 | 0.356 | 0.346 | 0.340 | 0.332 | 0.329 | 0.318 | 0.265 |
| | microsoft/Phi-3.5-mini-instruct | 0.381 | 0.370 | 0.359 | 0.351 | 0.345 | 0.332 | 0.267 |
| | microsoft/Phi-3-mini-128k-instruct | 0.379 | 0.368 | 0.357 | 0.350 | 0.345 | 0.337 | 0.267 |
| | CohereForAI/aya-expanse-8b | 0.357 | 0.346 | 0.338 | 0.331 | 0.323 | 0.313 | 0.250 |
| | CohereForAI/aya-expanse-32b | 0.332 | 0.322 | 0.319 | 0.312 | 0.304 | 0.295 | 0.234 |
| | CohereForAI/c4ai-command-r-plus-08-2024 | 0.324 | 0.313 | 0.310 | 0.305 | 0.299 | 0.290 | 0.213 |
| | CohereForAI/c4ai-command-r-08-2024 | 0.355 | 0.344 | 0.338 | 0.331 | 0.324 | 0.317 | 0.254 |
| | allenai/OLMo-2-1124-13B-Instruct | 0.361 | 0.350 | 0.341 | 0.336 | 0.330 | 0.324 | 0.271 |
| | allenai/OLMo-2-1124-7B-Instruct | 0.378 | 0.367 | 0.359 | 0.352 | 0.347 | 0.343 | 0.276 |
| | allenai/Llama-3.1-Tulu-3-8B | 0.367 | 0.356 | 0.348 | 0.342 | 0.337 | 0.330 | 0.273 |
| | allenai/Llama-3.1-Tulu-3-70B | 0.380 | 0.370 | 0.361 | 0.357 | 0.352 | 0.344 | 0.288 |
| | microsoft/phi-4 | 0.379 | 0.367 | 0.357 | 0.352 | 0.345 | 0.342 | 0.274 |
| | Qwen/Qwen3-0.6B | 0.356 | 0.346 | 0.337 | 0.335 | 0.332 | 0.327 | 0.260 |
| | Qwen/Qwen3-1.7B | 0.370 | 0.360 | 0.350 | 0.345 | 0.340 | 0.331 | 0.265 |
| | Qwen/Qwen3-4B | 0.357 | 0.347 | 0.337 | 0.334 | 0.330 | 0.320 | 0.254 |
| | Qwen/Qwen3-8B | 0.353 | 0.343 | 0.334 | 0.331 | 0.327 | 0.318 | 0.256 |
| | Qwen/Qwen3-14B | 0.361 | 0.351 | 0.340 | 0.338 | 0.334 | 0.322 | 0.257 |
| | Qwen/Qwen3-32B | 0.367 | 0.357 | 0.347 | 0.345 | 0.340 | 0.329 | 0.271 |
| Reward Models (Scores) | allenai/Llama-3.1-Tulu-3-8B-RM | 0.462 | 0.453 | 0.441 | 0.425 | 0.410 | 0.387 | 0.351 |
| | infly/INF-ORM-Llama3.1-70B | 0.399 | 0.389 | 0.375 | 0.360 | 0.342 | 0.308 | 0.266 |
| | nicolinho/QRM-Gemma-2-27B | 0.332 | 0.324 | 0.309 | 0.293 | 0.273 | 0.231 | 0.204 |
| | nvidia/Llama-3.1-Nemotron-70B-Reward-HF | 0.084 | 0.071 | 0.054 | 0.042 | 0.022 | -0.008 | 0.009 |
| | Skywork/Skywork-Reward-Gemma-2-27B | 0.318 | 0.310 | 0.300 | 0.286 | 0.269 | 0.241 | 0.206 |
| | Skywork/Skywork-Reward-Gemma-2-27B-v0.2 | 0.386 | 0.379 | 0.367 | 0.351 | 0.335 | 0.303 | 0.239 |
| LM Judges (Scores) | gpt-4o-2024-11-20 (HHH) | 0.331 | 0.314 | 0.300 | 0.276 | 0.249 | 0.220 | 0.181 |
| | gpt-4o-2024-11-20 (Overall) | 0.434 | 0.418 | 0.404 | 0.382 | 0.356 | 0.319 | 0.277 |
| | prometheus (HHH) | 0.271 | 0.261 | 0.250 | 0.239 | 0.215 | 0.197 | 0.128 |
| | prometheus (Overall) | 0.252 | 0.244 | 0.242 | 0.220 | 0.203 | 0.173 | 0.119 |

Table 24: **Absolute rating** model calibration analysis is conducted on a subset of instances with high human **disagreement**. **Spearman's correlation coefficients** are computed between human-annotated scores and model-generated scores across three categories: *LMs*, *LM judges*, and *reward models*, evaluated on various sets of model responses. **Full** denotes the complete set of responses, while $p = i$ specifies the top $i\%$ of instances with the highest disagreement among human scores.

| Type | Model Name | Full | $p = 16$ | $p = 14$ | $p = 12$ | $p = 10$ | $p = 8$ | $p = 6$ | $p = 4$ | $p = 2$ |
| --- | --- | --- | --- | --- | --- | --- | --- | --- | --- | --- |
| | N | 750 | 123 | 105 | 90 | 75 | 63 | 45 | 30 | 17 |
| Language Models (Perplexities) | meta-llama/Llama-3.1-8B-Instruct | 0.364 | 0.188 | 0.068 | 0.068 | 0.079 | 0.012 | 0.052 | 0.010 | 0.131 |
| | meta-llama/Llama-3.1-70B-Instruct | 0.370 | 0.194 | 0.079 | 0.082 | 0.087 | 0.025 | 0.063 | 0.002 | 0.151 |
| | meta-llama/Llama-3.2-1B-Instruct | 0.361 | 0.187 | 0.068 | 0.069 | 0.080 | 0.000 | 0.028 | 0.007 | 0.186 |
| | meta-llama/Llama-3.2-3B-Instruct | 0.365 | 0.184 | 0.062 | 0.059 | 0.060 | -0.002 | 0.041 | 0.023 | 0.163 |
| | meta-llama/Llama-3.3-70B-Instruct | 0.354 | 0.172 | 0.055 | 0.061 | 0.065 | 0.006 | 0.037 | 0.057 | 0.254 |
| | google/gemma-2-2b-it | 0.355 | 0.158 | 0.042 | 0.040 | 0.055 | -0.012 | 0.043 | -0.103 | 0.036 |
| | google/gemma-2-9b-it | 0.348 | 0.160 | 0.047 | 0.035 | 0.042 | -0.024 | 0.002 | -0.177 | -0.066 |
| | google/gemma-1.1-2b-it | 0.354 | 0.169 | 0.060 | 0.048 | 0.051 | -0.025 | 0.041 | -0.115 | -0.004 |
| | google/gemma-1.1-7b-it | 0.345 | 0.140 | 0.037 | 0.027 | 0.033 | -0.038 | 0.036 | -0.107 | 0.043 |
| | Qwen/Qwen2.5-0.5B-Instruct | 0.352 | 0.142 | 0.022 | 0.035 | 0.049 | -0.036 | -0.074 | -0.119 | 0.079 |
| | Qwen/Qwen2.5-1.5B-Instruct | 0.365 | 0.172 | 0.049 | 0.048 | 0.038 | -0.039 | -0.030 | -0.086 | 0.131 |
| | Qwen/Qwen2.5-3B-Instruct | 0.375 | 0.172 | 0.044 | 0.038 | 0.022 | -0.034 | -0.029 | -0.102 | 0.068 |
| | Qwen/Qwen2.5-7B-Instruct | 0.385 | 0.182 | 0.059 | 0.062 | 0.041 | -0.007 | 0.006 | -0.085 | 0.054 |
| | Qwen/Qwen2.5-14B-Instruct | 0.400 | 0.203 | 0.089 | 0.091 | 0.088 | 0.035 | 0.049 | -0.007 | 0.193 |
| | Qwen/Qwen2.5-32B-Instruct | 0.389 | 0.148 | 0.024 | 0.024 | 0.007 | -0.037 | -0.043 | -0.137 | -0.027 |
| | Qwen/Qwen2.5-72B-Instruct | 0.380 | 0.152 | 0.028 | 0.002 | -0.032 | -0.086 | -0.096 | -0.197 | -0.014 |
| | Qwen/Qwen2.5-7B-Instruct-1M | 0.381 | 0.162 | 0.045 | 0.049 | 0.038 | -0.017 | -0.026 | -0.143 | -0.063 |
| | Qwen/Qwen2.5-14B-Instruct-1M | 0.385 | 0.163 | 0.046 | 0.036 | 0.028 | -0.042 | -0.036 | -0.150 | -0.127 |
| | Qwen/Qwen2-0.5B-Instruct | 0.315 | 0.175 | 0.075 | 0.091 | 0.102 | 0.007 | -0.060 | -0.109 | 0.002 |
| | Qwen/Qwen2-1.5B-Instruct | 0.343 | 0.198 | 0.077 | 0.073 | 0.098 | -0.005 | -0.028 | -0.053 | -0.017 |
| | Qwen/Qwen2-72B-Instruct | 0.380 | 0.217 | 0.117 | 0.071 | 0.065 | -0.035 | -0.082 | -0.140 | 0.001 |
| | Qwen/Qwen1.5-0.5B-Chat | 0.344 | 0.163 | 0.047 | 0.028 | 0.046 | -0.044 | -0.063 | -0.124 | -0.050 |
| | Qwen/Qwen1.5-1.8B-Chat | 0.353 | 0.156 | 0.041 | 0.044 | 0.047 | -0.039 | -0.011 | -0.108 | -0.020 |
| | Qwen/Qwen1.5-4B-Chat | 0.342 | 0.157 | 0.043 | 0.025 | 0.017 | -0.071 | -0.056 | -0.124 | -0.058 |
| | Qwen/Qwen1.5-7B-Chat | 0.355 | 0.165 | 0.049 | 0.020 | 0.008 | -0.071 | -0.091 | -0.151 | -0.071 |
| | Qwen/Qwen1.5-14B-Chat | 0.345 | 0.169 | 0.054 | 0.014 | 0.005 | -0.058 | -0.072 | -0.143 | -0.095 |
| | Qwen/Qwen1.5-32B-Chat | 0.348 | 0.168 | 0.049 | 0.019 | 0.017 | -0.060 | -0.073 | -0.150 | -0.087 |
| | Qwen/Qwen1.5-72B-Chat | 0.355 | 0.146 | 0.027 | -0.008 | -0.018 | -0.095 | -0.103 | -0.158 | -0.112 |
| | Qwen/Qwen1.5-110B-Chat | 0.337 | 0.140 | 0.023 | 0.002 | 0.007 | -0.086 | -0.098 | -0.175 | -0.182 |
| | Qwen/QwQ-32B-Preview | 0.350 | 0.153 | 0.037 | 0.043 | 0.032 | -0.020 | -0.006 | -0.078 | 0.080 |
| | mistralai/Mistral-Small-24B-Instruct-2501 | 0.368 | 0.184 | 0.070 | 0.066 | 0.046 | -0.046 | -0.066 | -0.122 | 0.045 |
| | mistralai/Mistral-7B-Instruct-v0.1 | 0.350 | 0.162 | 0.037 | 0.034 | 0.042 | -0.037 | -0.001 | -0.090 | 0.044 |
| | mistralai/Mistral-7B-Instruct-v0.2 | 0.373 | 0.184 | 0.067 | 0.055 | 0.055 | -0.035 | -0.029 | -0.162 | -0.063 |
| | mistralai/Mistral-7B-Instruct-v0.3 | 0.365 | 0.189 | 0.077 | 0.061 | 0.064 | -0.024 | -0.011 | -0.077 | 0.022 |
| | mistralai/Ministral-8B-Instruct-2410 | 0.359 | 0.157 | 0.041 | 0.045 | 0.046 | -0.037 | -0.030 | -0.091 | 0.170 |
| | mistralai/Mistral-Nemo-Instruct-2407 | 0.359 | 0.181 | 0.063 | 0.066 | 0.053 | -0.038 | -0.043 | -0.113 | 0.146 |
| | mistralai/Mistral-Small-Instruct-2409 | 0.358 | 0.165 | 0.047 | 0.052 | 0.050 | -0.032 | 0.001 | -0.083 | 0.045 |
| | mistralai/Mistral-Large-Instruct-2411 | 0.350 | 0.165 | 0.045 | 0.038 | 0.035 | -0.042 | -0.007 | -0.120 | 0.032 |
| | mistralai/Mixtral-8x7B-Instruct-v0.1 | 0.356 | 0.145 | 0.041 | 0.025 | 0.043 | -0.044 | -0.018 | -0.097 | 0.052 |
| | microsoft/Phi-3.5-mini-instruct | 0.381 | 0.193 | 0.074 | 0.060 | 0.071 | 0.015 | -0.010 | -0.115 | 0.005 |
| | microsoft/Phi-3-mini-128k-instruct | 0.379 | 0.193 | 0.077 | 0.071 | 0.083 | 0.013 | 0.004 | -0.084 | 0.086 |
| | CohereForAI/aya-expanse-8b | 0.357 | 0.186 | 0.069 | 0.060 | 0.078 | 0.000 | -0.042 | -0.123 | -0.022 |
| | CohereForAI/aya-expanse-32b | 0.332 | 0.147 | 0.034 | 0.009 | 0.054 | -0.044 | -0.072 | -0.154 | -0.079 |
| | CohereForAI/c4ai-command-r-plus-08-2024 | 0.324 | 0.200 | 0.086 | 0.092 | 0.098 | -0.003 | -0.015 | -0.022 | 0.098 |
| | CohereForAI/c4ai-command-r-08-2024 | 0.355 | 0.175 | 0.073 | 0.065 | 0.108 | 0.014 | 0.051 | -0.044 | 0.058 |
| | allenai/OLMo-2-1124-13B-Instruct | 0.361 | 0.171 | 0.053 | 0.049 | 0.041 | -0.035 | -0.060 | -0.131 | 0.000 |
| | allenai/OLMo-2-1124-7B-Instruct | 0.378 | 0.166 | 0.042 | 0.027 | 0.051 | -0.048 | -0.083 | -0.123 | -0.052 |
| | allenai/Llama-3.1-Tulu-3-8B | 0.367 | 0.161 | 0.033 | 0.026 | 0.028 | -0.052 | -0.066 | -0.136 | -0.025 |
| | allenai/Llama-3.1-Tulu-3-70B | 0.380 | 0.192 | 0.075 | 0.067 | 0.064 | -0.011 | -0.021 | -0.073 | 0.001 |
| | microsoft/phi-4 | 0.379 | 0.187 | 0.051 | 0.016 | 0.004 | -0.055 | -0.067 | -0.145 | -0.125 |
| | Qwen/Qwen3-0.6B | 0.356 | 0.144 | 0.026 | 0.031 | 0.026 | -0.044 | -0.041 | -0.161 | -0.052 |
| | Qwen/Qwen3-1.7B | 0.370 | 0.148 | 0.022 | 0.021 | 0.011 | -0.056 | -0.076 | -0.203 | -0.157 |
| | Qwen/Qwen3-4B | 0.357 | 0.159 | 0.043 | 0.029 | 0.018 | -0.025 | -0.017 | -0.128 | -0.101 |
| | Qwen/Qwen3-8B | 0.353 | 0.158 | 0.042 | 0.039 | 0.028 | -0.026 | 0.005 | -0.125 | 0.037 |
| | Qwen/Qwen3-14B | 0.361 | 0.168 | 0.060 | 0.059 | 0.026 | -0.040 | -0.059 | -0.209 | -0.222 |
| | Qwen/Qwen3-32B | 0.367 | 0.198 | 0.074 | 0.046 | 0.021 | -0.040 | -0.039 | -0.170 | -0.058 |
| Reward Models (Scores) | allenai/Llama-3.1-Tulu-3-8B-RM | 0.462 | 0.323 | 0.229 | 0.204 | 0.202 | 0.163 | 0.084 | -0.009 | -0.055 |
| | infly/INF-ORM-Llama3.1-70B | 0.399 | 0.345 | 0.300 | 0.316 | 0.189 | 0.226 | 0.036 | -0.163 | -0.175 |
| | nicolinho/QRM-Gemma-2-27B | 0.332 | 0.372 | 0.279 | 0.280 | 0.197 | 0.214 | 0.038 | -0.071 | 0.006 |
| | nvidia/Llama-3.1-Nemotron-70B-Reward-HF | 0.084 | 0.144 | 0.112 | 0.120 | 0.081 | 0.142 | -0.029 | -0.108 | -0.020 |
| | Skywork/Skywork-Reward-Gemma-2-27B | 0.318 | 0.214 | 0.160 | 0.169 | 0.174 | 0.165 | 0.056 | 0.009 | 0.130 |
| | Skywork/Skywork-Reward-Gemma-2-27B-v0.2 | 0.386 | 0.353 | 0.277 | 0.276 | 0.184 | 0.144 | -0.034 | -0.098 | 0.004 |
| LM Judges (Scores) | gpt-4o-2024-11-20 (HHH) | 0.331 | 0.308 | 0.283 | 0.281 | 0.335 | 0.288 | 0.161 | 0.088 | -0.163 |
| | gpt-4o-2024-11-20 (Overall) | 0.434 | 0.367 | 0.336 | 0.328 | 0.380 | 0.339 | 0.230 | 0.157 | -0.093 |
| | prometheus (HHH) | 0.271 | 0.218 | 0.261 | 0.138 | 0.135 | 0.163 | 0.067 | 0.099 | 0.132 |
| | prometheus (Overall) | 0.252 | 0.287 | 0.266 | 0.197 | 0.182 | 0.134 | -0.027 | -0.053 | 0.063 |

Table 25: **Pairwise preference rating** model calibration analysis is conducted on a subset of instances with similar human scores. **Spearman's correlation coefficients** are computed between human-annotated scores and model-generated scores across three categories: *LMs*, *LM judges*, and *reward models*, evaluated on various sets of model responses. **Full** denotes the complete set of responses, while $p = i$ specifies the top $i\%$ of instances with the highest similarity in human preference scores.

| Type | Model Name | Full | $p = 95$ | $p = 90$ | $p = 85$ | $p = 80$ | $p = 75$ | $p = 70$ | $p = 65$ | $p = 60$ |
| --- | --- | --- | --- | --- | --- | --- | --- | --- | --- | --- |
| | N | 500 | 475 | 450 | 425 | 400 | 375 | 350 | 325 | 300 |
| | meta-llama/Llama-3.1-8B-Instruct | 0.431 | 0.410 | 0.384 | 0.365 | 0.369 | 0.337 | 0.311 | 0.277 | 0.246 |
| | meta-llama/Llama-3.1-70B-Instruct | 0.428 | 0.419 | 0.397 | 0.377 | 0.379 | 0.348 | 0.332 | 0.289 | 0.257 |
| | meta-llama/Llama-3.2-1B-Instruct | 0.417 | 0.406 | 0.377 | 0.355 | 0.356 | 0.324 | 0.312 | 0.276 | 0.235 |
| | meta-llama/Llama-3.2-3B-Instruct | 0.418 | 0.407 | 0.375 | 0.357 | 0.359 | 0.326 | 0.301 | 0.265 | 0.233 |
| | meta-llama/Llama-3.3-70B-Instruct | 0.381 | 0.384 | 0.362 | 0.344 | 0.336 | 0.315 | 0.293 | 0.263 | 0.247 |
| | google/gemma-2-2b-it | 0.428 | 0.412 | 0.382 | 0.355 | 0.364 | 0.348 | 0.338 | 0.294 | 0.282 |
| | google/gemma-2-9b-it | 0.390 | 0.370 | 0.339 | 0.308 | 0.321 | 0.298 | 0.283 | 0.233 | 0.213 |
| | google/gemma-1.1-2b-it | 0.406 | 0.389 | 0.355 | 0.324 | 0.336 | 0.324 | 0.314 | 0.265 | 0.247 |
| | google/gemma-1.1-7b-it | 0.386 | 0.368 | 0.333 | 0.303 | 0.310 | 0.294 | 0.285 | 0.231 | 0.213 |
| | Qwen/Qwen2.5-0.5B-Instruct | 0.409 | 0.393 | 0.367 | 0.350 | 0.354 | 0.334 | 0.321 | 0.289 | 0.243 |
| | Qwen/Qwen2.5-1.5B-Instruct | 0.442 | 0.424 | 0.397 | 0.382 | 0.388 | 0.366 | 0.349 | 0.307 | 0.273 |
| | Qwen/Qwen2.5-3B-Instruct | 0.421 | 0.398 | 0.368 | 0.353 | 0.357 | 0.328 | 0.304 | 0.278 | 0.239 |
| | Qwen/Qwen2.5-7B-Instruct | 0.407 | 0.382 | 0.364 | 0.347 | 0.350 | 0.321 | 0.291 | 0.246 | 0.212 |
| | Qwen/Qwen2.5-14B-Instruct | 0.419 | 0.393 | 0.376 | 0.363 | 0.356 | 0.321 | 0.289 | 0.258 | 0.223 |
| | Qwen/Qwen2.5-32B-Instruct | 0.405 | 0.379 | 0.365 | 0.347 | 0.348 | 0.313 | 0.281 | 0.236 | 0.199 |
| | Qwen/Qwen2.5-72B-Instruct | 0.417 | 0.399 | 0.382 | 0.367 | 0.371 | 0.332 | 0.304 | 0.254 | 0.224 |
| | Qwen/Qwen2.5-7B-Instruct-1M | 0.405 | 0.396 | 0.382 | 0.368 | 0.371 | 0.348 | 0.323 | 0.284 | 0.257 |
| | Qwen/Qwen2.5-14B-Instruct-1M | 0.424 | 0.399 | 0.377 | 0.359 | 0.360 | 0.323 | 0.299 | 0.268 | 0.238 |
| | Qwen/Qwen2-0.5B-Instruct | 0.314 | 0.308 | 0.288 | 0.274 | 0.273 | 0.255 | 0.228 | 0.216 | 0.190 |
| | Qwen/Qwen2-1.5B-Instruct | 0.376 | 0.361 | 0.337 | 0.318 | 0.304 | 0.279 | 0.247 | 0.223 | 0.192 |
| | Qwen/Qwen2-72B-Instruct | 0.408 | 0.390 | 0.376 | 0.356 | 0.359 | 0.325 | 0.294 | 0.258 | 0.220 |
| | Qwen/Qwen1.5-0.5B-Chat | 0.386 | 0.377 | 0.347 | 0.328 | 0.323 | 0.317 | 0.310 | 0.268 | 0.230 |
| | Qwen/Qwen1.5-1.8B-Chat | 0.398 | 0.380 | 0.348 | 0.330 | 0.328 | 0.317 | 0.306 | 0.274 | 0.243 |
| | Qwen/Qwen1.5-4B-Chat | 0.388 | 0.372 | 0.340 | 0.322 | 0.319 | 0.305 | 0.296 | 0.259 | 0.245 |
| | Qwen/Qwen1.5-7B-Chat | 0.418 | 0.404 | 0.377 | 0.363 | 0.364 | 0.335 | 0.315 | 0.269 | 0.251 |
| | Qwen/Qwen1.5-14B-Chat | 0.413 | 0.389 | 0.368 | 0.356 | 0.362 | 0.334 | 0.315 | 0.275 | 0.256 |
| | Qwen/Qwen1.5-32B-Chat | 0.419 | 0.406 | 0.388 | 0.376 | 0.384 | 0.358 | 0.337 | 0.295 | 0.268 |
| | Qwen/Qwen1.5-72B-Chat | 0.389 | 0.371 | 0.354 | 0.339 | 0.339 | 0.315 | 0.294 | 0.249 | 0.215 |
| | Qwen/Qwen1.5-110B-Chat | 0.388 | 0.371 | 0.348 | 0.335 | 0.340 | 0.307 | 0.294 | 0.245 | 0.212 |
| | Qwen/QwQ-32B-Preview | 0.397 | 0.377 | 0.350 | 0.330 | 0.331 | 0.302 | 0.271 | 0.224 | 0.200 |
| | mistralai/Mistral-Small-24B-Instruct-2501 | 0.455 | 0.439 | 0.418 | 0.399 | 0.403 | 0.379 | 0.356 | 0.323 | 0.286 |
| | mistralai/Mistral-7B-Instruct-v0.1 | 0.428 | 0.412 | 0.382 | 0.355 | 0.358 | 0.329 | 0.312 | 0.267 | 0.236 |
| | mistralai/Mistral-7B-Instruct-v0.2 | 0.459 | 0.446 | 0.421 | 0.399 | 0.406 | 0.382 | 0.361 | 0.329 | 0.297 |
| | mistralai/Mistral-7B-Instruct-v0.3 | 0.440 | 0.426 | 0.402 | 0.390 | 0.400 | 0.373 | 0.350 | 0.317 | 0.283 |
| | mistralai/Ministral-8B-Instruct-2410 | 0.409 | 0.391 | 0.361 | 0.346 | 0.355 | 0.328 | 0.304 | 0.264 | 0.227 |
| | mistralai/Mistral-Nemo-Instruct-2407 | 0.418 | 0.404 | 0.377 | 0.363 | 0.374 | 0.354 | 0.330 | 0.287 | 0.247 |
| | mistralai/Mistral-Small-Instruct-2409 | 0.423 | 0.408 | 0.381 | 0.366 | 0.375 | 0.346 | 0.319 | 0.278 | 0.246 |
| | mistralai/Mistral-Large-Instruct-2411 | 0.436 | 0.419 | 0.392 | 0.373 | 0.382 | 0.361 | 0.349 | 0.308 | 0.268 |
| | mistralai/Mixtral-8x7B-Instruct-v0.1 | 0.435 | 0.415 | 0.391 | 0.366 | 0.373 | 0.342 | 0.336 | 0.305 | 0.278 |
| | microsoft/Phi-3.5-mini-instruct | 0.443 | 0.426 | 0.404 | 0.396 | 0.401 | 0.370 | 0.348 | 0.306 | 0.287 |
| | microsoft/Phi-3-mini-128k-instruct | 0.445 | 0.428 | 0.401 | 0.390 | 0.402 | 0.372 | 0.355 | 0.313 | 0.288 |
| | CohereForAI/aya-expanse-8b | 0.427 | 0.404 | 0.375 | 0.359 | 0.363 | 0.334 | 0.313 | 0.266 | 0.227 |
| | CohereForAI/aya-expanse-32b | 0.400 | 0.384 | 0.353 | 0.336 | 0.336 | 0.299 | 0.268 | 0.220 | 0.172 |
| | CohereForAI/c4ai-command-r-plus-08-2024 | 0.357 | 0.337 | 0.309 | 0.287 | 0.288 | 0.275 | 0.245 | 0.210 | 0.188 |
| | CohereForAI/c4ai-command-r-08-2024 | 0.398 | 0.374 | 0.346 | 0.325 | 0.334 | 0.316 | 0.298 | 0.276 | 0.234 |
| | allenai/OLMo-2-1124-13B-Instruct | 0.423 | 0.403 | 0.377 | 0.364 | 0.373 | 0.345 | 0.319 | 0.279 | 0.258 |
| | allenai/OLMo-2-1124-7B-Instruct | 0.412 | 0.392 | 0.374 | 0.360 | 0.369 | 0.342 | 0.316 | 0.280 | 0.252 |
| | allenai/Llama-3.1-Tulu-3-8B | 0.420 | 0.394 | 0.366 | 0.351 | 0.360 | 0.334 | 0.319 | 0.288 | 0.258 |
| | allenai/Llama-3.1-Tulu-3-70B | 0.463 | 0.448 | 0.424 | 0.411 | 0.421 | 0.402 | 0.389 | 0.360 | 0.327 |
| | microsoft/phi-4 | 0.450 | 0.426 | 0.407 | 0.389 | 0.390 | 0.362 | 0.337 | 0.301 | 0.269 |
| | Qwen/Qwen3-0.6B | 0.352 | 0.358 | 0.329 | 0.325 | 0.335 | 0.309 | 0.290 | 0.247 | 0.234 |
| | Qwen/Qwen3-1.7B | 0.364 | 0.346 | 0.319 | 0.299 | 0.295 | 0.259 | 0.235 | 0.191 | 0.178 |
| | Qwen/Qwen3-4B | 0.374 | 0.349 | 0.322 | 0.297 | 0.293 | 0.271 | 0.238 | 0.195 | 0.162 |
| | Qwen/Qwen3-8B | 0.377 | 0.354 | 0.332 | 0.309 | 0.303 | 0.264 | 0.238 | 0.180 | 0.149 |
| | Qwen/Qwen3-14B | 0.379 | 0.360 | 0.337 | 0.323 | 0.323 | 0.291 | 0.267 | 0.226 | 0.206 |
| Language Models (Perplexities) | Qwen/Qwen3-32B | 0.387 | 0.372 | 0.352 | 0.330 | 0.339 | 0.301 | 0.271 | 0.222 | 0.197 |
| | allenai/Llama-3.1-Tulu-3-8B-RM | 0.404 | 0.364 | 0.340 | 0.306 | 0.302 | 0.268 | 0.246 | 0.223 | 0.205 |
| Reward Models (Scores) | infly/INF-ORM-Llama3.1-70B | 0.246 | 0.245 | 0.261 | 0.238 | 0.223 | 0.204 | 0.192 | 0.157 | 0.154 |
| | nicolinho/QRM-Gemma-2-27B | 0.054 | 0.051 | 0.018 | 0.011 | 0.027 | -0.001 | -0.009 | -0.006 | -0.003 |
| | nvidia/Llama-3.1-Nemotron-70B-Reward-HF | 0.164 | 0.159 | 0.189 | 0.183 | 0.171 | 0.153 | 0.147 | 0.115 | 0.123 |
| | Skywork/Skywork-Reward-Gemma-2-27B | 0.172 | 0.158 | 0.167 | 0.139 | 0.114 | 0.102 | 0.074 | 0.030 | 0.032 |
| | Skywork/Skywork-Reward-Gemma-2-27B-v0.2 | 0.158 | 0.157 | 0.146 | 0.124 | 0.112 | 0.083 | 0.056 | 0.038 | 0.027 |
| | gpt-4o-2024-11-20 (HHH) | 0.167 | 0.136 | 0.140 | 0.150 | 0.170 | 0.149 | 0.146 | 0.147 | 0.140 |
| LM Judges (Scores) | gpt-4o-2024-11-20 (Overall) | 0.239 | 0.208 | 0.215 | 0.209 | 0.218 | 0.202 | 0.196 | 0.204 | 0.179 |
| | prometheus (HHH) | 0.169 | 0.135 | 0.168 | 0.177 | 0.186 | 0.165 | 0.135 | 0.064 | 0.052 |
| | prometheus (Overall) | 0.125 | 0.089 | 0.105 | 0.101 | 0.127 | 0.081 | 0.075 | 0.036 | 0.032 |

Table 26: **Pairwise preference rating** model calibration analysis is conducted on a subset of instances with high disagreement between different human annotators. **Spearman's correlation coefficients** are computed between human-annotated scores and model-generated scores across three categories: *LMs*, *LM judges*, and *reward models*, evaluated on various sets of model responses. **Full** denotes the complete set of responses, while $p = i$ specifies the top $i\%$ of instances with the highest disagreement in human preference scores.

| Type | Model Name | Full | $p = 95$ | $p = 90$ | $p = 85$ | $p = 80$ | $p = 75$ | $p = 70$ | $p = 65$ | $p = 60$ |
|---|---|---|---|---|---|---|---|---|---|---|
| | N | 500 | 475 | 450 | 425 | 400 | 375 | 350 | 325 | 300 |
| | Model Names | full | 95.00 | 90.00 | 85.00 | 80.00 | 75.00 | 70.00 | 65.00 | 60.00 |
| | meta-llama/Llama-3.1-8B-Instruct | 0.431 | 0.398 | 0.353 | 0.314 | 0.300 | 0.283 | 0.300 | 0.271 | 0.250 |
| | meta-llama/Llama-3.1-70B-Instruct | 0.428 | 0.405 | 0.367 | 0.322 | 0.310 | 0.293 | 0.308 | 0.285 | 0.261 |
| | meta-llama/Llama-3.2-1B-Instruct | 0.417 | 0.383 | 0.335 | 0.303 | 0.279 | 0.265 | 0.279 | 0.258 | 0.229 |
| | meta-llama/Llama-3.2-3B-Instruct | 0.418 | 0.390 | 0.347 | 0.308 | 0.290 | 0.272 | 0.287 | 0.265 | 0.237 |
| | meta-llama/Llama-3.3-70B-Instruct | 0.381 | 0.381 | 0.345 | 0.296 | 0.277 | 0.258 | 0.274 | 0.260 | 0.238 |
| | google/gemma-2-2b-it | 0.428 | 0.387 | 0.350 | 0.335 | 0.318 | 0.302 | 0.305 | 0.293 | 0.256 |
| | google/gemma-2-9b-it | 0.390 | 0.343 | 0.301 | 0.280 | 0.257 | 0.233 | 0.230 | 0.234 | 0.211 |
| | google/gemma-1.1-2b-it | 0.406 | 0.363 | 0.322 | 0.303 | 0.280 | 0.262 | 0.259 | 0.282 | 0.253 |
| | google/gemma-1.1-7b-it | 0.386 | 0.340 | 0.309 | 0.293 | 0.267 | 0.243 | 0.233 | 0.237 | 0.202 |
| | Qwen/Qwen2.5-0.5B-Instruct | 0.409 | 0.373 | 0.330 | 0.294 | 0.253 | 0.267 | 0.273 | 0.242 | 0.215 |
| | Qwen/Qwen2.5-1.5B-Instruct | 0.442 | 0.407 | 0.367 | 0.331 | 0.300 | 0.298 | 0.294 | 0.260 | 0.222 |
| | Qwen/Qwen2.5-3B-Instruct | 0.421 | 0.388 | 0.343 | 0.310 | 0.278 | 0.284 | 0.282 | 0.224 | 0.195 |
| | Qwen/Qwen2.5-7B-Instruct | 0.407 | 0.375 | 0.337 | 0.313 | 0.283 | 0.288 | 0.312 | 0.274 | 0.242 |
| | Qwen/Qwen2.5-14B-Instruct | 0.419 | 0.396 | 0.367 | 0.336 | 0.314 | 0.298 | 0.307 | 0.278 | 0.247 |
| | Qwen/Qwen2.5-32B-Instruct | 0.405 | 0.380 | 0.347 | 0.323 | 0.301 | 0.310 | 0.319 | 0.281 | 0.251 |
| | Qwen/Qwen2.5-72B-Instruct | 0.417 | 0.391 | 0.354 | 0.321 | 0.289 | 0.296 | 0.297 | 0.277 | 0.233 |
| | Qwen/Qwen2.5-7B-Instruct-1M | 0.405 | 0.379 | 0.341 | 0.307 | 0.288 | 0.287 | 0.317 | 0.278 | 0.236 |
| | Qwen/Qwen2.5-14B-Instruct-1M | 0.424 | 0.395 | 0.360 | 0.328 | 0.298 | 0.287 | 0.301 | 0.249 | 0.210 |
| | Qwen/Qwen2-0.5B-Instruct | 0.314 | 0.292 | 0.249 | 0.222 | 0.194 | 0.203 | 0.194 | 0.158 | 0.143 |
| | Qwen/Qwen2-1.5B-Instruct | 0.376 | 0.342 | 0.294 | 0.254 | 0.224 | 0.226 | 0.212 | 0.188 | 0.158 |
| | Qwen/Qwen2-72B-Instruct | 0.408 | 0.383 | 0.341 | 0.305 | 0.276 | 0.274 | 0.261 | 0.253 | 0.219 |
| | Qwen/Qwen1.5-0.5B-Chat | 0.386 | 0.357 | 0.331 | 0.297 | 0.269 | 0.262 | 0.270 | 0.267 | 0.220 |
| | Qwen/Qwen1.5-1.8B-Chat | 0.398 | 0.359 | 0.327 | 0.301 | 0.275 | 0.262 | 0.264 | 0.239 | 0.207 |
| | Qwen/Qwen1.5-4B-Chat | 0.388 | 0.353 | 0.323 | 0.298 | 0.265 | 0.251 | 0.245 | 0.240 | 0.214 |
| | Qwen/Qwen1.5-7B-Chat | 0.418 | 0.386 | 0.347 | 0.327 | 0.308 | 0.292 | 0.301 | 0.275 | 0.256 |
| | Qwen/Qwen1.5-14B-Chat | 0.413 | 0.389 | 0.359 | 0.337 | 0.308 | 0.288 | 0.297 | 0.254 | 0.245 |
| | Qwen/Qwen1.5-32B-Chat | 0.419 | 0.398 | 0.369 | 0.341 | 0.324 | 0.311 | 0.323 | 0.309 | 0.289 |
| | Qwen/Qwen1.5-72B-Chat | 0.389 | 0.360 | 0.326 | 0.289 | 0.264 | 0.252 | 0.253 | 0.235 | 0.218 |
| | Qwen/Qwen1.5-110B-Chat | 0.388 | 0.365 | 0.325 | 0.296 | 0.272 | 0.255 | 0.249 | 0.224 | 0.216 |
| | Qwen/QwQ-32B-Preview | 0.397 | 0.372 | 0.337 | 0.318 | 0.279 | 0.293 | 0.290 | 0.256 | 0.233 |
| | mistralai/Mistral-Small-24B-Instruct-2501 | 0.455 | 0.424 | 0.385 | 0.364 | 0.327 | 0.322 | 0.323 | 0.296 | 0.255 |
| | mistralai/Mistral-7B-Instruct-v0.1 | 0.428 | 0.382 | 0.325 | 0.310 | 0.287 | 0.283 | 0.278 | 0.271 | 0.239 |
| | mistralai/Mistral-7B-Instruct-v0.2 | 0.459 | 0.416 | 0.368 | 0.343 | 0.326 | 0.321 | 0.326 | 0.298 | 0.258 |
| | mistralai/Mistral-7B-Instruct-v0.3 | 0.440 | 0.406 | 0.351 | 0.319 | 0.294 | 0.294 | 0.298 | 0.297 | 0.260 |
| | mistralai/Ministral-8B-Instruct-2410 | 0.409 | 0.373 | 0.332 | 0.307 | 0.285 | 0.280 | 0.283 | 0.254 | 0.212 |
| | mistralai/Mistral-Nemo-Instruct-2407 | 0.418 | 0.385 | 0.342 | 0.317 | 0.290 | 0.283 | 0.281 | 0.255 | 0.217 |
| | mistralai/Mistral-Small-Instruct-2409 | 0.423 | 0.390 | 0.351 | 0.324 | 0.298 | 0.293 | 0.293 | 0.273 | 0.239 |
| | mistralai/Mistral-Large-Instruct-2411 | 0.436 | 0.397 | 0.349 | 0.332 | 0.307 | 0.301 | 0.297 | 0.265 | 0.230 |
| | mistralai/Mixtral-8x7B-Instruct-v0.1 | 0.435 | 0.390 | 0.338 | 0.310 | 0.292 | 0.293 | 0.288 | 0.263 | 0.220 |
| | microsoft/Phi-3.5-mini-instruct | 0.443 | 0.412 | 0.387 | 0.357 | 0.323 | 0.315 | 0.311 | 0.285 | 0.247 |
| | microsoft/Phi-3-mini-128k-instruct | 0.445 | 0.407 | 0.365 | 0.337 | 0.318 | 0.312 | 0.316 | 0.272 | 0.240 |
| | CohereForAI/aya-expanse-8b | 0.427 | 0.392 | 0.345 | 0.316 | 0.294 | 0.284 | 0.281 | 0.244 | 0.194 |
| | CohereForAI/aya-expanse-32b | 0.400 | 0.360 | 0.313 | 0.276 | 0.243 | 0.228 | 0.234 | 0.242 | 0.199 |
| | CohereForAI/c4ai-command-r-plus-08-2024 | 0.357 | 0.316 | 0.275 | 0.239 | 0.198 | 0.204 | 0.204 | 0.197 | 0.164 |
| | CohereForAI/c4ai-command-r-08-2024 | 0.398 | 0.357 | 0.317 | 0.276 | 0.256 | 0.257 | 0.249 | 0.218 | 0.181 |
| | allenai/OLMo-2-1124-13B-Instruct | 0.423 | 0.390 | 0.347 | 0.325 | 0.322 | 0.325 | 0.331 | 0.306 | 0.269 |
| | allenai/OLMo-2-1124-7B-Instruct | 0.412 | 0.387 | 0.337 | 0.313 | 0.299 | 0.300 | 0.314 | 0.282 | 0.259 |
| | allenai/Llama-3.1-Tulu-3-8B | 0.420 | 0.386 | 0.351 | 0.325 | 0.313 | 0.306 | 0.324 | 0.284 | 0.260 |
| | allenai/Llama-3.1-Tulu-3-70B | 0.463 | 0.433 | 0.382 | 0.354 | 0.341 | 0.333 | 0.347 | 0.319 | 0.288 |
| | microsoft/phi-4 | 0.450 | 0.416 | 0.380 | 0.355 | 0.333 | 0.332 | 0.329 | 0.278 | 0.239 |
| | Qwen/Qwen3-0.6B | 0.352 | 0.339 | 0.305 | 0.282 | 0.251 | 0.263 | 0.277 | 0.237 | 0.197 |
| | Qwen/Qwen3-1.7B | 0.364 | 0.327 | 0.279 | 0.251 | 0.218 | 0.215 | 0.238 | 0.195 | 0.165 |
| | Qwen/Qwen3-4B | 0.374 | 0.331 | 0.273 | 0.236 | 0.208 | 0.217 | 0.230 | 0.219 | 0.203 |
| | Qwen/Qwen3-8B | 0.377 | 0.343 | 0.298 | 0.256 | 0.230 | 0.222 | 0.229 | 0.192 | 0.141 |
| | Qwen/Qwen3-14B | 0.379 | 0.352 | 0.290 | 0.252 | 0.231 | 0.231 | 0.243 | 0.194 | 0.165 |
| Language Models (Perplexities) | Qwen/Qwen3-32B | 0.387 | 0.360 | 0.308 | 0.273 | 0.253 | 0.247 | 0.252 | 0.219 | 0.195 |
| | allenai/Llama-3.1-Tulu-3-8B-RM | 0.404 | 0.347 | 0.287 | 0.249 | 0.232 | 0.220 | 0.233 | 0.169 | 0.151 |
| | infly/INF-ORM-Llama3.1-70B | 0.246 | 0.200 | 0.197 | 0.157 | 0.177 | 0.167 | 0.149 | 0.108 | 0.128 |
| Reward Models (Scores) | nicolinho/QRM-Gemma-2-27B 0 | .054 | 0.027 | 0.030 | 0.002 | 0.013 | -0.022 | -0.010 | -0.055 | -0.029 |
| | nvidia/Llama-3.1-Nemotron-70B-Reward-HF | 0.164 | 0.120 | 0.105 | 0.060 | 0.070 | 0.067 | 0.060 | 0.035 | 0.060 |
| | Skywork/Skywork-Reward-Gemma-2-27B | 0.172 | 0.126 | 0.081 | 0.043 | 0.052 | 0.094 | 0.076 | 0.020 | 0.054 |
| | Skywork/Skywork-Reward-Gemma-2-27B-v0.2 | 0.158 | 0.112 | 0.102 | 0.078 | 0.073 | 0.059 | 0.034 | -0.001 | 0.021 |
| | gpt-4o-2024-11-20 (HHH) | 0.167 | 0.120 | 0.092 | 0.053 | 0.014 | 0.029 | 0.042 | 0.024 | 0.038 |
| LM Judges (Scores) | gpt-4o-2024-11-20 (Overall) | 0.239 | 0.186 | 0.152 | 0.099 | 0.054 | 0.035 | 0.060 | 0.032 | 0.080 |
| | prometheus (HHH) | 0.169 | 0.141 | 0.122 | 0.098 | 0.069 | 0.068 | 0.059 | 0.041 | 0.046 |
| | prometheus (Overall) | 0.125 | 0.086 | 0.082 | 0.060 | 0.041 | 0.073 | 0.061 | 0.078 | 0.065 |

