# OpenReview forum: "Artificial Hivemind: The Open-Ended Homogeneity of Language Models (and Beyond)"
_NeurIPS.cc/2025/Datasets_and_Benchmarks_Track — NeurIPS 2025 Datasets and Benchmarks Track oral_

### Official Review · Reviewer_cdsj · 2025-06-30

**Rating:** 5
**Confidence:** 4

**Summary:**

This paper introduces a new dataset for studying the diversity of LLM responses to open-ended queries. It consists of 26k real-world user queries, LLM responses to a subset of these queries, and corresponding human ratings. A key feature of the dataset is its emphasis on evaluating diversity by sampling multiple model responses to each prompt and collecting multiple human annotations for each response or response-pair.

The authors also empirically measured the similarity of responses along various dimensions, as well as the correlations between the similarity of responses and the uncertainty in human ratings. Overall, the paper provides plausible evidence for the "artificial hivemind" effect, i.e. the lack of diversity of LLMs' responses to open-ended queries.

**Additional Feedback:**

I believe this is an interesting paper that could be valuable to the community. My main concerns are surrounding the rigour and soundness of the conclusions drawn from the experiments. Specifically:
- Verifying the "open-endedness" of collected queries
- Contextualization of similarity scores through quantifying "open-endedness" or baselines
- Motivating and justifying the analysis in Section 4

**Dataset Code Accessibility:**

Partly

**Dataset Code Comments:**

Code is available but data is not.

**Ethical Comments:**

Sufficient amounts of details are provided regarding the data collection process. Data is collected properly and ethically and I have no ethical concern.

**Ethical Considerations:**

No, there are no or only very minor ethics concerns

**Final Justification:**

I believe the data included in this paper is a valuable resource to the community, because it is unusual to have multiple model responses, each rated by multiple human annotators, to real-world open-ended queries. The homogeneity phenomenon shown through experiments is also interesting, and can inspire future work on this topic.

Before rebuttal, my main concerns were:
1. The extent of the "open-endedness" of queries was not verified.
2. The cosine similarity scores used to show the existence of the homogeneity phenomenon lack reference and thus cannot be interpreted meaningfully.
3. The analysis on correlations with human disagreement is technically flawed and is not well motivated.

These concerns spoke directly to the core contributions of this paper and challenged its main values, which was why I recommended rejection.

Through rebuttal, the authors addressed these concerns to a reasonable degree that I am convinced of the soundness of the claims made in the paper, and therefore its values. Therefore, I now recommend acceptance.

**Limitations Weaknesses:**

### Dataset construction (Section 2) and artificial hivemind experiments (Section 3)

The primary weakness of this paper is twofold: a lack of verification for the "open-endedness" of queries during dataset construction, and the absence of a meaningful baseline for evaluating LLM response diversity in the experiments.

Although most queries will indeed allow multiple answers given the way they are collected, it does not necessarily mean 1) there will be many answers, 2) they will be diverse, and 3) they will be equally diverse across queries. For example, a query asking for a joke should elicit a much wider range of diverse responses than a query about the advantages of hardwood flooring. Because of this, I believe it is crucial to verify that queries can indeed elicit open-ended responses (based on some kind of definition of "open-ended").

This issue is compounded by a lack of experimental baselines to contextualize the similarity scores. As a result, although the cosine similarities seem high, their significance is unclear without a point of comparison. As an example of contextualization, the "open-endedness" of queries can be quantified in some way. This might not be easy and it does not need to be perfect, but a crude measure is still useful for fairly interpreting the similarity scores of LLMs. Another example might be providing the similarity scores of human responses to the same queries as a baseline. Without contextualization, the results as they are shown now in Figure 4, 5 and 6 do not convincingly show the existence of homogeneity.

Additionally, in Figure 6, most average similarity scores are tightly clustered in the range of 0.7-0.8 with little difference among many of them, and I think this makes it hard to meaningfully compare the numbers. It might be more insightful to compare the full distributions of the similarity scores, rather than just the averages, to reveal other distinguishing characteristics.

Finally, I do not understand the paragraph from line 167 to line 185 and the corresponding Figure 7. If the goal is to compare inter- and intra-model similarity, why not use the cosine similarity as was done in Figure 4 and 6? And how does the metric used here compare inter- and intra-model similarity exactly?

### Correlating automatic ratings with human ratings (Section 4)

For this set of experiments, I have doubts about the methodology for selecting "similar-quality" responses and those with "high annotator disagreement." I also have doubts about how correlations are calculated and interpreted.

The methodology for selecting "similar-quality" responses in Section 4.3 has two key issues. First, Tukey's fences would actually select median-quality responses, and I am not convinced that they are necessarily of similar quality (e.g. the maximal difference would be 2kIQR which is not small), or that this is capturing all similar-quality responses (e.g. two responses that are similar but both beyond the 75% percentile would not be selected). Second, for the pairwise setup, I am not sure I understand the selection methodology. The choice of the 60%-95% range seems arbitrary. And why the subset where a lower percentage of annotators marking responses similar would be the subset of higher similarity?

Furthermore, are correlations calculated with the average human ratings? If so, given the fact that these subsets are defined by high annotator uncertainty, I am not sure it is reasonable to measure the correlation with the "average" rating which ignores the annotator disagreement. Consequently, it is unclear what, if anything, can be concluded from a lower correlation score under this setup, as it may be an artifact of the aggregation method.

Similarly, in Section 4.4, the ways responses are selected lack justification. For the absolute setup, entropy does not measure annotator disagreement in my opinion. For instance, a bimodal, polarized distribution (e.g. ratings are concentrated on "strong prefer 1" and "strong prefer 2") indicates stronger disagreement but can have lower entropy than a more uniform distribution (e.g. ratings are evenly distributed across 5 categories). Is there a reason why entropy is used instead of e.g. Krippendorff's alpha? And for pairwise setup, similarly, the metric is proposed without justification.

Finally and more fundamentally, what is the conceptual distinction between "similar-quality alternative responses" (Section 4.3) and "responses with high annotator disagreement" (Section 4.4)? Are they equivalent? What kinds of responses would false under one category but not the other? Probably there are differences between the two, but they are not discussed in the paper, which makes it hard to understand intuitively what is being done and why.

In summary, the motivation for the experiments in Section 4 is unclear, and the methodologies used for selecting and analyzing data subsets have several points that require significant clarification and justification.

**Strengths Contributions:**

The paper addresses the under-explored phenomenon of LLM response diversity, providing a valuable dataset that can lay the foundation for future research on this topic. Key strengths of the dataset include its use of valuable open-ended queries from real-world users and its collection of multiple LLM responses and human ratings per query, which enables a more reliable study of diversity. The selection of models covers a wide range of state-of-the-art LLMs, and the dataset is of a respectable scale given the sophistication of its curation.

The paper's empirical experiments also provide plausible evidence for the homogeneity of LLM responses, shedding light on its varying extents across different models and contexts.

---

> ### Author Rebuttal · Authors · 2025-07-31
>
> We thank reviewer `cdsj` for their insightful and constructive feedback, recognizing that “the paper addresses the under-explored phenomenon of LLM response diversity, providing a valuable dataset that can lay the foundation for future research on this topic." **Below we provide responses within the space limit, and we are happy to provide further clarifications during the discussion period!**
>
> ## **Verify the open-endedness of queries**
>
> Thanks for the suggestion! We agree that verifying the open-endedness of collected queries will strengthens our dataset’s validity. We conducted two new human studies.
>
> We evaluated 100 examples from our dataset with 86 Prolific annotators, assigning 3 per question. Eligible participants were fluent English speakers with >99% approval rates, 500–10,000 prior submissions, and education beyond high school. Each answered two questions per query.
>
> > Q1: *Is the user query open-ended?* In other words, can it reasonably allow for several different, similarly valid answers, rather than just a single, fairly deterministic one?
>
> - A: Yes
>
> - B: No
>
> We used majority vote among 3 annotators to assess query open-endedness: **89% of queries were judged open-ended**, providing strong evidence of high open-endedness of collected queries. With a more inclusive criterion, counting a query as open-ended if at least one annotator agreed, **100% of queries qualified as open-ended.**
>
> > Q2: If the query is open-ended, how many different, reasonable alternative answers do you think could apply?
>
> - A: Fewer than 3
>
> - B: 3 to 10
>
> - C: 10 to 20
>
> - D: More than 20
>
> Beyond the binary classification of open-endedness, we assessed the breadth of reasonable answers. Annotators judged **81.27% of queries to allow 3+ alternatives**, and **34.66% to permit over 20**, showing that real-world queries are rarely narrow or deterministic. This underscores the need for diversity-aware LLM evaluation, as simplistic benchmarks with limited answer ranges fail to capture the true complexity of user queries.
>
> | Answer | % |
> |---|---:|
> | Fewer than 3 | 18.73 |
> | 3 to 10 | 31.87 |
> | 10 to 20 | 14.74 |
> | More than 20 | 34.66 |
>
> We will include these new human annotation results in the camera-ready to provide a more grounded interpretation and to strengthen our definition of open-endedness early in the paper.
>
> ## **Clarify the Interpretation of Similarity Scores**
>
> Thank you for raising this important point! We agree that interpreting similarity scores is essential. Our primary measure, i.e., cosine similarity between text embeddings, is widely used and generalizable, but not always intuitive. To address this, we present analyses in lines 167–185 and Figure 7 to make the scores’ meaning more concrete.
>
> Specifically, we sampled 25 models, each generating 50 responses to 100 open‑ended queries, creating 1,250 responses per query. Clustering responses by embedding similarity, we examined the top‑N closest clusters. If similarity were purely within‑model, clusters would be dominated by a single model; instead, at N = 50, clusters averaged ~8 distinct models (often >10). This shows that, despite architectural and training differences, models frequently converge, producing highly overlapping outputs that clustering by sentence embeddings cannot distinguish.
>
> While cosine similarity does not capture every aspect of diversity, we use it as a consistent and scalable diagnostic. To aid interpretation, we will add an appendix table with example response pairs at different similarity levels. Regarding Figure 6, Tables 6–10 in the appendix provide finer‑grained repetition scores, revealing nuanced patterns such as stronger similarity between OpenAI and Qwen models (L141–L150).
>
> We will ensure these clarifications are made explicit in the final camera-ready.
>
> ## **Clarify Human vs. Model Ratings**
>
> Thanks for your very thoughtful & constructive feedback! **Given space constraints in both the submission and this rebuttal, we provide key clarifications and additional analyses here, with further details to be included in the appendix of the camera-ready. We are also very happy to elaborate on specific points during the discussion period if requested.**
>
> ### **“Similar-Quality” Responses**
>
> There is no gold-standard for selecting responses of similar quality. We used Tukey’s fences as a representative method, which stratifies responses by excluding outliers while retaining typical cases. To check robustness, we **additionally tested several alternatives (sliding window, median expansion, gap-based, centroid- and distance-based, both local and global; details available upon request)**. Across all methods and subset sizes (40, 60, 80% of data), the trend is consistent: similar-quality subsets yield lower human–model correlation.
>
> | **Method**                | LM Judges  |    (full=   |    .305)     | RM        |     (full=    |     .330)   | LLM PPL   |    (full=   |     .361)    |
> |---|---:|---:|---:|---:|---:|---:|---:|---:|---:|
> | %                    | **80**  | **60**| **40**| **80**  | **60**| **40**| **80**  | **60**| **40**|
> | Optimized Sliding (L)   | .268     | .248  | .226  | .316     | .312  | .300  | .365    | .412  | .341  |
> | Centroid-Based (L)      | .259     | .230  | .206  | .300     | .297  | .278  | .347    | .372  | .357  |
> | Distance-Based (L)      | .259     | .230  | .206  | .300     | .297  | .278  | .346    | .372  | .357  |
> | Tukey (L)               | .268     | .248  | .226  | .316     | .312  | .300  | .365    | .412  | .341  |
> | Median Expansion (L)    | .249     | .222  | .244  | .290     | .280  | .332  | .351    | .399  | .428  |
> | Gap-Based (L)           | .265     | .260  | .226  | .318     | .314  | .326  | .373    | .414  | .387  |
> | Tukey (G)               | .157     | .138  | .121  | .183     | .178  | .096  | .247    | .242  | .149  |
> | Median Expansion (G)    | .192     | .174  | .118  | .254     | .237  | .164  | .302    | .262  | .265  |
> | Optimized Sliding (G)   | .157     | .138  | .121  | .183     | .178  | .096  | .247    | .242  | .149  |
>
> *L = Local; G = Global. Numbers are Spearman correlation.*
>
> For the **pairwise setup**, we rely on direct human annotations: annotators can mark pairs as “similar quality.” Ranking pairs by this frequency, we again find that narrower subsets yield even lower correlation, reinforcing the trend (Fig. 10).
>
> ### **Responses with “High Disagreement”**
>
> We used entropy as our primary disagreement measure, given 15 annotations per response. A bimodal 50/50 split indicates less disagreement (half the annotators agree within each class), whereas a uniform distribution shows higher disagreement (~20% agreement per label), consistent with entropy. To ensure robustness, we also **additionally tested other methods: Gini impurity, pairwise disagreement, majority‑vs‑minority, Fleiss’ κ, and grouped entropy.** All show the same trend: **higher‑disagreement subsets yield lower human–model correlation.**
>
> | **Method**               | LM Judges  |   (full= |   .305) | RM        |   (full= |   .330) | LLM PPL   |   (full= |   .361) |
> |---|---:|---:|---:|---:|---:|---:|---:|---:|---:|
> | Top N most disagreed  =                       | **120**   | **90**  | **60**  | **30**    | **120** | **90**  | **60**    | **30**  |         |
> | entropy                   | .287      | .254    | .276    | .070      | .292    | .228    | .175      | -.073   |         |
> | entropy_grouped           | .247      | .227    | .121    | .139      | .293    | .246    | .081      | .153    |         |
> | gini_impurity             | .299      | .278    | .216    | .162      | .331    | .284    | .129      | -.043   |         |
> | pairwise_disagreement     | .295      | .278    | .216    | .103      | .337    | .284    | .129      | -.012   |         |
> | majority_vs_minority      | .338      | .277    | .281    | .188      | .363    | .377    | .322      | .378    |         |
> | fleiss_kappa_single       | .278      | .316    | .326    | .316      | .231    | .272    | .168      | .208    |         |
>
> For the **pairwise setup**, we ranked examples by majority-vs-minority scores, observing the same pattern. Detailed discussions of alternative criteria will be added in the appendix.
>
> ### **Why Average Human Scores**
>
> Our motivation stems from how reward models and LM judges are used in training open-ended responses without a single “ground truth.” While individual annotators may prefer different answers, similar average scores often indicate comparable quality. Yet current reward models overlook this equivalence, assigning divergent scores that cause downstream models to favor one option despite balanced human preferences.
>
> We therefore use average human scores, which capture when multiple responses are high-quality across different subpopulations and avoid the misleading signals of isolated ratings. With 15 annotations per data point, our resource robustly reflects collective quality, providing a sound basis for recognizing when responses deserve comparable credit. This approach better aligns training with the nuanced quality landscape rather than artificially narrowing preferences.
>
> ### **Distinguish “Similar-Quality” from “High-Disagreement”**
>
> - **Similar-quality:** Responses with comparable average human scores.
> - **High-disagreement:** Responses where different annotators have different labels, regardless of average.
>
> They capture overlapped but distinct phenomena: responses may be similar-quality without disagreement, or vice versa. Studying both reveals complementary ways models fail to capture the breadth of valid human judgments.
>
> ## **Data availability**
>
> Thanks for pointing this out! The correct dataset link is: https://huggingface.co/collections/liweijiang/artificial-hivemind. The link in our paper was accidentally broken, so access was only possible via OpenReview. We apologize for the confusion and will fix this in camera-ready.

---

> > ### Comment · Reviewer_cdsj · 2025-08-02
> >
> > Thanks for the response, that's helpful. The first of my 3 main concerns is addressed, and I have some follow-ups on a few points regarding the 2 other concerns.
> >
> > # Verifying open-endedness
> >
> > This new survey is indeed very helpful for gauging the open-endedness of queries in the dataset. The statistics verify that the most queries do invite open-ended responses.
> >
> > Meanwhile, the extent of open-endedness varies a lot across queries, and a non-trivial amount allow only few answers. I think this is important for interpreting the performance of models. I'd encourage the authors to analyze the results in this context. For instance, does the similarity score correlate with the statistics? Within which group is the hivemind phenomenon most prominent? Can the responses be clustered into the corresponding numbers of clusters?
> >
> > Relatedly, is it possible to provide a baseline for the similarity scores? Human responses are ideal but probably too difficult to collect, what about response-pairs from randomly-drawn unrelated prompts (random baseline)?
> >
> > Overall, my goal here is to be able to interpret the similarity scores within a frame of reference, otherwise it's hard to conclude if those scores are high or not.
> >
> > # Similar-quality responses
> >
> > I'd like to know more about the alternative methods. My main concern is removing outliers is neither necessary nor sufficient for selecting similar-quality responses. Responses can be typical while different (differences up to 2kIQR), and two similar responses can be both outliers (though this is probably less likely). With that said, I do think the trend is interesting (though it doesn't appear in perplexity), but perhaps needs an alternative characterization than "similar-quality".
> >
> > # High-disagreement responses
> >
> > In my opinion, the added metrics are more suitable for measuring disagreement than entropy, and I encourage the authors to use one of them as the main metric e.g. Fleiss' kappa. The trend is much weaker for majority vs minority and Fleiss' kappa metrics though, and I encourage the authors to amend the analysis accordingly.
> >
> > # Justifying averaging human ratings
> >
> > I agree that for responses with high annotator disagreement, looking only at individual ratings is inappropriate. But, at the same time, looking only at the means is also inappropriate as doing so ignores the disagreement. I think what is needed is some kind of measure that takes into account the rating disagreement.
> >
> > On the other hand, model training is a good argument for averaging ratings, in the sense that if one were to use these ratings to train a reward model, they would probably be averaged.
> >
> > I admit I don't have a good solution, but I encourage the authors to explore if annotator disagreement can be taken into account in some way. At least, I suggest acknowledging this shortcoming and motivating from the model training argument.

---

> > > ### Author Response · Authors · 2025-08-05
> > >
> > > Dear reviewer `cdsj`,
> > >
> > > Thank you for your response and for the thoughtful follow‑up questions! We are actively working on addressing your remaining concerns and will be posting our responses tomorrow. Thank you again for your time and valuable feedback.

---

> > > > ### Author Response · Authors · 2025-08-08
> > > >
> > > > Thank you again for your thoughtful comments and for your dedication to helping us improve the paper. We apologize for the delay in our follow-up, as we faced some challenges in getting compute resources for the additional experiments due to some accidental deletion of our previous data. We have since conducted the requested experiments and would like to share the resulting insights, along with further discussion and clarifications. If you have further questions regarding our paper, we’ll be more than happy to quickly answer them. (Due to space constraints, we’ll share our responses across separate threads.)
> > > >
> > > > ## **Random similarity baseline**
> > > >
> > > > Thanks for helping us to brainstorm ways for interpreting the similarity scores within a frame of reference! You are totally right that the human baseline here might be hard to collect within the scope of this project, so we have computed the random baseline that you suggested. Below shows the results.
> > > >
> > > > We added the random baseline for Figure 4 (here we show a couple of  representative models; see full results in Figure 4):
> > > >
> > > > | Similarity Score Ranges | **random** | avg. | gpt-4o | claude3.5 | gemini1.5-pro | Llama3.1-1405B-Instruct |
> > > > |-------------------------|--------:|-----:|--------:|-----------:|---------------:|-------------------------:|
> > > > | 0.9-1.0                 | 0       | 43.8 | 51.0    | 59.0       | 55.0           | 33.0                     |
> > > > | 0.8-0.9                 | 0       | 35.2 | 36.0    | 27.0       | 37.0           | 43.0                     |
> > > > | 0.7-0.8                 | 0       | 12.6 | 10.0    | 12.0       | 11.0           | 14.0                     |
> > > > | 0.6-0.7                 | 0       | 4.9  | 1.0     | 3.0        | 3.0            | 7.0                      |
> > > > | 0.5-0.6                 | 0       | 1.9  | 2.0     | 1.0        | 1.0            | 0.0                      |
> > > > | 0.4-0.5                 | 0       | 0.7  | 0.0     | 0.0        | 0.0            | 0.0                      |
> > > > | 0.3-0.4                 | 0       | 0.6  | 0.0     | 1.0        | 2.0            | 1.0                      |
> > > > | 0.2-0.3                 | 0       | 0.2  | 0.0     | 0.0        | 0.0            | 0.0                      |
> > > > | 0.1-0.2                 | 100     | 0.0  | 0.0     | 0.0        | 0.0            | 0.0                      |
> > > > | 0.0-0.1                 | 0       | 0.0  | 0.0     | 0.0        | 0.0            | 0.0                      |
> > > >
> > > > We observe that randomly selected response pairs consistently fall into very low similarity ranges (e.g., 0.1–0.2), indicating high dissimilarity.
> > > >
> > > > In contrast, for Figure 6, the pairwise similarities mostly fall between 0.7 and 0.9, while the random baseline remains around **0.14**.
> > > >
> > > > We will include these new results in the camera-ready version to better contextualize the meaning of the similarity scores. Thank you again for your helpful suggestion!
> > > >
> > > > ## **Clarification around high-disagreement responses**
> > > >
> > > > Thank you for suggesting the use of Fleiss’ kappa as the primary metric for the analysis of high-disagreement responses. **We will update the paper to reflect this change. The other results will be retained in the appendix for complete reference for the readers, and we will revise the analysis accordingly.**
> > > >
> > > >
> > > > ## **Justification of averaging human ratings**
> > > >
> > > > Thank you again for your thoughtful suggestion! **We will include the motivation around model training in the final paper to better ground the design of our experimental setup and clarify our goals for readers.**
> > > >
> > > > In parallel, we have already released all human annotation data, including individual annotations for both absolute and pairwise judgments. To the best of our knowledge, this is the first human-annotated preference dataset with 25 annotations per example for open-ended queries and their alternative responses. We hope this resource will be valuable to the community and help support future work on disagreement analysis.

---

> > > > > ### Author Response · Authors · 2025-08-08
> > > > >
> > > > > ## **Clarification around similar-quality responses**
> > > > >
> > > > > Thanks again for your thought-provoking questions! In response to your suggestion for an alternative characterization of "similar-quality," we will clearly define this term in the paper as follows:
> > > > >
> > > > > > “Similar-quality” refers to responses that receive comparable average human scores across a broad pool of annotators, even if individual preferences vary. This framing is motivated by open-ended queries that naturally admit multiple high-quality responses, where individuals may differ in their preferences, but collectively tend to rate the responses similarly on average.
> > > > >
> > > > > **We will incorporate this clarification in the final camera-ready version and revise the surrounding text to more clearly communicate our motivation.** Below, we also provide additional details on alternative methods for quantifying “similar-quality.” While there is no single gold-standard method, we believe each approach offers valuable perspective. **To provide full context, we will include all these methods in the appendix of the final camera-ready version.**
> > > > >
> > > > > ### Details of additional methods:
> > > > >
> > > > > > distance_based
> > > > >
> > > > > This brute-force method checks all possible subsets of size N and computes the sum of pairwise distances within each subset. It selects the subset with the smallest total distance, guaranteeing the most tightly packed group. It’s exact but computationally expensive since it explores every combination.
> > > > >
> > > > > > centroid_based
> > > > >
> > > > > Also a brute-force method, this one evaluates all possible subsets of size N and measures how tightly the numbers cluster around their mean (centroid) using variance. The subset with the smallest variance is chosen, ensuring the numbers are closely grouped around a central value. It’s conceptually similar to k-means clustering in 1D.
> > > > >
> > > > > > optimized_sliding
> > > > >
> > > > > This method sorts the list and uses a sliding window of size N to find the segment with the smallest range. It’s efficient and guarantees a contiguous cluster. An early exit occurs if a window with zero range (identical values) is found, since that’s the best possible outcome.
> > > > >
> > > > > > median_expansion
> > > > >
> > > > > This method starts from the median of the sorted list and expands outward to include the closest values until reaching the desired subset size. The idea is that the median anchors the subset in the center of the data, ensuring the selected numbers are as balanced and tightly clustered as possible.
> > > > >
> > > > > > gap_based
> > > > >
> > > > > This method sorts the values and identifies the smallest N-1 gaps between consecutive numbers. It then builds a subset spanning from the first to last chosen gap, ensuring the selected values are tightly packed together. If the range is larger than N, it applies a local sliding window to refine. The approach balances efficiency with a direct focus on minimizing the spacing between included numbers.
> > > > >
> > > > > *Local* refers to measuring similarity among responses to the same user query, whereas *global* refers to measuring similarity across the entire pool of responses.

---

> > ### Author Response · Authors · 2025-08-08
> >
> > ## **Analyzing the results based on different levels of open-endedness**
> >
> > Thank you for raising this insightful question! We explored this idea and summarize our findings below.
> >
> > We first examined the “level of open-endedness” human annotations that we collected. Interestingly, we found notable variation across annotators in how they judged the level of open-endedness. For example, below is a sample of annotations from three annotators for each prompt, illustrating that perceptions can differ substantially across individuals in terms of the “level of open-endedness”:
> >
> > > **Sample Annotations**
> > > `['More than 20', '10 to 20', '3 to 10']`
> > > `['More than 20', '3 to 10', 'More than 20']`
> > > `['More than 20', 'More than 20', 'More than 20']`
> > > `['Fewer than 3', 'More than 20', 'More than 20']`
> > > `['Fewer than 3', '10 to 20', 'More than 20']`
> >
> > This variability suggests that it's difficult to rely on such annotations to assign a definitive level of open-endedness to any single example.
> >
> > Nonetheless, we conducted a coarse-grained analysis using a simplified categorization:
> >
> > - If **any** annotator labeled a prompt as having *"More than 20 responses"*, we categorized it as **high open-endedness**.
> >
> > - If **no** annotator gave that label, we categorized it as **low open-endedness**.
> >
> > Across responses from 42 models, we found the following average similarity scores:
> >
> > - **High open-endedness**: 0.800
> >
> > - **Low open-endedness**: 0.837
> >
> > This result suggests that for prompts perceived as more open-ended with our coarse-grained estimation, models do not necessarily produce more diverse responses.
> >
> > We will include this analysis in the final camera-ready version. Thank you again for the thought-provoking question!

---

> > > ### Comment · Reviewer_cdsj · 2025-08-08
> > >
> > > Thank you for your thorough responses! I appreciate the efforts put into them. The new results and analysis have addressed the remaining concerns, and I am now satisfied this paper should be accepted.

---

> > > > ### Author Response · Authors · 2025-08-08
> > > >
> > > > We’re very glad our responses have addressed your concerns, and we truly appreciate your thoughtful, constructive, and genuine feedback, which has been invaluable in improving our paper! We will make sure to incorporate all the discussed changes and additional experiments into the final version. Thank you once again for your time and consideration!

---

### Official Review · Reviewer_WB9g · 2025-07-01

**Rating:** 4
**Confidence:** 1

**Summary:**

This paper presents CHATS, a large dataset for studying how language models handle open-ended questions. The authors show that models often produce repetitive and similar answers, which they call the "Artificial Hivemind" effect. With extensive human annotations, CHATS reveals that current models struggle to match diverse human preferences, making this dataset a useful resource for future research on improving creativity and diversity in AI outputs.

**Dataset Code Accessibility:**

Yes

**Dataset Code Comments:**

The authors provide code and dataset link.

**Ethical Considerations:**

No, there are no or only very minor ethics concerns

**Final Justification:**

Thanks for the response, I will keep my score.

**Limitations Weaknesses:**

1. The figure1 can be more concise.
2. Maybe the authors can add some discussion about taxonomy problem in related works as it is discussed in the main body.

**Strengths Contributions:**

1.The paper offers a systematic taxonomy for open-ended prompts, which helps structure analysis and enables more targeted research on different types of queries.
2.The study thoroughly investigates the "Artificial Hivemind" effect, revealing both intra-model repetition and inter-model similarity, which are important findings for the field.
3.With over 31,000 human annotations, the dataset allows for fine-grained evaluation and analysis of how models align (or fail to align) with human preferences.

---

> ### Author Rebuttal · Authors · 2025-07-31
>
> We thank reviewer `WB9g` for their positive and constructive feedback. We especially appreciate their recognition that “the study thoroughly investigates the 'Artificial Hivemind' effect” and that “they are important findings for the field.” We are also grateful for their callout of our dataset’s value for “fine-grained evaluation and analysis of how models align (or fail to align) with human preferences.”
>
> Below, we provide a point-by-point response to their comments and questions, and we are happy to follow up during the discussion period for any additional questions!
>
> ## **Making Figure 1 More Concise**
>
> We thank the reviewer for suggesting improvements to the design of Figure 1. To enhance clarity, we will reduce the number of representative models shown in the main figure to declutter the visual presentation. The complete version of the figure will be included in the appendix for readers who wish to see the full details.
>
> ## **Clarification on adding discussion about taxonomy problem in related works**
>
> We appreciate the reviewer’s suggestion on this point. We will incorporate a discussion of prior work on open-endedness taxonomies in the Related Work section to better ground our study.

---

### Official Review · Reviewer_zt6D · 2025-07-02

**Rating:** 4
**Confidence:** 4

**Summary:**

This paper introduces $\infty$-CHATS, a user query dataset that covers 6 top-level and 17 fine-grained categories for ambiguity resolution evaluation of LLMs (i.e., respond properly when no golden answer exists). Authors observed an “Artificial Hivemind” effect, where LLMs produce highly similar responses both within and across models, despite different decoding methods. Extensive human annotations further reveal current reward models and LLM judges poorly capture nuanced human preferences in ambiguous settings, highlighting the need for better diversity and alignment in open-ended generation.

**Additional Feedback:**

I like the general idea of the paper, I encourage the authors to conduct a deeper study on this direction as it might help the community to better understand how LLMs' behavior is shaped during pre/post-training.

**Dataset Code Accessibility:**

Yes

**Dataset Code Comments:**

The code is clear and well-structured. It would be better if the scripts to test each model candidate can also be included, so that the readers can better understand if the system prompt also has some impact on how the conclusion of the paper is drawn.

**Ethical Comments:**

This paper reports the collective similarity of LLMs with significant differences in model family, training corpus and other details. It serves as more of a testbed to reflect an existing problem rather than a corpus that facilitate future training. It is unlikely that it should amplify societal biases or raise further ethical concerns.

**Ethical Considerations:**

No, there are no or only very minor ethics concerns

**Final Justification:**

The authors have taken additional effort to address my concerns. While I still believe there could be deeper discussion to interpret the results, I agree these might be out of the scope of a **dataset** paper. I'm increasing my score drastically from 2 to 4 to reflect these updates.

**Limitations Weaknesses:**

- [Addressed] I respectfully challenge the authors of their core claim. In my opinion, the so-called Artificial Hivemind could be much less worrisome than what this paper claims. Considering each chunk of the publicly available data collected to train LLMs as a sequence of prefix-next_token projections, our training target of current autoregressive LLMs is simply to capture the pattern of all such projections. Now, as a tractable, explicit density/mass model, autoregressive language models are **designed** to approximate the **exactly average** behavior to predict the next-token given the prefix with usually one implicit assumption that **the geometry of "tokens" is radial** (this is imposed by the usage of the linear transformation from the last layer hidden vector to the vocabulary space). By the *law of large numbers*, it is naturally the case that if most LLMs' corpus are 1) collected roughly unbiased and 2) having more tokens exceeding a sufficiently large number, then the average (distributional) behavior given a particular prefix shall naturally be expected converge to a similar pattern, regardless of the LLMs' minor differences like architecture. tokenizer etc. As long as we still stick with autoregressive models, I won't be super surprised if this "Hivemind" phenomenon  continue to be the case.

- [Addressed] It is arguable if we indeed don't want this cross-model similarity at all. While this "Hivemind" may be explained using the law of large numbers and my previous hypothesis, it needs further validation by designing a testbed that shares the idea of this work but applies to the foundation models w/o instruction following capabilities. Future work can also measure that given the same foundation model, if and how much post-training will shape the behavior of different models given open-ended queries.

**Strengths Contributions:**

- The paper provides a testbed hybriding automatically crafted queries and hand-writtent ones (for blindspot sweeping) that reveals the unnecessary collective similarity for open-ended text generation queries among different families of language models using various decoding strategies. Such an "artificial hivemind" effect is worrisome, call on the community to think deeper towards improving the basic diversity in models' default behaviors.

---

> ### Author Rebuttal · Authors · 2025-07-31
>
> We thank reviewer `zt6D` for their valuable and thoughtful feedback. We especially appreciate their recognition that our paper "provides a testbed hybriding automatically crafted queries and hand-written ones (for blindspot sweeping)" and that it "reveals the unnecessary collective similarity for open-ended text generation queries among different families of language models using various decoding strategies."
>
> Below, we provide point-by-point responses to their comments and questions, and we would be more than happy to address any further clarifications during the discussion period!
>
> ## **Response to the Core Challenge: Impact of the "Artificial Hivemind" Observation**
>
> We sincerely appreciate the reviewer's insightful theoretical perspective on the Artificial Hivemind phenomenon. The connection to the law of large numbers is indeed thought-provoking and adds valuable depth to our understanding of why convergent behaviors might emerge in autoregressive models. We believe this theoretical framework actually strengthens rather than diminishes the importance of our empirical findings, for several key reasons:
>
> > **Societal Implications of Homogenization**
>
> Even if convergence is statistically expected, its societal consequences cannot be overlooked. As billions of users increasingly rely on LLMs for creative, educational, and decision-making tasks, understanding and measuring behavioral convergence becomes crucial. Recent studies have already begun documenting changes in human writing patterns, divergent thinking, creative ideation following ChatGPT's release, suggesting that model homogenization may have cascading effects on human expression [1-6]. Our work provides the tools to quantify and track these important societal shifts.
>
> - [1] https://arxiv.org/abs/2410.03703
> - [2] https://arxiv.org/abs/2402.01536
> - [3] https://arxiv.org/abs/2401.13481
> - [4] https://www.science.org/doi/10.1126/sciadv.adn5290
> - [5] https://arxiv.org/abs/2506.08872
> - [6] https://arxiv.org/pdf/2502.11266
>
> > **Implications for Data Distillation and Model Training**
>
> The reviewer’s point about convergence is particularly relevant to current practices in the field. Synthetic data generation has already played a pivotal role in advancing model capabilities and is now a common strategy for obtaining large-scale training data. However, it is well established that distilling from a single model can exacerbate mode collapse, reinforcing narrow response patterns, reducing output diversity, and even cause the model to degenerate [7,8]. To address this, many recent works propose model swarms or multi-agent frameworks, where multiple language models are used collectively in the hope of generating more diverse data [9,10].
>
> Our findings reveal a critical limitation of this approach: even when multiple state-of-the-art models are involved, diversity in open-ended cases is not guaranteed. In fact, our evidence shows that different models often converge toward highly similar responses, undermining the intended benefits of multi-model distillation. This has immediate implications for research efforts relying on model ensembles for data quality improvement, synthetic data generation, and constitutional AI. Without careful diagnostic tools like ours, practitioners may overestimate the effectiveness of cross-model diversity and inadvertently perpetuate homogenization.
>
> - [7] https://openreview.net/forum?id=ShjMHfmPs0
> - [8] https://arxiv.org/abs/2410.16713
> - [9] https://arxiv.org/abs/2410.11163
> - [10] https://arxiv.org/abs/2506.04721
>
> > **The Gap Between Statistical Expectation and Desired Behavior**
>
> While the reviewer is correct that convergent behavior may be statistically expected under the current model development paradigm, this does not make it desirable or inevitable. Our work underscores the need for training methodologies, architectures, datasets, and post-training techniques that preserve diversity even at scale. The fact that a phenomenon is explainable does not diminish the urgency of addressing it; rather, it highlights the importance of proactive intervention.
>
> Current models systematically underdeliver in ambiguous contexts, exposing a preference–behavior gap that represents a fundamental alignment challenge. Importantly, we do not argue against all forms of convergence. In factual or verifiable domains, alignment toward correct answers is both expected and desirable. Our concern lies specifically with ambiguous, open-ended queries, where multiple reasonable answers exist but models nonetheless collapse toward uniformity. Recognizing this distinction is crucial to appreciating the scope and impact of our contribution.
>
> > **Final Notes**
>
> We greatly appreciate the reviewer’s theoretical perspective on model convergence, which we agree enriches the broader understanding of LLM behavior. At the same time, we kindly invite the reviewer to consider our work’s empirical and technical contributions. Our paper offers:
>
> - **A novel evaluation framework** built on a large-scale dataset of ambiguous queries, organized under a comprehensive open-ended taxonomy with 17 fine-grained categories
> - **Rigorous empirical analyses** uncovering consistent behavioral patterns across 70+ state-of-the-art models
> - **Robust quantitative metrics** for assessing response diversity and alignment with human preferences in open-ended settings
> - **Systematic evidence** that current language models, reward models, and LM judges fail to capture human preferences for diversity
> - **Comprehensive human annotations**, with 25 independent ratings per example across absolute and pairwise judgments, enabling fine-grained and reliable evaluation of distributional preferences in open-ended contexts
> - **A large-scale training data resource** that can be used to enhance models’ ability to handle open-ended queries during alignment and post-training
>
> While we welcome debate on whether the Hivemind phenomenon should be concerning from a theoretical standpoint, we emphasize that our dataset, evaluation methodology, and empirical findings stand independently as valuable resources. These tools allow the community to measure and potentially address diversity issues, regardless of one’s stance on convergence.
>
> We view our work as complementary to theoretical perspectives like the reviewer’s: by providing concrete measurement tools and robust evidence, we enable the community to make more informed decisions about whether and how to address these phenomena. Our primary contribution is thus empirical and methodological, equipping the field with new means to understand and evaluate LLM behavior in ambiguous contexts.
>
>
> ## **Clarification of the Scope and Future Directions**
>
> Thanks for encouraging deeper exploration of how LLM diversity behaviors are shaped during different stages of training. While tracing the precise roots of the Artificial Hivemind effect is beyond the scope of this work, we agree that this is an important direction for the field. To support such inquiry, we will expand our discussion to highlight several promising avenues for future research:
>
> * **Foundation model analysis:** Extending our testbed to foundation models without instruction-following capabilities, as suggested, to help disentangle the respective roles of pre-training and post-training in driving behavioral convergence.
> * **Training stage attribution:** Quantifying the relative contribution of different post-training pipelines (e.g., supervised fine-tuning, RLHF/RLAIF, constitutional training) to the emergence of the Hivemind effect.
> * **Mitigation strategies:** Exploring diversity-aware training objectives, alignment strategies, and decoding methods designed to maintain response variety without sacrificing quality.
>
> Finally, to ground how our resources can inform future research in LM diversity. In the camera-ready version, we will add actionable uses that extend beyond diagnosis, including:
>
> * **Integration into red-teaming workflows** to stress-test model coverage and robustness in open-ended settings.
> * **Using our dataset as a training prompt resource** for developing reinforcement learning approaches that explicitly reward diverse mode exploration during alignment.
> * **Benchmarking decoding strategies** under the Artificial Hivemind metric to evaluate their ability to counteract homogenization.
> * **Providing diagnostic signals for curriculum design**, enabling systematic exposure to open-ended prompts that are most susceptible to mode collapse.
>
> Together, these directions and applications frame the Artificial Hivemind not merely as an observational finding but as a resource to guide more diverse and human-aligned model development.
>
> ## **Improvement of the codebase**
>
> Thanks for your suggestion in improving our code base to include the scripts for testing each model candidate. We will update our codebase to reflect this change to assist readers easy follow-up and close examination with our infrastructure.

---

> > ### Comment · Reviewer_zt6D · 2025-08-01
> > **Re Rebuttal**
> >
> > Thanks for the authors' thoughtful response. I value this paper very much and really want to find solid reasons to calibrate my rating. I'd like to follow up with the following questions:
> >
> > 1) I think to highlight the finding in this paper, one crucial way to disprove my attack against you is to test whether **paraphrasing** the queries shall have significant impact on the continuations generated by different models. You may want to check each candidate model's self-consistency (i.e. tend to use similar rhetorical ways to continue the semantically-controlled prefices that are superficially/literally different). Then, show if the same paraphrase of the same semantic meaning generates similar responses among different models, or it is rather the case that all paraphrases of the same semantic meaning will regardlessly trigger all models to generate collectively mode-collapsed responses. Since this is just the beginning of the second round discussion, I assume this experiment result shall be easy to collect. You are more than welcomed to use a much simplified setting with less models and prompts, as long as you can show a reasonable explanation. If you should encounter any difficulities, please let me know as it is definitely understandable. If there could be any significant results for this point, I'm more than willing to significantly raise my scores.
> >
> > 2) Do you think the entropy of the open-ended query question should have an impact on the strength of the "Hivemind Effect"? While all these less definitive questions are marked as open-ended queries, there should still be some ones that match more diverse questions than others. How do the models tend to collectively mimic each other with the entropy of the response space growing?

---

> > > ### Author Response · Authors · 2025-08-01
> > >
> > > Thanks for your quick response and thoughtful follow-up questions! We're actively working on investigating these points and will follow up soon.

---

> > > ### Author Response · Authors · 2025-08-07
> > > **Follow-up on Q1 from the reviewer’s comments (we’ll address Q2 in a separate thread due to space limit)**
> > >
> > > Thank you again for your thoughtful comments. We apologize for the delay in following up, as we faced some challenges in getting compute resources for the additional experiments. We sincerely appreciate the reviewer’s patience and commitment to helping us improve the paper. We have now completed the requested experiments and would like to share the resulting insights, along with additional discussion and clarifications.
> > >
> > > ## **Additional experiments with paraphrases**
> > >
> > > We conducted an experiment to examine how prompt paraphrasing affects response similarity across language models. Starting with 30 prompts from our evaluation set, we used `gpt-4.1-2025-04-14`  to generate 4 paraphrases for each original prompt, ensuring the paraphrases maintained similar semantic meaning without drastic changes in connotation with further LLM judge verification.
> > >
> > > For each original prompt and its paraphrases (150 prompts total), we generated 20 responses using 42 representative models from different model families and sizes. We used consistent sampling parameters from our previous experiments: top-p=0.9 and temperature=1. We then obtained sentence embeddings for all responses using OpenAI's `text-embedding-3-small` model.
> > >
> > > To measure response consistency, we computed two types of semantic similarity scores:
> > >
> > > - **Within-prompt similarity**: Average pairwise similarity among the 20 responses generated from the same original prompt
> > >
> > > - **Cross-paraphrase similarity**: Average similarity between responses from the original prompt and responses from its paraphrases
> > >
> > > Our results across all 42 models show that the **within-prompt similarity averaged 0.821**, while the **cross-paraphrase similarity averaged 0.781**. Although responses to the original prompts showed slightly higher similarity scores, responses to paraphrased prompts also demonstrated high similarity (difference of only 0.04). This suggests that language models generate relatively consistent responses even when prompts are paraphrased.
> > >
> > > To better illustrate the observed similarities, we include concrete examples that highlight how model responses vary across paraphrased prompts and across different models. In some cases, models exhibit high-level conceptual similarity, for instance, in Example 1, all responses frame the metaphor around the idea that "time is a river." In other cases, the similarity is more surface-level; for example, in Example 2, many responses reuse the word “profound” in the opening sentence.
> > >
> > > **We hope these results help address the reviewer’s question. We plan to include both the quantitative findings and illustrative qualitative examples in the final camera-ready version. We are also happy to follow up on any additional questions the reviewer may have.**
> > >
> > > (continue examples in the next message due to space limit)

---

> > > > ### Author Response · Authors · 2025-08-07
> > > > **Follow-up on Q1 from the reviewer’s comments: Example Set 1**
> > > >
> > > > ## **Example Set 1: Write a metaphor involving time.**
> > > >
> > > > ### **Llama-3.1-8B-Instruct**
> > > >
> > > > > Q: Write a metaphor involving time.
> > > >
> > > > A: Time is a river that flows relentlessly, carrying memories and experiences along its currents, shaping the shores of our lives with every passing moment, and ultimately emptying into the vast ocean of eternity.
> > > >
> > > > > Q: Create a metaphor that relates to time.
> > > >
> > > > A: A river of moments flows through the expanse of existence, ever-changing and ever-present. Its waters, like grains of sand, slip through our fingers as the currents of time carry us along. Each eddy and ripple forms a distinct memory, while the relentless downstream drifts away with the passage of days.
> > > >
> > > > > Q: Compose a metaphor centered around time.
> > > >
> > > > A: Time is a river, its waters constantly flowing, carrying moments past, present, and future along with them. It carves paths through the landscape of our lives, wearing down the stones of yesterday, sculpting the shores of today, and shaping the mountains of tomorrow. Its currents twist and turn, sometimes swift, sometimes still, but always moving forward, carrying us with it, ever-changing, ever-flowing.
> > > >
> > > > > Q: Come up with a metaphor that involves time.
> > > >
> > > > A: Time is a river, flowing constantly and effortlessly, carrying moments, memories, and experiences with it.
> > > >
> > > > > Q: Generate a metaphor that features time.
> > > >
> > > > A: Time is a river, flowing relentlessly forward, shaping the landscape of our lives as it twists and turns, carving out moments and memories, sometimes gently, sometimes violently, but always relentlessly moving, carrying us along with it, leaving behind a trail of past and present, as we drift toward the unknown shores of the future.
> > > >
> > > >
> > > > ### **Qwen1.5-110B-Chat**
> > > >
> > > > > Q: Create a metaphor that relates to time.
> > > >
> > > > A: Time is like a river that flows constantly, carrying us along its current. Just as a river carves its path through the landscape, time shapes our lives and creates a unique journey for each of us. We can dip our toes in to pause and reflect, but we can never stop the flow or turn back to where we started. Every moment is a new downstream destination, and once it passes, it becomes a part of the vast ocean of memories.
> > > >
> > > > > Q: Compose a metaphor centered around time.
> > > >
> > > > A: Time is a river that flows relentlessly, carrying us forward on its current, while the memories of our past are like pebbles on its banks, polished smooth by the passage of each moment.", "Time is a river that flows incessantly, carrying us along its current, where moments are like pebbles polished smooth by its touch, and memories are the ripples that linger long after the waves have passed.
> > > >
> > > > > Q: Come up with a metaphor that involves time.
> > > >
> > > > A: Time is like a river that flows continuously, carrying us along its current. Just as a river carves its path through the earth, time shapes our lives and leaves its mark on us. We can dip our toes in to savor a moment, or let it sweep us away in a rush of events. But once it passes, we can never recapture the same moment again, for time's current is relentless and always moving forward.
> > > >
> > > > > Q: Generate a metaphor that features time.
> > > >
> > > > A: Time is a river that flows continuously, carrying us along its current, never pausing to let us catch our breath, but always moving us forward towards an unknown destination.
> > > >
> > > > ### **gemma-2-27b-it**
> > > >
> > > > > Q: Write a metaphor involving time.
> > > >
> > > > A: Time is a river, relentlessly flowing, carrying us all downstream. Some days we float peacefully on its surface, enjoying the sun, while others we're tossed by rapids, struggling to keep our heads above water. But no matter how we navigate its currents, we are all carried towards the inevitable sea.
> > > >
> > > > > Q: Create a metaphor that relates to time.
> > > >
> > > > A: Time is a river, constantly flowing, carrying us along its current. We can paddle and steer, but ultimately, we are swept towards the sea. \n\n\nLet me know if you'd like to explore other metaphors!
> > > >
> > > > > Q: Compose a metaphor centered around time.
> > > >
> > > > A: Time is a river, rushing onward, carrying us all on its current. Some days we paddle gently, enjoying the scenery. Others, we're caught in rapids, struggling to stay afloat. But the river flows ever onward, relentless and unwavering, towards the vast, unknown ocean of eternity.
> > > >
> > > > > Q: Come up with a metaphor that involves time.
> > > >
> > > > A: Time is a river, rushing forward relentlessly, carrying us all along its current. We can dip our toes in, swim against it for a while, or let it carry us to new shores, but ultimately, we cannot escape its flow. \n\n\nLet me know if you'd like to hear another one!\n
> > > >
> > > > > Q: Generate a metaphor that features time.
> > > >
> > > > A: Time is a river, rushing relentlessly towards the sea. We are all leaves, swept along its currents, some skimming the surface, others caught in eddies and whirlpools, but all ultimately heading towards the same destination. \n\n\nLet me know if you'd like another metaphor!

---

> > > > > ### Author Response · Authors · 2025-08-07
> > > > > **Follow-up on Q1 from the reviewer’s comments: Example Set 2**
> > > > >
> > > > > ## **Example Set 2: Write a paragraph about how the internet shaped society.**
> > > > >
> > > > > ### **Llama-3.3-70B-Instruct**
> > > > >
> > > > > > Q: Write a paragraph about how the internet shaped society.
> > > > >
> > > > > A: The internet has **profoundly** shaped society in countless ways, revolutionizing the way we communicate, access information, and interact with one another. [...]
> > > > >
> > > > > > Q: Compose a paragraph discussing the impact the internet has had on society.
> > > > >
> > > > > A: The advent of the internet has **profoundly** impacted society, revolutionizing the way we communicate, access information, and conduct our daily lives. [...]
> > > > >
> > > > > > Q: Draft a paragraph explaining the ways in which the internet has influenced society.
> > > > >
> > > > > A: The advent of the internet has **profoundly** impacted society, transforming the way we communicate, access information, and interact with one another. [...]
> > > > >
> > > > > > Q: Write a paragraph describing how society has been transformed by the internet.
> > > > >
> > > > > A: The advent of the internet has revolutionized modern society in **profound** ways, transforming the way we communicate, access information, and interact with one another. [...]
> > > > >
> > > > > > Q: Create a paragraph outlining the effects of the internet on society.
> > > > >
> > > > > A: The internet has **profoundly** impacted society, bringing about numerous transformative effects that have reshaped the way we live, interact, and access information. [...]
> > > > >
> > > > > ### **Qwen2.5-72B-Instruct**
> > > > >
> > > > > > Q: Write a paragraph about how the internet shaped society.
> > > > >
> > > > > A: The internet has **profoundly** reshaped society in numerous ways, transforming how we communicate, work, learn, and interact with the world. [...]
> > > > >
> > > > > > Q: Compose a paragraph discussing the impact the internet has had on society.
> > > > >
> > > > > A: The internet has **profoundly** transformed society in myriad ways, reshaping how we communicate, access information, and conduct business. [...]
> > > > >
> > > > > > Q: Draft a paragraph explaining the ways in which the internet has influenced society.
> > > > >
> > > > > A: The internet has **profoundly** transformed society in numerous ways, impacting nearly every aspect of human life. [...]
> > > > >
> > > > > > Q: Write a paragraph describing how society has been transformed by the internet.
> > > > >
> > > > > A: The internet has **profoundly** transformed society in myriad ways, redefining how we communicate, work, learn, and socialize. [...]
> > > > >
> > > > > > Q: Create a paragraph outlining the effects of the internet on society.
> > > > >
> > > > > A: The internet has **profoundly** transformed society in numerous ways, affecting communication, commerce, education, and social interactions. [...]
> > > > >
> > > > > A: The internet has **profoundly** transformed society in numerous ways, influencing how people communicate, work, learn, and entertain themselves. [...]
> > > > >
> > > > >
> > > > > ### **gemma-2-27b-it**
> > > > >
> > > > > > Q: Write a paragraph about how the internet shaped society.
> > > > >
> > > > > A: The internet has **profoundly** reshaped society, revolutionizing communication, access to information, and the way we interact with the world. [...]
> > > > >
> > > > > A: The internet has **profoundly** reshaped society, ushering in a new era of interconnectedness and information access. Its decentralized nature has democratized knowledge, allowing individuals to access vast repositories of information and connect with others across geographical boundaries. [...]
> > > > >
> > > > > > Q: Compose a paragraph discussing the impact the internet has had on society.
> > > > >
> > > > > A: The internet has revolutionized society in **profound** ways, impacting nearly every facet of human life. [...]
> > > > >
> > > > > A: The internet has **profoundly** reshaped society, revolutionizing communication, access to information, and social interaction. [...]
> > > > >
> > > > > > Q: Draft a paragraph explaining the ways in which the internet has influenced society.
> > > > >
> > > > > A: The internet has **profoundly** reshaped society, revolutionizing communication, information access, and daily life. [...]
> > > > >
> > > > > A: The internet has **profoundly** transformed society, revolutionizing communication, information access, commerce, and social interaction. [...]
> > > > >
> > > > > > Q: Write a paragraph describing how society has been transformed by the internet.
> > > > >
> > > > > A: The internet has fundamentally reshaped society, acting as a catalyst for **profound** changes in communication, commerce, information access, and social interaction. [...]
> > > > >
> > > > > > Q: Create a paragraph outlining the effects of the internet on society.
> > > > >
> > > > > A: The internet has **profoundly** transformed society, revolutionizing communication, information access, and social interactions. [...]
> > > > >
> > > > > A: The internet has **profoundly** reshaped society, fostering unprecedented connectivity and access to information. [...]

---

> > > > > > ### Comment · Reviewer_zt6D · 2025-08-07
> > > > > > **Please accept my sincerest appreciation of the further responses**
> > > > > >
> > > > > > Thank you for your very thorough discussion and addressing of my last few concerns. I'm ready to re-evaluate the work , and now instead leaning to endorse the acceptance of the paper.

---

> > > > > > > ### Author Response · Authors · 2025-08-08
> > > > > > >
> > > > > > > Dear reviewer `zt6D`,
> > > > > > >
> > > > > > > We are very glad that our results addressed your concerns, and we truly appreciate you raised the score to help support the acceptance of our paper!
> > > > > > >
> > > > > > > We would like to share a final note regarding your question about the *entropy of the open-ended query*. Thank you again for your constructive feedback throughout the review!
> > > > > > >
> > > > > > > ---
> > > > > > >
> > > > > > > ## **Additional Discussion: Entropy of Open-Ended Queries**
> > > > > > >
> > > > > > > Thank you for raising the important point that different open-ended questions may vary in how open-ended they truly are. You are absolutely right, and in response, we conducted additional human annotation during the rebuttal period to better quantify the level of open-endedness.
> > > > > > >
> > > > > > > ### Annotation Details
> > > > > > >
> > > > > > > We selected 100 evaluation examples from our dataset and recruited 86 participants via Prolific, assigning 3 participants per query. Participants met the following qualifications: English fluency, approval rate >99%, 500–10,000 prior submissions, and education beyond high school. Each participant was asked to answer the following  question:
> > > > > > >
> > > > > > > > **Q: How many different, reasonable alternative answers do you think could apply?**
> > > > > > >
> > > > > > > - **A:** Fewer than 3
> > > > > > > - **B:** 3 to 10
> > > > > > > - **C:** 10 to 20
> > > > > > > - **D:** More than 20
> > > > > > >
> > > > > > > To go beyond binary classification, we assessed the breadth of plausible responses. Annotators judged **81.27% of queries** as allowing **3 or more** valid alternatives, and **34.66%** as allowing **more than 20**. This reinforces the idea that real-world queries often support a wide spectrum of answers, emphasizing the need for diversity-aware LLM evaluation.
> > > > > > >
> > > > > > > | Answer           | Percentage (%) |
> > > > > > > |------------------|----------------:|
> > > > > > > | Fewer than 3     | 18.73           |
> > > > > > > | 3 to 10          | 31.87           |
> > > > > > > | 10 to 20         | 14.74           |
> > > > > > > | More than 20     | 34.66           |
> > > > > > >
> > > > > > > ---
> > > > > > >
> > > > > > > ## Analyzing Results by Level of Open-Endedness
> > > > > > >
> > > > > > > We examined how annotators perceived the *level of open-endedness* across prompts. Interestingly, we observed notable variation across individuals. Below is a sample of the annotations from three annotators per prompt:
> > > > > > >
> > > > > > > > **Sample Annotations**
> > > > > > > > `['More than 20', '10 to 20', '3 to 10']`
> > > > > > > > `['More than 20', '3 to 10', 'More than 20']`
> > > > > > > > `['More than 20', 'More than 20', 'More than 20']`
> > > > > > > > `['Fewer than 3', 'More than 20', 'More than 20']`
> > > > > > > > `['Fewer than 3', '10 to 20', 'More than 20']`
> > > > > > >
> > > > > > > This variability shows that assigning a definitive level of open-endedness to each query is nontrivial.
> > > > > > >
> > > > > > > Still, we performed a coarse-grained analysis using the following categorization:
> > > > > > >
> > > > > > > - If **any annotator** labeled a prompt as having *"More than 20 responses"*, we categorized it as **high open-endedness**.
> > > > > > > - If **no annotator** gave that label, we categorized it as **low open-endedness**.
> > > > > > >
> > > > > > > We then analyzed model response diversity using sentence similarity across 42 models. The average similarity scores were:
> > > > > > >
> > > > > > > - **High open-endedness**: 0.800
> > > > > > > - **Low open-endedness**: 0.837
> > > > > > >
> > > > > > > These results suggest that, even for prompts perceived as more open-ended, models do not always generate more diverse responses.
> > > > > > >
> > > > > > > We will include this analysis in the final camera-ready version too in the appendix. Thank you again for the thought-provoking and valuable question!

---

> ### Comment · Reviewer_zt6D · 2025-08-05
>
> Hey dear authors, as we are approaching the end of the discussion session, would you mind to share some, should you have any updates?

---

> > ### Author Response · Authors · 2025-08-05
> >
> > Dear Reviewer `zt6D`,
> >
> > Thank you for checking in! We’ve experienced a slight unexpected delay, but we’ll be posting updates tomorrow. We’ll make sure there is sufficient time to address any additional questions you may have. Thanks so much again for your thoughtful feedback!

---

> ### Public Comment · ~Claude_Ross1 · 2026-01-06
>
> I appreciate how carefully you went over my last few worries and how thoroughly you discussed them https://basketballstarsonline.io. Instead of reevaluating the work, I'm leaning toward endorsing the paper's acceptance.

---

### Official Review · Reviewer_1cqC · 2025-07-03

**Rating:** 6
**Confidence:** 5

**Summary:**

This paper presents $\infty$-CHATS, a large-scale benchmark that contains 26K real-world, open-ended user queries and 31K human annotations, designed to evaluate the diversity and pluralism of language model outputs. The authors propose a comprehensive evaluation framework that (1) builds a novel taxonomy of open-ended prompts, (2) introduces the concept of the “Artificial Hivemind” to characterize intra- and inter-model homogenization; then, they analyze model behavior across 70+ LMs, measure mode collapse, and evaluate alignment with dense human preferences. The paper experiments the leading LLMs on this dataset and shows severe repetition across both individual and multiple models. They highlight trends of homogenization and misalignment in open-ended generation for current language models.

**Dataset Code Accessibility:**

Yes

**Ethical Considerations:**

No, there are no or only very minor ethics concerns

**Final Justification:**

I think this is a well-qualified work and would interest the community. I will keep the high score.

**Limitations Weaknesses:**

I see the following minor issues:

i. The prompt taxonomy and query sources are somewhat limited to English web queries and manually filtered data. This may constrain the generalizability of findings across cultures and languages.

ii. How to interpret and adopt the Artificial Hivemind metric for model development or training feedback should be better presented. This current formulation is insightful but the authors may add more actionable guidance.

iii. Although the method focuses on evaluating diversity and human preference alignment, the analysis does not clearly provide sufficient insights into the underlying causes of inter-model homogenization.

But overall, this work is valuable to the LM community.

**Strengths Contributions:**

The proposed benchmark has been meticulously constructed. It develops a rich collection of open-ended prompts and high-density human preference annotations of real-world user queries. Further, the paper offers a fine-grained and comprehensive evaluation framework that measures response diversity, human alignment, and inter-model homogenization.

Overall this paper is well-written and easy to follow. The figures clearly illustrate prompt diversity, homogenization trends across model families, and alignment gaps across preference dimensions. Further, the code is well structured and documented.

The experimental results underscore the benchmark's challenges for current aligned LLMs and yield valuable insights for future research. Notably, the critical role of capturing human pluralism in open-ended generation is highlighted, along with the inherent challenges posed by mode collapse, redundancy, and the tension between alignment and diversity.

---

> ### Author Rebuttal · Authors · 2025-07-31
>
> We thank reviewer `1cqC` for their highly positive and constructive feedback! We especially appreciate their recognition that “this paper is well-written and easy to follow,” that our dataset “has been meticulously constructed,” and that we “develop a rich collection of open-ended prompts and high-density human preference annotations of real-world user queries.” We also value their observation that “the experimental results underscore the benchmark's challenges for current aligned LLMs and yield valuable insights for future research.”
>
> Below, we provide point-by-point responses to their comments and questions, and we would be glad to address any further clarifications during the discussion period!
>
> ## **Discussion of the limitation of English-only data and cross-cultural implications**
>
> Thanks for bringing up this point! We agree that focusing on English web queries constrains generalizability. While extending infinite-CHATS to multilingual and multicultural settings is an important direction, we anticipate that similar homogenization issues will likely arise across languages and cultures, given the global overlap in pretraining data sources and alignment practices. We encourage future work to pursue such extensions, and we hope our benchmark provides a comprehensive foundation that multilingual research can build upon. Our taxonomy was designed to be language-agnostic and can support adaptation when appropriate resources and expertise are available. We'll further strengthen this discussion in camera-ready.
>
> ## **Discussion of actionable steps for leveraging the Artificial Hivemind metric**
>
> Thanks for requesting more concrete steps on how our resources can support the improvement of future models! In the camera-ready version, we will expand our discussion to include actionable uses, in addition to our current formulation that provides diagnostic insight into homogenization. Specifically:
>
> - **Integration into red-teaming workflows** to stress-test how language models handle open-endedness.
> - **Leveraging our dataset** as a rich source of open-ended prompts for developing novel RL approaches that explicitly reward diverse mode exploration during alignment.
> - **Benchmarking decoding strategies** (e.g., diverse beam search, nucleus sampling variants) under the Artificial Hivemind metric to assess their effectiveness in mitigating homogenization.
> - **Providing diagnostic signals for curriculum design**, enabling models to be gradually exposed to increasingly open-ended prompts that are prone to homogenization.
>
> Together, these steps position the Artificial Hivemind as not only a descriptive tool but also a practically actionable resource for improving both the evaluation and training of language models. We will incorporate this discussion into the final camera-ready.
>
> ## **Clarification around underlying causes of inter-model homogenization**
>
> Thank you for suggesting more explicit causal analysis between model mechanism and homogenization effects. The primary goal of our paper is to diagnose the phenomenon of inter-model homogenization, for which we provide an exhaustive analysis. To complement this, we will expand our discussion to highlight potential contributing factors: (1) substantial overlap in pretraining corpora, (2) alignment processes that may overlook diversity due to the absence of measurable evaluation resources, and (3) convergence in decoding strategies such as temperature-controlled sampling with similar defaults. We also note that reliance on shared reinforcement learning or preference datasets may further amplify similarity. While a full causal tracing is beyond the scope of this work and merits dedicated investigation, we believe that including these discussions in the final camera-ready will provide a strong foundation and inspire further exploration by the community.

---

> > ### Comment · Reviewer_1cqC · 2025-08-05
> >
> > Thank you for your reply. I think this is a well-qualified work and would interest the community.

---

> > > ### Author Response · Authors · 2025-08-07
> > >
> > > Thank you for your positive evaluation and constructive feedback! We will incorporate the revisions discussed throughout the rebuttal period into the final camera-ready version of the paper.

---

### Comment · Area_Chair_PFyH · 2025-08-04

Dear Reviewers,

As the rebuttal period approaches, please read the authors' rebuttal and submit your acknowledgement. If possible, please kindly let the authors know your remaining concerns.

Best,

AC

---

### Author Response · Authors · 2025-08-09
**Thank you for all reviewers!**

As we approach the end of the discussion period, we would like to sincerely thank all the reviewers for their constructive feedback, dedication in helping us improve our paper during the discussion, and support for its acceptance. We are grateful for the opportunity to address your questions, and we are committed to incorporating all the valuable points raised during the rebuttal and discussion period into the camera-ready version. We truly appreciate your time, expertise, and thoughtful engagement with our work.

---

### Decision · Program_Chairs · 2025-09-18

**Decision:**

Accept (oral)

**Comment:**

**Summary**:

This paper presents $\infty$-CHATS, a large-scale benchmark comprising 26K real-world, open-ended user queries and 31K human annotations, designed to evaluate diversity and pluralism in language model outputs. The authors propose a comprehensive evaluation framework that (1) develops a novel taxonomy of open-ended prompts, and (2) introduces the concept of the Artificial Hivemind to characterize intra- and inter-model homogenization. They evaluate over 70 LMs using this framework, analyzing response diversity, measuring mode collapse, and assessing alignment with dense human preferences. Experiments reveal severe repetition both within and across models, highlighting concerning trends of homogenization and misalignment in open-ended generation.

**Strengths**:

- The benchmark is meticulously constructed, offering a rich set of open-ended prompts and high-density human preference annotations.

- The evaluation framework is fine-grained and comprehensive, measuring response diversity, human alignment, and inter-model homogenization.

- The paper is well-written, with clear figures that effectively illustrate prompt diversity, homogenization trends, and alignment gaps. The released code is well-structured and well-documented.

- Experimental results highlight critical challenges for aligned LLMs, particularly the tension between alignment and diversity, while underscoring the importance of capturing human pluralism.

- The testbed hybridizes automatically crafted and hand-written queries (for blindspot coverage), revealing the “artificial hivemind” effect—an important finding that calls for further community attention.

**Weaknesses**:

- Limited discussion of the dataset’s English-only scope and its cross-cultural implications.

- Limited guidance on actionable steps for applying the Artificial Hivemind metric.

- The results would benefit from deeper interpretation and discussion.

- The new human annotation results introduced in the rebuttal should be fully included in the camera-ready version to strengthen the grounding and definition of open-endedness.

During the rebuttal, the authors addressed most concerns raised by reviewers, leading some reviewers to increase their ratings. Overall, this is a strong and timely contribution that will be of significant interest to the community. I therefore recommend acceptance.